# Fast heterogeneous $N_2O_5$ uptake and $ClNO_2$ production in coal-fired plumes observed in the nocturnal residual layer over the North China Plain

Zhe Wang[1], Weihao Wang[1], Yee Jun Tham[1, a], Qinyi Li[1], Hao Wang[2], Liang Wen[2], Xinfeng Wang[2], Tao Wang[1]

[1]Department of Civil and Environmental Engineering, The Hong Kong Polytechnic University, Hong Kong, China
[2]Environment Research Institute, Shandong University, Jinan, China
[a]Now at: Department of Physics, University of Helsinki, Finland

*Correspondence to*: Zhe Wang (z.wang@polyu.edu.hk), Tao Wang (cetwang@polyu.edu.hk)

**Abstract.** Dinitrogen pentoxide ($N_2O_5$) and nitryl chloride ($ClNO_2$) are key species in nocturnal tropospheric chemistry, and have significant effects on particulate nitrate formation and the following day's photochemistry through chlorine radical production and $NO_x$ recycling upon photolysis of $ClNO_2$. To better understand the roles of $N_2O_5$ and $ClNO_2$ in the high aerosol loading environment of northern China, an intensive field study was carried out at a high-altitude site (Mt. Tai, 1465 m a.s.l) in the North China Plain (NCP) during the summer of 2014. Elevated $ClNO_2$ plumes were frequently observed in the nocturnal residual layer with a maximum mixing ratio of 2.1 ppbv (1-min), whilst $N_2O_5$ was typically present at very low levels (<30 pptv), indicating fast heterogeneous $N_2O_5$ hydrolysis. Combined analyses of chemical characteristics and backward trajectories indicated that the $ClNO_2$-laden air was caused by the transport of $NO_x$-rich plumes from the coal-fired industry and power plants in the NCP. The heterogeneous $N_2O_5$ uptake coefficient ($\gamma$) and $ClNO_2$ yield ($\phi$) were estimated from steady-state analysis and observed growth rate of $ClNO_2$. The derived $\gamma$ and $\phi$ exhibited high variability, with means of $0.061 \pm 0.025$ and $0.28 \pm 0.24$, respectively. These values are higher than those derived from previous laboratory and field studies in other regions, and cannot be well characterized by model parameterizations. Fast heterogeneous $N_2O_5$ reactions dominated the nocturnal $NO_x$ loss in the residual layer over this region, and contributed to substantial nitrate formation of up to 17 $\mu g\ m^{-3}$. The estimated nocturnal nitrate formation rates ranged from 0.2 to 4.8 $\mu g\ m^{-3}\ h^{-1}$ in various plumes, with a mean of $2.2 \pm 1.4$ $\mu g\ m^{-3}\ h^{-1}$. The results demonstrate the significance of heterogeneous $N_2O_5$ reactivity and chlorine activation in the NCP, and their unique and universal roles in fine aerosol formation and $NO_x$ transformation, and thus potential impacts on regional haze pollution in northern China.

# 1 Introduction

Nitrogen oxides ($NO_x = NO + NO_2$) plays central roles in the oxidative capacity of the atmosphere and photochemical air pollution. Dinitrogen pentoxide ($N_2O_5$) is an important reactive intermediate in the oxidation of $NO_x$, and exists in rapid thermal equilibrium with nitrate radical ($NO_3$) formed via the reaction between $NO_2$ and $O_3$. The heterogeneous hydrolysis of $N_2O_5$ has been recognized as a key step in nocturnal $NO_x$ removal, and can affect regional air quality by regulating the reactive nitrogen budget and nitrate aerosol formation (e.g., Brown et al., 2006; Abbatt et al., 2012). The heterogeneous reaction of $N_2O_5$ on and within atmospheric aerosols, fog or cloud droplets, produces soluble nitrate ($HNO_3$ or $NO_3^-$) and nitryl chloride ($ClNO_2$) when chloride is available in the aerosols (Finlayson-Pitts et al., 1989).

$$N_2O_5 + H_2O \ (het) \rightarrow 2HNO_3(aq) \qquad (R1)$$

$$N_2O_5 + Cl^- \ (het) \rightarrow NO_3^-(aq) + ClNO_2 \qquad (R2)$$

The rate coefficient of the heterogeneous $N_2O_5$ reactions is governed by the available reaction surface and $N_2O_5$ reaction probability (also known as the uptake coefficient $\gamma_{N2O5}$), and can be described by the following expression when the gas-phase diffusive effect is negligible.

$$k(N_2O_5)_{het} = \frac{1}{4} c_{N2O5} \, \gamma_{N2O5} \, S_a \qquad (1)$$

Here, $c_{N2O5}$ is the mean molecular speed of $N_2O_5$, and $S_a$ is the aerosol (or cloud) surface area density. The yield of $ClNO_2$ ($\phi$) is defined as the amount of $ClNO_2$ formed per loss of $N_2O_5$, representing the fraction to $ClNO_2$ formation. Hence, the net reaction of R1 and R2 can be written as follows:

$$N_2O_5 + (H_2O \text{ or } Cl^-)(het) \rightarrow (2 - \phi) \, NO_3^-(aq) + \phi \, ClNO_2 \qquad (R3)$$

The $\gamma_{N2O5}$ has been experimentally measured on various types of aerosols surfaces (including sulfate, nitrate, black carbon, organic carbon, organic coating sulfate, sea salts, and dust, etc.) in the laboratory, and different parameterizations based on aerosol composition have been proposed in varying degree of complexity (e.g., Evans and Jacob, 2005; Anttila et al., 2006; Davis et al., 2008; Bertram and Thornton, 2009; Griffiths et al., 2009; Riemer et al., 2009; Roberts et al., 2009; Simon et al., 2009; Foley et al., 2010; Chang et al., 2011; Ammann et al., 2013; Tang et al., 2014). Recently, field studies have been carried out to measure ambient $N_2O_5$ and to derive the $\gamma_{N2O5}$ from atmospheric observations (e.g., Bertram et al., 2009b; Brown et al., 2009; Morgan et al., 2015; Brown et al., 2016; Chang et al., 2016; Phillips et al., 2016). These field-derived/measured $\gamma_{N2O5}$ values were found to vary considerably, and the observed range to be significantly larger than that from laboratory studies using synthetic aerosols (Chang et al., 2011; Phillips et al., 2016). Furthermore, inconsistencies between $\gamma_{N2O5}$ values derived from field measurements and parameterizations were observed in some locations, which implies that $\gamma_{N2O5}$ has a complex dependence on the aerosol composition, physico-chemical characteristics, and environmental parameters (Chang et al., 2011 and references therein). Similarly, for the $ClNO_2$ yield, the field-determined values exhibited significant variability, ranging from 0.01 to close to unity (Thornton et al., 2010; Riedel et al., 2013; Wagner et al., 2013; Phillips et al., 2016), which could

not be well reproduced (exhibiting a tenfold difference in some cases) by parameterization based on only aerosol chloride and water content (Wagner et al., 2013; Wang et al., 2017b). There are only few studies on the determination of $\phi$ from field measurement, and the possible effects of real atmospheric aerosols (including organic composition, mixing state, and chloride partitioning between particle sizes, etc.) have not been well characterized (Mielke et al., 2013; Phillips et al., 2016). This

incomplete understanding suggests the necessity of more field measurements of $\gamma$ and $\phi$ in various environments, to facilitate the validation and construction of parameterizations suitable for use in air quality models.

$ClNO_2$ formed from nocturnal heterogeneous $N_2O_5$ uptake can potentially affect the atmospheric oxidative capacity via the production of highly reactive chlorine radicals (Cl) and the recycling of $NO_x$ after photolysis (Simpson et al., 2015) . Elevated $ClNO_2$ mixing ratios were firstly observed in several polluted coast regions (for instance, the coasts of Texas and California,

and the Los Angeles Basin), resulting from the strong emission of $NO_x$ and abundant chloride from sea salt aerosols (Osthoff et al., 2008; Riedel et al., 2012; Mielke et al., 2013; Tham et al., 2014). Recently, significant $ClNO_2$ production was also observed in some inland areas (such as Colorado, Hessen, and Alberta), with mixing ratio up to several hundreds of pptv or even exceeding 1.0 ppbv (e.g., Thornton et al., 2010; Mielke et al., 2011; Phillips et al., 2012; Riedel et al., 2013; Faxon et al., 2015; Mielke et al., 2016). Anthropogenic sources of chlorine including coal combustion in power plants, industries, and

biomass burning may potentially facilitate $ClNO_2$ production (Riedel et al., 2013). The highest $ClNO_2$ mixing ratio yet reported ($4.7 \pm 0.8$ ppbv, 1-min average) was recently observed in the regional plumes at a mountaintop site in southern China, indicating the importance of $N_2O_5/ClNO_2$ chemistry in polluted environments (Wang et al., 2016).

Large anthropogenic emissions of $NO_x$ and increasing $O_3$ concentrations have been reported in many urban cluster regions in China (Wang et al., 2006; Wang et al., 2017a). Hence, in these regions, nocturnal nitrogen chemistry may be particularly

important in the transformation of $NO_x$ and the subsequent effects on daytime photochemistry and secondary aerosol formation. In the areas downwind of Beijing and Shanghai, high concentrations of particulate nitrate (up to 40 μg m³) have been observed and attributed to heterogeneous $N_2O_5$ uptake on acidic aerosols (Pathak et al., 2009; Pathak et al., 2011). During a more recent field study in a rural site in the North China Plain (NCP), elevated fine nitrate concentrations were observed at night and in the early morning, with hourly maxima of up to 87.2 μg m³ and a 30% contribution to $PM_{2.5}$, which was mainly attributed to the

heterogeneous hydrolysis of $N_2O_5$ (Wen et al., 2015). Active heterogeneous $N_2O_5$ chemistry has been recently characterized in both rural and urban areas of the NCP via direct measurements of $N_2O_5$ and $ClNO_2$. Rapid heterogeneous $N_2O_5$ loss and efficient $ClNO_2$ production were observed, with a maximum $ClNO_2$ mixing ratio of 2.07 ppbv at Wangdu and 0.77 ppbv at Jinan (Tham et al., 2016; Wang et al., 2017b). Moreover, sustained $ClNO_2$ peaks were observed after sunrise in the region, and the downward mixing of $ClNO_2$-rich air in the residual layer was proposed to be the cause of morning peaks (Tham et al.,

2016). To confirm these findings and better characterize the chemistry of $N_2O_5/ClNO_2$ and their impacts on regional air quality, it is of great interest to conduct direct field measurements of $N_2O_5/ClNO_2$ in the polluted residual layer.

In the present study, we measured the concentrations of $N_2O_5$, $ClNO_2$, and related species at a mountaintop site in the NCP during the summer of 2014, and characterized the nighttime nitrogen chemistry within the residual layer over a polluted region

of northern China. We examined the frequently intercepted $ClNO_2$-rich plumes at this high elevation site, and investigated nocturnal $N_2O_5$ reactivity to determine the heterogeneous $N_2O_5$ uptake coefficients and $ClNO_2$ yields in a variety of air masses, which were also compared to parameterizations utilized in existing models. The effects of heterogeneous $N_2O_5$ chemistry on particulate nitrate formation and nocturnal $NO_x$ loss were then evaluated based on the observation data.

## 2. Methodology

### 2.1 Field Study Site

The measurement site was located on Mount Tai (36.25°N, 117.10°E, 1465 m above sea level) in Shandong Province, China. Mt. Tai is located between the two most developed regions in China (Jing-Jin-Ji and the Yangtze River Delta), and its peak (1545 m a.s.l.) is the highest point within the NCP. Figure 1 shows the location of the measurement site in relation to the surrounding topography. Mt. Tai is 230 km away from the Bohai and Yellow Seas, and the cities of Tai'an and Jinan (the capital of Shandong Province) are located 15 km south and 60 km north of the measurement site, respectively. The altitude of the measurement site is near the top of the boundary layer in the daytime during the summer, and is typically in the residual layer or, occasionally, in the free troposphere at night. This mountaintop site has been previously used in many atmospheric chemistry field studies (e.g., Gao et al., 2005; Wang et al., 2011; Guo et al., 2012; Sun et al., 2016). Previous studies at this site indicated that the site is regionally representative without significant local anthropogenic emissions, and affected by the regional aged air masses and occasional combustion plumes from fossil fuel or biomass in the region (e.g., Zhou et al., 2009; Wang et al., 2011, Guo et al., 2012). Intensive measurements were performed from July 24 to August 27, 2014. During this period, the prevailing winds originated from the northeast and northwest. Shandong province is the largest producer of thermal power in China, and dozens of coal-fired industry and power plants are situated within a radius of 200 km from the mountain site.

### 2.2 Instrumentation

$N_2O_5$ and $ClNO_2$ were measured concurrently using iodide ion chemical ionization mass spectrometry (CIMS) with a quadrupole mass spectrometer (THS Instruments Inc., USA). The principle and detailed calibration of this CIMS system have been described previously by Wang et al. (2016) and Tham et al. (2016). The same configuration was used in the present study. Briefly, $N_2O_5$ and $ClNO_2$ were detected as $I(N_2O_5)^-$ and $I(ClNO_2)^-$ clusters via reaction with iodide ions ($I^-$), which were generated from a mixture of $CH_3I$ (0.3% v/v) and $N_2$ using an alpha radioactive source, $^{210}Po$ (NRD, P-2031-2000). The inlet was installed ~ 1.5 m above the roof of a single-story building, and the sampling line was a 5.5 m PFA-Teflon tubing (1/4 in. o.d.) which was replaced daily in the afternoon before sunset and washed in the ultrasonic bath to minimize wall loss caused by deposited particles (Wang et al., 2016). A small proportion (1.7 SLPM) of total sampling flow (~ 11 SLPM) was diverted to the CIMS system, to reduce the residence time of the air samples in the sampling line. A standard addition of $N_2O_5$ into the ambient inlet was performed before and after the tubing replacement to monitor the transmission efficiency, and this practice

limited the loss of $N_2O_5$ in the inlet to <10% in the 'clean' tubing and about 30% in the next afternoon. Manual calibrations of $N_2O_5$ and $ClNO_2$ were conducted daily to determine the instrument sensitivity, the average of which during the observation period was 2.0 ± 0.6 for $N_2O_5$ and 2.2 ± 0.6 Hz pptv$^{-1}$ for $ClNO_2$, respectively. The $N_2O_5$ standard was synthesized on-line from the reaction between $NO_2$ and $O_3$, and the produced $N_2O_5$ were determined from the decrease in $NO_2$ (Wang et al., 2014).

This method has been validated with a Cavity Ring Down Spectrometer (CRDS) measurement in previous campaign (Wang et al., 2016). The $ClNO_2$ was produced by passing a known concentration of $N_2O_5$ through a NaCl slurry assuming unity conversion efficiency (Roberts et al., 2009) and negligible $ClNO_2$ loss in the system (Wang et al., 2016). The field background was determined by passing the ambient sample through a filter packed with activated carbon, with average levels of 7.8 ± 1.9 and 6.0 ± 1.6 Hz for $N_2O_5$ and $ClNO_2$, respectively. The reported concentrations were derived by subtracting the background

levels. The detection limit was 4 pptv for both $N_2O_5$ and $ClNO_2$ (2σ, 1 min-averaged data), and the uncertainty of the nighttime measurement was estimated to be ± 25% (Tham et al., 2016).

The related trace gases and aerosol compositions were also measured concurrently during the campaign. All of the instruments were used in our previous field studies, and the setup, precision, and accuracies of these instruments were described previously (Wen et al., 2015; Tham et al., 2016; Wang et al., 2016; Wang et al., 2017b). Briefly, NO and $NO_2$ were measured using a

chemiluminescence analyzer equipped with a blue-light converter (TEI, Model 42I-TL). Total gaseous reactive nitrogen ($NO_y$) was determined using a chemiluminescence analyzer with an external molybdenum oxide (MoO) catalytic converter (TEI, Model 42CY) with an inlet filter. The $NO_y$ described here is different from that in previous reports (Tham et al., 2016; Wang et al., 2016), because that the particulate nitrate was not included but removed by the filter in the present study. $O_3$, $SO_2$, and CO were measured using the ultraviolet photometry, pulsed-UV fluorescence, and IR photometry techniques (TEI, Model 49I,

43C and 48C), respectively. Zero and span calibrations for trace gases were performed weekly during the campaign. Water-soluble ionic compositions of $PM_{2.5}$ (including $NH_4^+$, $Na^+$, $Ca^{2+}$, $Mg^{2+}$, $Cl^-$, $SO_4^{2-}$, and $NO_3^-$) were measured hourly by a Monitor for Aerosols and Gases (MARGA ADI 2080, Applikon-ECN) using on-line ion chromatography.

The particle number and size distribution (5 nm to 10 μm) were measured using a Wide-Range Particle Spectrometer (WPS, Model 1000XP, MSP Corporation, USA). The particle diameters were corrected for particle hygroscopicity to determine the

actual ambient aerosol surface density, and the wet diameters were calculated using growth factors from a size-resolved kappa-Köhler function obtained in a rural site in the NCP (Ma et al., 2016; Tham et al., 2016). The uncertainties associated with the aerosol surface area determination was estimated to be around 30% (Liu et al., 2010; Tham et al., 2016). Meteorological data, including temperature, relative humidity (RH), wind vectors and photolysis frequency of $NO_2$ ($J_{NO2}$) were measured by an automated meteorological station (JZYG, PC-4) and a filter radiometer (Metcon, Germany). In addition, a Lagrangian particle

dispersion model, Hybrid Single-Particle Lagrangian Integrated Trajectory (HYSPLIT) model (Draxler and Hess, 1998; Wang et al., 2016), driven by high spatial and temporal meteorological fields from the WRF model, was used to investigate potential source regions of the air masses intercepted at the measurement site. The HYSPLIT model was run 12-h backward with 2500

particles released at the measurement site. Detailed parameterization and setup of the HYSPLIT and WRF models were previously described by Wang et al. (2016) and Tham et al. (2016).

## 3. Results and Discussion

### 3.1 Overview of $N_2O_5$ and $ClNO_2$ measurement

The temporal variations of $ClNO_2$, $N_2O_5$, related trace gases, aerosol properties, and selected meteorological parameters during the field study at Mt. Tai are depicted in Figure 2. Overall, the observed mixing ratios of $ClNO_2$ were higher than those of $N_2O_5$, and exhibited significant variations. The average mixing ratios of $N_2O_5$ and $ClNO_2$ were $6.8 \pm 7.7$ pptv and $54 \pm 106$ pptv, respectively. The maximum mixing ratio of $N_2O_5$ (167 pptv) was observed at 21:00 on August 26, 2014, and most of the other nights during the observation period exhibited peak $N_2O_5$ mixing ratios below 30 pptv. The average nighttime mixing ratios of $O_3$ and $NO_2$ were 77 and 3.0 ppbv, respectively, with an average nitrate radical production rate $p(NO_3)$ of $0.45 \pm 0.40$ ppb h$^{-1}$, which is indicative of potentially active $NO_3$ and $N_2O_5$ chemistry during the study period. However, the low $N_2O_5$ mixing ratios observed during most of the nights suggest a rapid loss of $N_2O_5$, which is consistent with the observed high aerosol surface area ($S_a$), varied from ~100 to 7800 $\mu m^2$ cm$^{-3}$ with a mean value of 1440 $\mu m^2$ cm$^{-3}$. The higher RH during nighttime and the frequent occurrence of clouds at the mountaintop site could also account for low $N_2O_5$ concentrations, because of its rapid heterogeneous loss on cloud droplets.

The highest $ClNO_2$ mixing ratio of 2065 pptv was observed on August 8, 2014, and on 8 of the 35 nights the peak $ClNO_2$ mixing ratios were higher than 500 pptv. The simultaneous increases of $SO_2$, $NO_x$ and CO with these $ClNO_2$ peaks suggest these air masses originated from coal combustion sources, such as industry and power plants, which will be further discussed in the next section. The elevated $ClNO_2$ levels observed at Mt. Tai are similar to recent measurements at a surface rural site (Wangdu) in northern China (Tham et al., 2016) and a mountain site (Tai Mo Shan) in southern China (Wang et al., 2016), but are slightly higher than previous measurements conducted in coastal (e.g., Osthoff et al., 2008; Riedel et al., 2012; Mielke et al., 2013) and inland sites (e.g., Thornton et al., 2010; Phillips et al., 2012; Riedel et al., 2013) in other regions of the world. During the campaign at Mt. Tai, the average concentrations of aerosol sulfate and nitrate were $14.8 \pm 9.0$ and $6.0 \pm 4.7$ $\mu g$ m$^{-3}$, accounting for 29.5% and 12.0% of $PM_{2.5}$ mass, respectively. The aerosol organic-to-sulfate ratio, a parameter that potentially affects the uptake process (Bertram et al., 2009b), was 0.74 on average and much lower than those from studies mentioned above in Europe and US. Moreover, the nighttime averaged Cl$^-$ concentration was $0.89 \pm 0.86$ $\mu g$ cm$^{-3}$, and was an order of magnitude higher than Na$^+$, indicating abundant non-oceanic sources of chloride (e.g., from coal combustion and biomass burning in the NCP) (Tham et al., 2016), which could enhance the production of $ClNO_2$. The mean diurnal variations of $N_2O_5$, $ClNO_2$, and other relevant chemical species during the study period are shown in Figure 3. Ozone exhibited a typical diurnal pattern for a polluted mountaintop site (Sun et al., 2016), and it began to increase in the late morning and reached an afternoon peak of 88.6 ($\pm$ 16.6) ppbv with a daily average rise of 24.4 ppbv. The average $O_3$ kept at elevated levels after sunset

and did not begin to decrease until 22:00, and $NO_x$ exhibited a diel maximum of 6.1 ppbv before sunset, resulting in a peak in $p(NO_3)$ just before sunset and relatively high levels in the early night. Gaseous $NO_y$ reached a maximum of 16.4 ($\pm$ 6.1) ppbv in the morning, and remained stable at a high level during the daytime; the air masses were more aged during the daytime, as indicated by the persistent low $NO_x/NO_y$ ratios (0.2-0.25). Small $N_2O_5$ peaks were observed immediately after sunset, resulting from the abundant $O_3$ and $NO_2$, and was present at low levels near to the detection limit of the CIMS throughout the rest of the night. $ClNO_2$ exhibited clear nighttime elevations resulting from the heterogeneous production after sunset, and reached a diel maximum around midnight. The low $N_2O_5$ and high $ClNO_2$ concentrations observed at Mt. Tai are similar to the measurement at a rural surface site within the NCP (Tham et al., 2016), suggesting rapid heterogeneous loss of $N_2O_5$ and production of $ClNO_2$ in this region.

It was also noted that a small $N_2O_5$ peak (~10 pptv) with larger variability was present in the early afternoon. A simplified photostationary analysis following Brown et al. (2005; 2016) was performed to predict the daytime steady-state $N_2O_5$ concentrations for the few cases with daytime peaks. The predicted concentrations all showed increasing trends in the afternoon, similar to the observation pattern. However, for individual cases, the absolute values around 15:00 were much lower than observation under clean sky condition, but of the same magnitude as the observation for reduced photolysis and foggy conditions with higher $NO_3$ production rate (c.f. Figure S1 in the supplement). Daytime $N_2O_5$ signals with few pptv have also been observed by a CRDS at a mountain site in southern China (Brown et al., 2016), where the concentrations were in accord with steady state estimation in an average sense. Because daily maintenance and calibrations of the CIMS were usually performed during early afternoon periods, the limited daytime data in the present study was not sufficient to make clear whether there were any daytime interferences or sensitivity fluctuations. Thus additional studies are needed to validate the daytime phenomenon and examine the potential reasons, and the following analysis in the present work will mostly focus on nocturnal process.

## 3.2 High $ClNO_2$ plumes from coal-fired point sources

As described above, several plumes with elevated $ClNO_2$ concentrations (> 500 pptv) were observed during the measurement period. Figure 4a illustrates the high $ClNO_2$ case observed during the night of July 30-31, 2014. The $ClNO_2$ concentration peaked sharply at 1265 pptv, which was accompanied by a steep rise in the concentrations of $SO_2$, $NO_x$ and CO. The $SO_2/NO_y$ ratio increased from ~0.1 to 0.6 in the plume center, with a $\Delta SO_2/\Delta NO_y$ slope of 0.57, indicating the coal combustion source of the plume. The coincident increase in $CO/NO_y$ ratio from ~30 to 90 suggests that it was likely originated from coal-fired industry facilities, such as cement and steel production plants, which is the largest emitting sector of CO in north China (Streets et al., 2006; Zhang et al., 2009). The 12-h backward particle dispersion trajectories calculated from the HYSPLIT model revealed that the air masses mostly moved slowly from the west, and passed over the region with cement and steel production industry and power plants before arriving at the measurement site. Figure 5a shows the highest $ClNO_2$ case (2065 ppbv) observed on the night of August 8, 2014. The simultaneous increases in $SO_2$, $NO_x$ and CO concentrations, together with the

higher $SO_2/NO_y$ ratio (~0.5) comparing to that outside of plume (~0.1) and the campaign average (0.24), again indicate the coal combustion origin of the plume. The relatively lower $CO/NO_y$ ratio of ~50 possibly suggests the plume affected by power plant emission, as shown by the derived backward particle dispersion trajectories. Table 1 summarizes the chemical characteristics of the eight cases of high-$ClNO_2$ coal-fired plumes during the study period. In these cases, the average $SO_2$ mixing ratios ranged from 2.3 to 18.7 ppbv, and the maximum $ClNO_2$ and $N_2O_5$ mixing ratios ranged from 534 to 2065 ppbv and 7.3 to 40.1 ppbv, respectively, with corresponding $ClNO_2/N_2O_5$ ratios of 25 to 118. The mixing ratios for $O_3$ and $NO_2$ ranged from 60 to 106 ppbv and 2.8 to 11.8 ppbv, respectively, resulting in high $p(NO_3)$ values of 0.60 to 1.59 ppbv h$^{-1}$. The aerosol chloride concentration ranged from 1.01 to 2.34 μg cm$^{-3}$, which was higher than the nighttime average (0.89 μg cm$^{-3}$) and conducive to $ClNO_2$ production from R3.

$NO_x$ emissions from the coal combustion sources contain abundant NO, which is oxidized rapidly to $NO_2$ by ambient $O_3$. Thus, the anti-correlation between $O_3$ and $NO_2$ within the observed plumes (cf. Figure 4b and 5b) can be another indicator of the large combustion sources (such as coal-fired power or industry plants). Furthermore, the slope of a plot of $O_3$ vs. $NO_2$ for nighttime plumes can be considered as an approximate measure of the plume age, with the assumption of pseudo-first-order kinetics and when the input of $NO_x$ is small comparing to the excess $O_3$ (Brown et al., 2006). The estimated plume age can be determined as follows:

$$t_{plumes} \approx \ln(1 - S(m + 1))/(Sk\overline{O_3}) \qquad (2)$$

where $m$ is the derived slope, $k$ is the rate coefficient for the reaction of $NO_2$ with $O_3$, $\bar{O}_3$ is the average $O_3$ concentration in the plume, and $S$ is a stoichiometric factor that varies between 1 for dominant $NO_3$ loss and 2 for dominant $N_2O_5$ loss (Brown et al., 2006). In the present study, heterogeneous $N_2O_5$ uptake dominated the reactive nitrogen loss, therefore $S = 2$ was used in the calculation. The plume ages for the July 30-31 and August 8 cases were calculated to be 3.2 and 2.1 h, respectively, which are consistent with the moderate $NO_x/NO_y$ ratios of 0.4-0.5, and comparable to those observed in nocturnal power plant plumes in the eastern coast of the USA (Brown et al., 2006; Brown et al., 2007). The slopes of $O_3$ vs. $NO_2$ in Figure 4b and 5b steeper than -1.0 also indicate the further reactions of $NO_2$ with $O_3$, which favor the formation of $NO_3$ and $N_2O_5$. However, the $N_2O_5$ concentrations only showed a slight increase (Figure 4 case) or no apparent change (Figure 5 case), in contrast to the significant increases in $ClNO_2$ and high $p(NO_3)$ values, which suggests rapid heterogeneous loss of $N_2O_5$ and significant $ClNO_2$ production during transport of these plumes from their sources.

The elevated $ClNO_2$ concentrations in the coal-fired plumes here are comparable to previous observation of power plant plumes via tower measurements in Colorado (Riedel et al., 2013) and at a mountain site in southern China (Wang et al., 2016), but the observed $N_2O_5$ within the plumes are significantly lower than those in other coal-fired plumes observed via aircraft, tower, and at mountain sites (Brown et al., 2007; Riedel et al., 2013; Brown et al., 2016). The previous measurement at a surface site in the NCP has observed sustained $ClNO_2$ peaks after sunrise, which was proposed to be the cause of the downward mixing of $ClNO_2$-rich air (estimated values of 1.7-4.0 ppbv) in the residual layer (Tham et al., 2016). In the present study, the frequent

intercepts of coal-fired plumes with elevated $ClNO_2$ concentrations at Mt. Tai, which was typically above the nocturnal boundary layer, affirm this hypothesis and provide direct evidence that significant $ClNO_2$ production occurred in the residual layer from the abundant nocturnal $NO_x$, chloride and background $O_3$ over the NCP. The similar $ClNO_2$-laden air frequently observed at high-elevation sites in northern and southern China suggest ubiquitous $ClNO_2$ in the polluted residual layer and

its importance in the daytime production of ozone in China (Tham et al., 2016; Wang et al., 2016). Moreover, the concurrent nitrate production from heterogeneous $N_2O_5$ reactions (cf. R3) may also contribute to the formation of haze pollution in these regions.

### 3.3 $N_2O_5$ reactivity and heterogeneous uptake coefficient

### 3.3.1 Reactivity of $N_2O_5$ and $NO_3$

The mixing ratios of $N_2O_5$ depend on the nitrate radical production rate and the reactivity of $N_2O_5$ and $NO_3$, including the individual loss rates for $N_2O_5$ or $NO_3$ that contribute to the removal of the pair. $N_2O_5$ reactivity can be assessed using the inverse $N_2O_5$ steady state lifetime, which is the ratio of $p(NO_3)$ to the observed $N_2O_5$ mixing ratios (e.g., Brown et al., 2006; Brown et al., 2009; Brown et al., 2016):

$$\tau(N_2O_5)^{-1} = \frac{p(NO_3)}{[N_2O_5]} \approx \frac{k(NO_3)}{K_{eq}[NO_2]} + k(N_2O_5)_{het} \qquad (3)$$

The steady state inverse lifetime of $N_2O_5$, $\tau(N_2O_5)^{-1}$, is the sum of the $N_2O_5$ loss rate via heterogeneous loss ($k(N_2O_5)_{het}$) and $NO_3$ reactions with VOCs ($k(NO_3)$) with a ratio of $K_{eq}[NO_2]$. $K_{eq}$ is the temperature-dependent $N_2O_5$-$NO_3$ equilibrium coefficient. High $N_2O_5$ reactivity was observed in the present study, with average nighttime $\tau(N_2O_5)^{-1}$ of $1.41 \times 10^{-2}$ s$^{-1}$ before midnight and $1.30 \times 10^{-2}$ s$^{-1}$ after midnight, corresponding to a nighttime $N_2O_5$ lifetime of 1.2-1.3 min. This rapid $N_2O_5$ loss rate is comparable to the results from surface measurements in both urban and rural sites in the NCP (Tham et al., 2016; Wang et

al., 2017b). However, this loss rate is significantly higher than those determined from a mountain site in southern China (Brown et al., 2016) and tower and aircraft measurements in the USA (e.g., Brown et al., 2009; Wagner et al., 2013).

The $NO_3$ reactivity, or loss rate coefficient $k(NO_3)$, can be estimated from the sum of the products of measured VOC concentrations and the bimolecular rate coefficients for the corresponding $NO_3$-VOC reactions (Atkinson and Arey, 2003):

$$k(NO_3) = k_{NO+NO3}[NO] + \sum_i k_i [VOC_i] \qquad (4)$$

Because of the lack of concurrent VOCs measurements in the present study, we used the average VOC speciations measured before sunrise and in the evening at Mt. Tai during our previous study in 2007 (c.f. Table S1) to estimate $k(NO_3)$. The determined nighttime $k(NO_3)$ was $1.33 \times 10^{-2}$ s$^{-1}$ for the first half of the night and $1.07 \times 10^{-2}$ s$^{-1}$ for the period after midnight, which is equivalent to an $NO_3$ lifetime of approximately 1.5 min. The estimated $k(NO_3)$ could be considered as an upper limit for coal-fired plumes because of potential lower biogenic VOC levels within the plumes. The estimation here does not account

for the VOC changes between years and the night to night variability, which may result in uncertainties. The $k(NO_3)$ derived

by another approach, i.e., from the nighttime steady state fits, provides a consistency check and evaluation of the errors, as described below. The heterogeneous loss rate, $k(N_2O_5)_{het}$, can be obtained by subtracting the $k(NO_3)/K_{eq}[NO_2]$ from the determined $\tau(N_2O_5)^{-1}$ in Eq.3. Figure 6a shows the averaged total $N_2O_5$ reactivity and fractions of $N_2O_5$ loss via $NO_3$ ($k(NO_3)/K_{eq}[NO_2]$) and heterogeneous $N_2O_5$ loss during the study period. As shown, the heterogeneous loss was dominant, accounting for 70-80% of total $N_2O_5$ reactivity with higher fraction before midnight. Figure 6b shows the contribution of different VOC categories to the average first-order $NO_3$ loss rate coefficients, $k(NO_3)$. Biogenic monoterpenes accounted for more than half of the $NO_3$ reactivity, followed by anthropogenic alkenes (such as butene), isoprene and dimethyl sulfide (DMS). Aromatics and alkanes made small contributions (<1%) to the total $NO_3$ reactivity. Although some unmeasured organic species (e.g., peroxy radicals) could also contribute to a small fraction of $NO_3$ loss (Brown et al., 2011; Edwards et al., 2017), the dominant $NO_3$ reactivity by biogenic VOCs is similar to that observed at a mountain site in southern China (Brown et al., 2016) and aircraft measurement in residual layer in southeast US (Edwards, et al., 2017), whereas the anthropogenic contribution is much higher in the present study. The estimated $NO_3$ activity is slightly lower than that obtained from surface site measurements in the NCP (Tham et al., 2016; Wang et al., 2017b), which is in line with the higher abundances of VOCs in the polluted boundary layer.

### 3.3.2 N₂O₅ uptake coefficient

Because the $N_2O_5$ uptake coefficient $\gamma$ is related to the first-order loss rate coefficient of $N_2O_5$, $k(N_2O_5)_{het}$ (Eq. (1)), then the Eq. (3) can be expressed as follows:

$$\tau(N_2O_5)^{-1} K_{eq}[NO_2] \approx k(NO_3) + \frac{1}{4}c_{N2O5}S_a K_{eq}[NO_2]\gamma(N_2O_5) \qquad (5)$$

The linear relationship between the left-hand side of Eq. (5) and $1/4c_{N2O5}S_a K_{eq}[NO_2]$ will give the $N_2O_5$ uptake coefficient $\gamma$ as the slope, and the $NO_3$ loss rate coefficient $k(NO_3)$ as the intercept (Brown et al., 2009). We selected data for periods in which $d[N_2O_5]/dt$ is close to zero and the lifetime is relatively stable, which best corresponds to steady-state conditions. Figure 7 shows two examples of $\tau(N_2O_5)^{-1}K_{eq}[NO_2]$ versus $1/4c_{N2O5}S_a K_{eq}[NO_2]$ for cases observed on the nights of August 2 and 21, 2014. The $\gamma$ and $k(NO_3)$ values derived from the linear fits are $\gamma = 0.040$ and $k(NO_3) = 0.025$ s$^{-1}$ for August 2 case and $\gamma = 0.078$ and $k(NO_3) = 0.011$ s$^{-1}$ for August 21 case. Similar analyses were performed for 11 additional cases during the campaign, and the derived results are summarized in Table 2. The determined $\gamma$ values range from 0.021 to 0.102, with a mean value of $0.061 \pm 0.025$. The average $k(NO_3)$ derived from the steady state fits is $0.015 \pm 0.010$ s$^{-1}$, which is comparable to that predicted from the VOC concentrations described above, indicating that the estimated results in the present study are reliable and likely representative of averaged conditions in the region. The agreement between these two methods also corroborates the determination of the uptake coefficient from steady state analysis. The estimated uncertainty in each individual determination varied from 35 to 100%, including statistical errors and uncertainty associated with measurements of gaseous and aerosol species (Tham et al., 2016).

Compared with the previous field-determined $N_2O_5$ uptake coefficients (0.002-0.04) from aircraft, tower, and mountaintop measurements in the USA and southern China (e.g., Brown et al., 2006; Morgan et al., 2015; Brown et al., 2016), the observed γ values in the present study are significantly higher. The large variability of γ at Mt. Tai is similar to that observed at a rural high-elevation site in Germany and a tower measurement in Colorado, with γ ranging from $10^{-3}$ to 0.11 (Wagner et al., 2013; Phillips et al., 2016). The overall higher averaged γ value at Mt Tai is likely associate with the high RH and aerosol composition with high sulfate but low organic fractions, the condition of which favors more efficient $N_2O_5$ uptake (Brown et al., 2006; Wagner et al., 2013; Phillips et al., 2016). A recent laboratory study has reported high γ (> 0.05) of isotope-labeled $N_2O_5$ into aqueous nitrate-containing aerosols and largely enhancement of uptake at higher RH conditions (Gržinić et al., 2016), which help rationalize our field results with larger uptake coefficient than many previous studies. Moreover, a measurement at an urban surface site in Jinan close to Mt. Tai gave similarly high values of γ (0.042-0.092) (Wang et al., 2017b). This may suggest a unique feature of the reactive nitrogen chemistry with rapid heterogeneous $N_2O_5$ loss over this region, and is consistent with the observed low $N_2O_5$ levels but relatively high $ClNO_2$ and particulate nitrate produced from the heterogeneous reactions.

Previous laboratory studies have investigated the dependence of γ on aerosol compositions, and have developed mechanistic parameterizations of γ that can be employed in air quality models (Chang et al., 2011 and references therein). A commonly used parameterization was proposed by Bertram and Thornton (2009) and considered the aerosol volume-to-surface ratio ($V/S$), concentrations of nitrate, chloride, and water. For comparison, γ values were calculated using this parameterization based on the measured aerosol composition and molarity of water determined from the thermodynamic model with inputs of $NH_4^+$, $Na^+$, $SO_4^{2-}$, $NO_3^-$ and $Cl^-$ (E-AIM model IV, http://www.aim.env.uea.ac.uk/aim/model4/model4a.php) (Wexler and Clegg, 2002). An error estimation showed that a 3% change in RH implies an uncertainty in the particle liquid water content of ~5%. In the calculation, mean values of $V/S$ (64.8 - 77.2 nm) measured in the present study instead of empirical pre-factor $A$ were used, and the reaction rate coefficients were employed as the empirical values suggested by Bertram and Thornton (2009).

Figure 8 shows a comparison of the γ values (with total uncertainty) determined from parameterization and measurements. Overall, the parameterized γ shows good correlation ($r = 0.87$) with the observation determined values, and gives an average of 0.063 ± 0.006, which is in good agreement with the average of 0.061 ± 0.025 derived from steady state analysis. However, the γ values from BT-parameterization are in the range of 0.052-0.070, with much lower variability than the measurement determined values. Similar results with compatible averaged γ values between measurements and parameterization predictions but higher variability for measurement derived γ have been reported at a mountain measurement in Germany (Phillips et al., 2016). A distinct difference of γ between the steady-state analysis and the parameterization has also been reported by Chang et al., (2016), who suggested that the uncertainty in determining aerosol water content would introduce errors in the parameterization. Bertram and Thornton (2009) suggested that predicted γ values would plateau and be independent of particulate chemical composition at particle water molarity above 15M. In the present study, the particle water molarity in

these cases was consistently above 25 M because of the high RH and frequent cloud cover at the mountain site, which may explain the lower variability of γ values predicted by parameterization.

A moderate negative dependence ($r = 0.54$) of determined γ on aerosol nitrate concentration can be inferred, with lower values of γ associated with higher nitrate content (cf. Figure S2a). This pattern is consistent with the nitrate suppress effect on $N_2O_5$ uptake identified from previous laboratory studies (Mentel et al., 1999), and also similar to the anti-correlation of γ and nitrate from tower measurements in the USA and aircraft measurements over the UK (Wagner et al., 2013; Morgan et al., 2015). The relationship between the γ with the aerosol water to nitrate ratio also exhibits consistent trend with the previous observations and parameterizations (e.g., Bertram and Thornton, 2009; Morgan et al., 2015), with increasing uptake as the ratio increases (Figure S2b).

Furthermore, as suggested by Bertram and Thornton (2009), the presence of chloride can offset the suppression of $N_2O_5$ uptake by nitrate. The determined γ in the present study also show positive dependence on aerosol chloride concentration ($r = 0.59$), indicating the enhancement of $N_2O_5$ uptake by increased chloride contents in aerosols. This can be better described by the clear positive dependence ($r = 0.84$) of γ on the molar ratio of particulate chloride to nitrate, as illustrated by the color-coded data in Figure 8 and Figure S3b. The variation in γ values determined in the present study appears to be controlled largely by the particulate chloride-to-nitrate ratio, broadly following the competing effects of nitrate and chloride in the parameterization (Bertram and Thornton, 2009; Ryder et al., 2014). However, the discrepancy between the measurement- and parameterization-derived values may imply that some mechanisms and factors affecting γ under conditions of high humid and pollution (e.g., reacto-diffusive length, salting effects, etc.) (Gaston and Thornton, 2016; Gržinić et al., 2016) should be further explicitly considered in the parameterization. The in situ $\gamma_{N_2O_5}$ measurement technique developed by Bertram et al. (2009a) may be useful in directly investigating the complex dependence of γ on different factors in a range of environments.

### 3.4 ClNO₂ production yield

To characterize the formation of $ClNO_2$ from rapid heterogeneous $N_2O_5$ uptake and sufficient particulate chloride, the yields of $ClNO_2$ ($\phi$) were examined for different plumes. For regional diffuse pollution cases, the $\phi$ defined in R3 can be estimated from the ratio between $ClNO_2$ production rate and $N_2O_5$ loss rate, as the first term in below equation.

$$\phi = \frac{d\text{ClNO}_2/dt}{k(\text{N}_2\text{O}_5)_{het}[\text{N}_2\text{O}_5]} = \frac{[\text{ClNO}_2]}{\int k(\text{N}_2\text{O}_5)_{het}[\text{N}_2\text{O}_5]\,dt} \qquad (6)$$

$k(\text{N}_2\text{O}_5)$ values can be determined using the inverse steady-state lifetime analysis described above in Eq. 3, and the production rate of $ClNO_2$ can be derived from the near-linear increase in $ClNO_2$ mixing ratio observed during a period, when the related species (e.g., $NO_x$, $SO_2$) and environmental variables (e.g., temperature, RH) were roughly constant. The approach here assumes that the relevant properties of the nocturnal air mass are conserved, and neglects other possible sources and sinks of $ClNO_2$ in the air mass history. For the intercepted coal-fired plumes exhibiting sharp $ClNO_2$ peaks, the $ClNO_2$ yield can be estimated from the ratio of the observed $ClNO_2$ mixing ratio to the integrated $N_2O_5$ uptake loss over the plume age (i.e., the

second term in Eq. 6). The analysis assumes that no $ClNO_2$ was present at the point of plume emission from the combustion sources and no $ClNO_2$ formation before sunset, and that the $\gamma$ and $\phi$ within the plumes did not change during the transport from the source to the measurement site. The potential variability in these quantities likely bias the estimates, but these assumptions are a necessary simplification to represent the averaged values that best describe the observations. It should be noted that the

steady-state $N_2O_5$ loss rate is crucial in the yield estimation, which could be underestimated by potentially overestimating the loss rate in some cases with large uncertainties in $N_2O_5$ measurement and $NO_3$ reactivity analysis. Therefore, an alternative approach suggested by Riedel et al. (2013) was also applied to derive the $ClNO_2$ yield from the ratio of enhancements of $ClNO_2$ and total nitrate (aerosol $NO_3^-$ + $HNO_3$) in the cases. Given the low time resolution of nitrate data that could potentially introduce large uncertainties, this approach will only be used as a reference to validate the former analysis based on Eq. 6.

Two examples of the yield analysis are shown in Figure 9, which indicate the time periods in which $ClNO_2$ concentration increased while other parameters (such as $N_2O_5$, $NO_x$, $O_3$, and $SO_2$ concentrations) were relatively stable. The $\phi$ values obtained for these two cases were 0.26 and 0.05 for July 27 and August 6, respectively. Similar analyses were performed for all of other selected cases in which the $ClNO_2$ concentration increased and other relevant parameters were relatively constant for a short period, typically 2-3 h, and the obtained results were summarized in Table 2. The determined $\phi$ for the seven coal-fired plumes

are also listed in Table 1. During the measurement period, $\phi$ varied from 0.02 to 0.90, with an average of 0.28 ± 0.24 and a median of 0.22. In comparison, the $\phi$ derived from the production ratio approach showed comparable results with an average of 0.25 ± 0.17, and the $\phi$ values from two different approaches match reasonably well with a Reduced Major Axis Regression (RMA) slope of 0.78 ± 0.08 and $r^2$ of 0.73 (cf Figure S4), which corroborates the yield analysis and indicates that the differences are within the overall uncertainty of 40%. The large variability of $\phi$ is similar to field-derived values in most previous studies,

and the mean value is comparable to that in the nocturnal residual layer over continental Colorado (0.18) (Thornton et al., 2010), but lower than that observed at a mountain site in Germany (0.49) (Phillips et al., 2016). The $\phi$ values for the coal-fired plumes (range of 0.20-0.90; average: 0.46 ± 0.24) are generally higher than the campaign average and those from regional diffuse pollution cases. The maximum $\phi$ (0.90) corresponds to the plume with the highest $ClNO_2$ mixing ratio observed during the campaign. This is consistent with a tower measurement in Colorado, in which higher $ClNO_2$ yields were also observed in

inland power plant plumes (Riedel et al., 2013). Similar to that developed for $\gamma$, a parameterization of $ClNO_2$ yield as a function of aerosol water and chlorine composition has been proposed based on laboratory studies (Bertram and Thornton, 2009; Roberts et al., 2009):

$$\phi = \frac{[Cl^-]}{k'[H_2O]+[Cl^-]} \qquad (7)$$

We compared the field-derived values to the parameterization for cases with available aerosol compositions, using an empirical

$k'$ of 1/450, as recommended by Roberts et al. (2009). The particle liquid water content [$H_2O$] was calculated from the thermodynamic model (E-AIM model IV) based on measured aerosols composition, as described above. As shown in Figure 10a, the $\phi$ values predicted by the parameterization are generally higher than those determined from observed $ClNO_2$

production rates, especially at low measurement-determined yields. For measured $\phi$ values higher than 0.4, smaller differences (<20%) were observed between the two methods, which are within the aggregate uncertainty associated with measurement and derivation. The parameterized $\phi$ values exhibit positive dependence on the aerosol chloride concentration and the $Cl^-/H_2O$ ratio, as shown by the color code in Fig 10a. The measurement-determined values only exhibit measurable such dependence at low yields, implying the possible biased relationship due to higher aerosol water conditions in the present work. The discrepancy between the parameterization $\phi$ based upon aerosol composition and those derived from measured $ClNO_2$ concentrations has been found previously (e.g., Wagner et al., 2013), and the underlying causes have not been resolved.

By examining the relationships between the determined yield and other parameters, we found a slightly negative relationship between $\phi$ and particulate nitrate concentration, as depicted in Figure 10b. Although the data are scattered, the high-yield cases are mostly associated with lower nitrate concentrations, while the $\phi$ for the high nitrate cases (>15 µg m$^{-3}$) are smaller. A similar trend was observed for the $NO_x/NO_y$ ratio, which indicates the 'age' of the air masses, suggesting that higher $\phi$ are usually associated with relatively 'young' air masses exhibiting low nitrate concentrations. More secondary and dissolved organic matters in aged aerosols could be a possible factor contributing to the reduction of $ClNO_2$ production efficiency (Mielke et al., 2013; Ryder et al., 2015; Phillips et al., 2016). Further studies are needed to characterize the combined effects of various parameters on $ClNO_2$ yields, in particular the influences of the aerosol mixing state, chloride availability distribution among particle sizes, organic matter, acidity, other possible loss ways of $ClNO_2$, and potential factors affecting in high humid and polluted conditions  (Laskin et al., 2012; Mielke et al., 2013; Wagner et al., 2013; Ryder et al., 2015; Li et al., 2016; Phillips et al., 2016).

**3.5 Effects of heterogeneous $N_2O_5$ reactions on nitrate formation and $NO_x$ processing**

In addition to abundant $ClNO_2$ formation, rapid heterogeneous $N_2O_5$ uptake may also lead to the production of a large amount of nitrate, which is one of the main components of fine particles contributing to haze pollution in northern China (e.g., Huang et al., 2014). Based on the reactions described above, the formation rate of soluble nitrate from $N_2O_5$ reactions, $p(NO_3^-)$, can be determined from the $ClNO_2$ yield and $N_2O_5$ heterogeneous loss rate as follows:

$$p(NO_3^-) = (2 - \phi)k_{N_2O_5}[N_2O_5] \qquad (8)$$

The $p(NO_3^-)$ values obtained for the select cases during the study period ranged from 0.02 to 0.62 ppt s$^{-1}$, with a mean value of $0.29 \pm 0.18$ ppt s$^{-1}$, corresponding to 0.2- 4.8 µg m$^{-3}$ h$^{-1}$ and $2.2 \pm 1.4$ µg m$^{-3}$ h$^{-1}$ (Table 2). The derived rates are comparable to the observed increases in nitrate concentrations (2-5 µg m$^{-3}$ h$^{-1}$) during haze episodes in summer nights at a rural site in the NCP (Wen et al., 2015). By assuming that produced nitrate is conserved and neglecting the deposition and volatilization loss (e.g., via ammonium nitrate), the in-situ $NO_3^-$ formation could be predicted by integrating each derived formation rate over the corresponding analysis period. Similar to $N_2O_5$ uptake coefficient and $ClNO_2$ yield determination above, the nitrate formation estimation here assumes a conserved air mass with a constant formation rate over the study period. For coal-fired plumes, we equated the measured nitrate concentrations with the increases by assuming that no aerosol nitrate was directly emitted from

the nocturnal point sources. As shown in Figure 11, the predicted nitrate formation shows reasonable agreement with the measured increases in nitrate concentrations ($\Delta NO_3^-$) (RMA slope of 1.14 and $r = 0.81$). This consistency also can serve as a check to validate the reliability of above determined heterogeneous $N_2O_5$ reactivity and parameters of $\gamma$ and $\phi$. The in-situ nitrate formation from heterogeneous $N_2O_5$ reactions was predicted to be as high as 17 $\mu g\ m^{-3}$, with a mean value of $4.3 \pm 4.5$ $\mu g\ m^{-3}$, accounting for $32\ (\pm\ 27)$ % of the observed average nitrate concentration during the cases. This is consistent with the maximum nitrate increase of 14.9 $\mu g\ m^{-3}$ over south China (Li et al., 2016) and 21% nitrate increase in polluted episodes in Beijing (Su et al., 2017) after considering the heterogeneous $N_2O_5$ uptake in the regional model simulation. As for a plume undergo continuous chemical processing from dusk to sunrise, the heterogeneous $N_2O_5$ reactions would lead to substantial nitrate formation (e.g., 22 $\mu g\ m^{-3}$ production for a 10-h night), and could contribute significantly to secondary fine aerosols as the main driver of the persistent haze pollution in northern China.

The formation of nitrate (including $HNO_3$) and its subsequent removal by deposition is the predominant removal mechanism of nitrogen oxides from the atmosphere (Chang et al., 2011). The nocturnal $NO_x$ removal rate depends on the $NO_3$ radical production rate, heterogeneous $N_2O_5$ loss rate, $NO_3$ reaction rate with VOCs, the partitioning between $N_2O_5$ and $NO_3$ concentrations, and $ClNO_2$ yield. $ClNO_2$ mainly functions as a reservoir of $NO_x$, rather than as a sink, because the formation of $ClNO_2$ throughout the night with subsequent morning photolysis recycles $NO_2$ (Behnke et al., 1997). The reactions of $NO_3$ with VOCs would predominantly produce organic nitrate products (Brown and Stutz, 2012 and references therein), but some fraction of $NO_2$ can be regenerated in the $NO_3$ reactions (i.e., with terpenes) (e.g., Wängberg et al., 1997) or released from the decomposition of organic nitrate during the transport (e.g., Francisco and Krylowski, 2005). For simplicity, we neglect the recycling of $NO_2$ from $NO_3$-VOC reactions by assuming the complete removal of reactive nitrogen (Wagner et al., 2013). This would overestimate the $NO_x$ loss since the monoterpenes contribute to around half of $NO_3$ reactivity in the present study, but this assumption does not significantly affect the conclusion because the $NO_3$ loss with VOCs was the minor path comparing to $N_2O_5$ heterogeneous loss. Thus, the nocturnal $NO_x$ loss rate can be quantified by the following equation:

$$L(NO_x) = (2 - \phi)k_{N_2O_5}[N_2O_5] + k_{NO_3}[NO_3] = (1 - \phi)k_{N_2O_5}[N_2O_5] + p[NO_3] \qquad (9)$$

Using the coefficients described above, we calculated the nocturnal loss rate of $NO_x$ for each case, as summarized in Table 2. The $NO_x$ removal rate varied from 0.19 to 2.34 ppb $h^{-1}$, with a mean of $1.12 \pm 0.63$ ppb $h^{-1}$, which corresponds to a pseudo-first order loss rate coefficient of $0.24 \pm 0.08\ h^{-1}$ in average for the studied cases. This loss rate is higher than that determined from a mountain site measurement in Taunus, Germany (~0.2 ppb $h^{-1}$ with typical $NO_2$ level of 1-2 ppb) (Crowley et al., 2010), and the results from aircraft measurements in US over Ohio and Pennsylvania and downwind region of New York (90% and 50% $NO_x$ loss in a 10-hour night, respectively) (Brown et al., 2006). For reference, this nocturnal average loss rate is approximately equivalent to $NO_2$ loss via reaction with OH at afternoon condition assuming OH concentration around $2\times 10^6$ molecules $cm^{-3}$, indicating the importance of nocturnal heterogeneous reactions on $NO_x$ processing and budget. Figure 12 shows the relationship between determined $NO_x$ loss rate and observed ambient $NO_x$ concentration at the measurement site. $NO_x$ loss rate appears to be strongly dependent upon $NO_x$ concentrations below 6 pptv (slope $= 0.32\ h^{-1}$; $r = 0.93$); the loss rate

became more scattered at higher $NO_x$ conditions, which were typically observed in the coal-fired plumes. This result implies that for low $NO_x$ condition (<6 ppbv), 96% of $NO_x$ would be removed after 3-h of nocturnal processing, if no additional $NO_x$ emissions affect the plume during this period.

Comparing $NO_x$ loss to the nitrate formation rates, it can be inferred that the nitrate formation from heterogeneous $N_2O_5$ uptake
is predominant in reactive $NO_x$ loss and account for an average of 87% of the $NO_x$ loss, although this fraction of individual cases varied between 35 to 100%. A box model simulation based on tower measurements at Colorado also reported that the largest proportion of the nitrate radical chemistry is $N_2O_5$ hydrolysis, which typically accounted for 80% of nitrate radical production, whereas the losses to $NO_3$-VOC reactions are less than 10% (Wagner et al., 2013). A recent model simulation for southern China also suggested that considering the $N_2O_5$ uptake and subsequent Cl activation could decrease regional $NO_x$ by
more than 16% (Li et al., 2016). The results obtained in the present study demonstrate the significance of fast heterogeneous $N_2O_5$ chemistry on nocturnal $NO_x$ removal and fine nitrate formation in the polluted residual layer over the NCP.

## 4. Summary and Conclusions

An intensive field study was conducted at a high-altitude site to characterize the reactive nitrogen chemistry in the polluted nocturnal residual layer over the NCP. The results revealed the frequently elevated $ClNO_2$ mixing ratios (maximum: 2065 pptv)
and efficient $ClNO_2$ yields (0.46 ± 0.24) resulting from coal-fired plumes in the residual layer. The presence of $ClNO_2$-laden air in the nocturnal residual layer confirms our previous hypothesis based on a measurement in a rural site in the NCP, that the downward mixing of $ClNO_2$-rich air to the surface in the next morning would have large impacts on early morning photochemistry and ozone production. Rapid heterogeneous $N_2O_5$ uptake and efficient $ClNO_2$ and nitrate formation were observed during the study period. The $\gamma$ determined in the present study (average: 0.061 ± 0.025) exhibited a clear dependence
on the particulate chloride-to-nitrate ratio, and are higher than those observed in other locations, but consistent with those obtained at a surface site in the same region of the NCP. Laboratory-derived parameterizations predicted comparable mean $\gamma$ values, but did not represent the high variability of the measured values, and tended to overestimate $\phi$ in the low yields. These discrepancies suggest that various aerosol physicochemical parameters have complicated effects on $N_2O_5$ uptake and $ClNO_2$ yield, in particular in high humid and polluted residual layer, which requires further investigation.

Fast heterogeneous $N_2O_5$ uptake dominated and accounted for a mean of 87% of the regional nocturnal $NO_x$ loss during the study periods in the NCP. The estimated nocturnal loss rate of $NO_x$ is higher than that previously observed in US and Europe, with averaged loss rate and rate coefficient of 1.12 ± 0.63 ppb h$^{-1}$ and 0.24 ± 0.08 h$^{-1}$, respectively. Moreover, heterogeneous reactions contributed to substantial nitrate production up to 17 µg m$^{-3}$, with a mean nocturnal formation rate of 2.2 ± 1.4 µg m$^{-3}$ h$^{-1}$, and in-situ production could account for 32 ± 27% of the observed nitrate concentrations in the studied cases. The results
may help explain the previously observed rapid nighttime growth of fine nitrate aerosols in the NCP, and demonstrate the

importance of heterogeneous $N_2O_5$-$ClNO_2$ chemistry on $NO_x$ and aerosol budgets in the polluted residual layer over the NCP, which underpins the need for further studies regarding their roles in the formation of complex haze pollution in northern China.

**Acknowledgments.**

The authors would like to thank Dr. Fu Xiao for help in providing the location information of power plants in Shandong. HYSPLIT model is made available by the NOAA Air Resources Laboratory. This work was funded by the National Natural Science Foundation of China (91544213, 41505103, 41275123) and the PolyU Project of Strategic Importance (1-ZE13). The authors also acknowledge the support of the Research Institute for Sustainable Urban Development (RISUD).

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

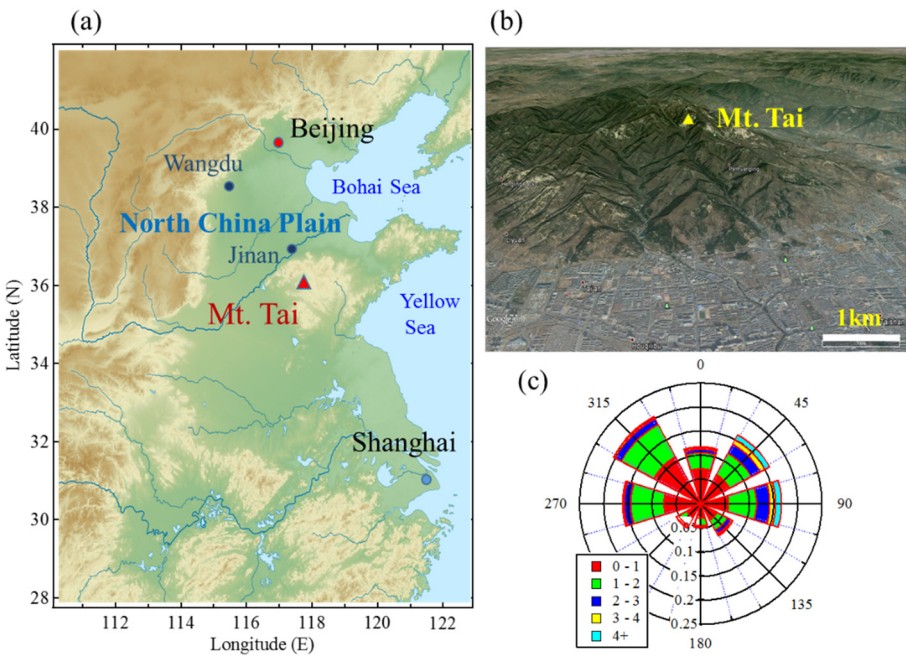

**Figure 1:(a) Map of northern China showing the location of the mountaintop measurement site (Mt. Tai) in the North China Plain,**
5   **(b) expanded topographic view of Mt. Tai and surrounding areas, and (c) a wind rose for the study period of summer 2014.**

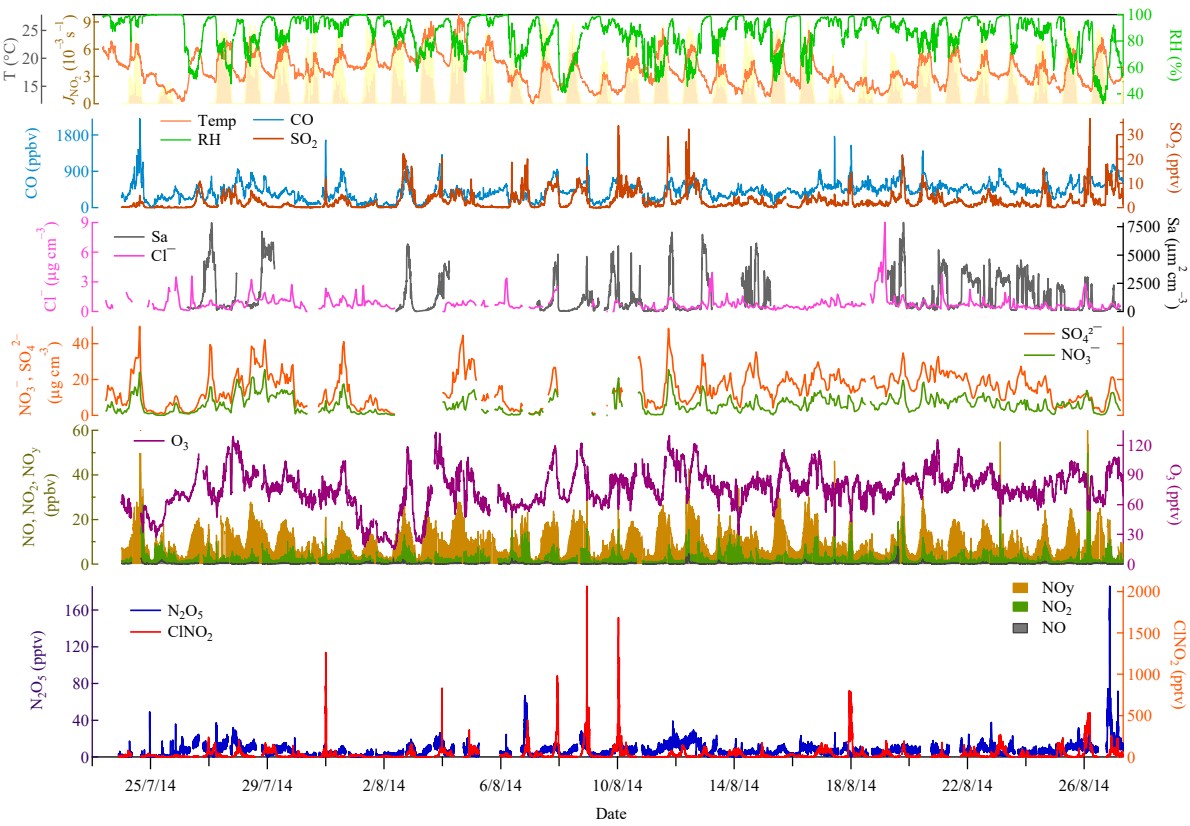

**Figure 2**:**Time series (1-min time resolution) for N₂O₅, ClNO₂, related trace gases, aerosol properties, and meteorological data measured at Mt. Tai from July 24 to August 27, 2014.**

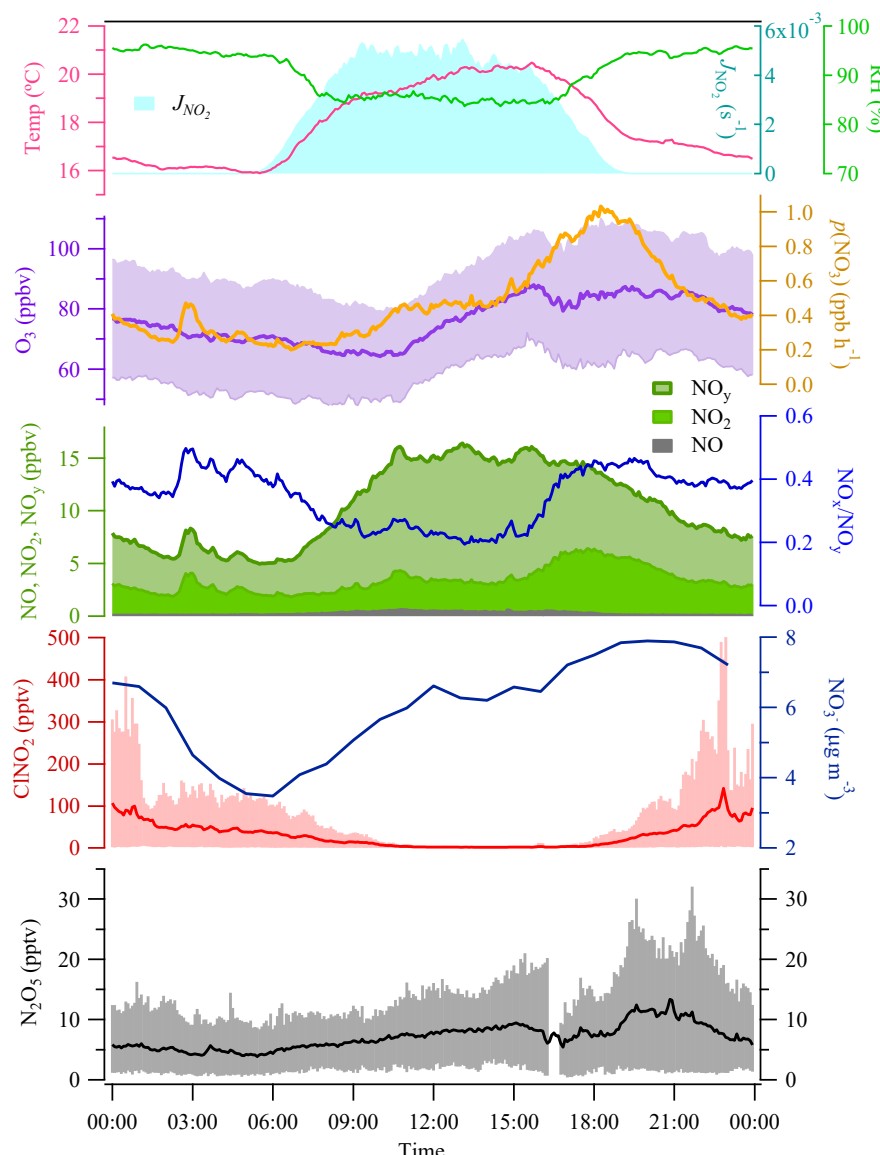

**Figure 3: Diurnal variations of N₂O₅, ClNO₂, NOₓ, NOᵧ, O₃, particulate nitrate, nitrate radical production rate *p*(NO₃) and meteorological parameters during the study period at Mt. Tai. Shaded area in O₃ shows 2σ variation, and vertical bars in N₂O₅ and ClNO₂ represent 10-90ᵗʰ percentile ranges.**

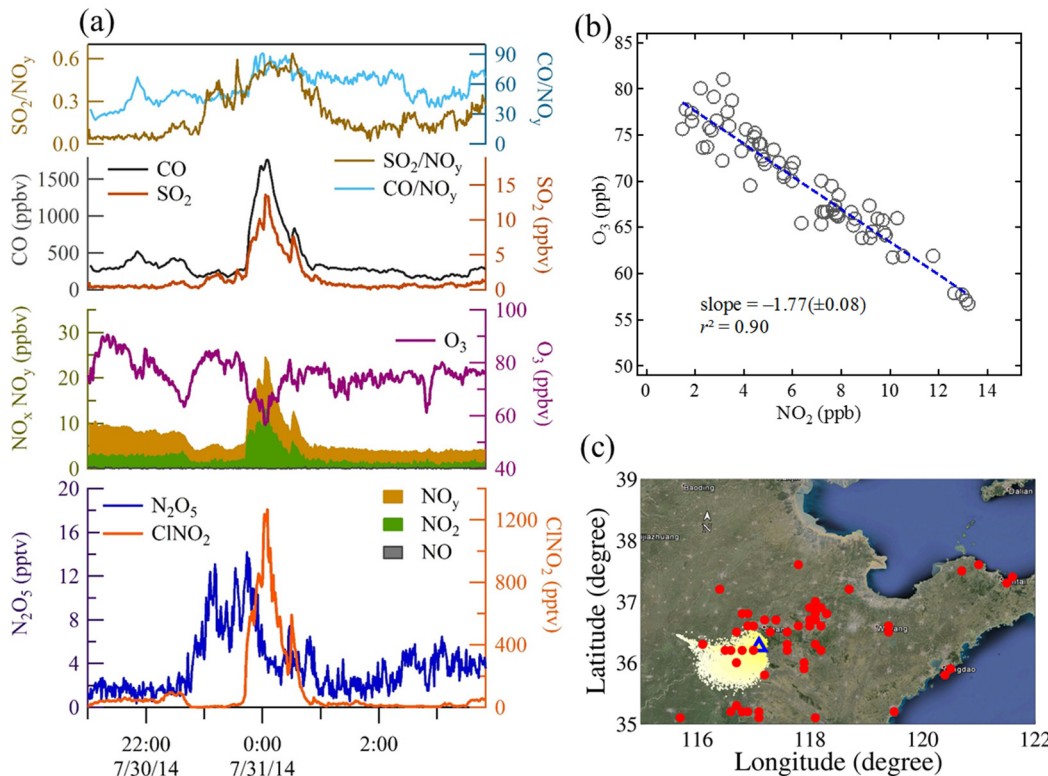

**Figure 4: (a)** Time series for ClNO₂, N₂O₅, and related trace gases observed within the high-ClNO₂ coal-fired plume during the night of July 30-31, 2014. **(b)** Plot of O₃ versus NO₂ concentrations for the plume; plume age was determined from the plot using Eq. (2). **(c)** 12-h HYSPLIT backward particle dispersion image depicting air masses arriving at the measurement site (blue triangle) at the time of the plume, and red dots indicating the location of major coal-fired facilities in cement and steel production and power plants in the region.

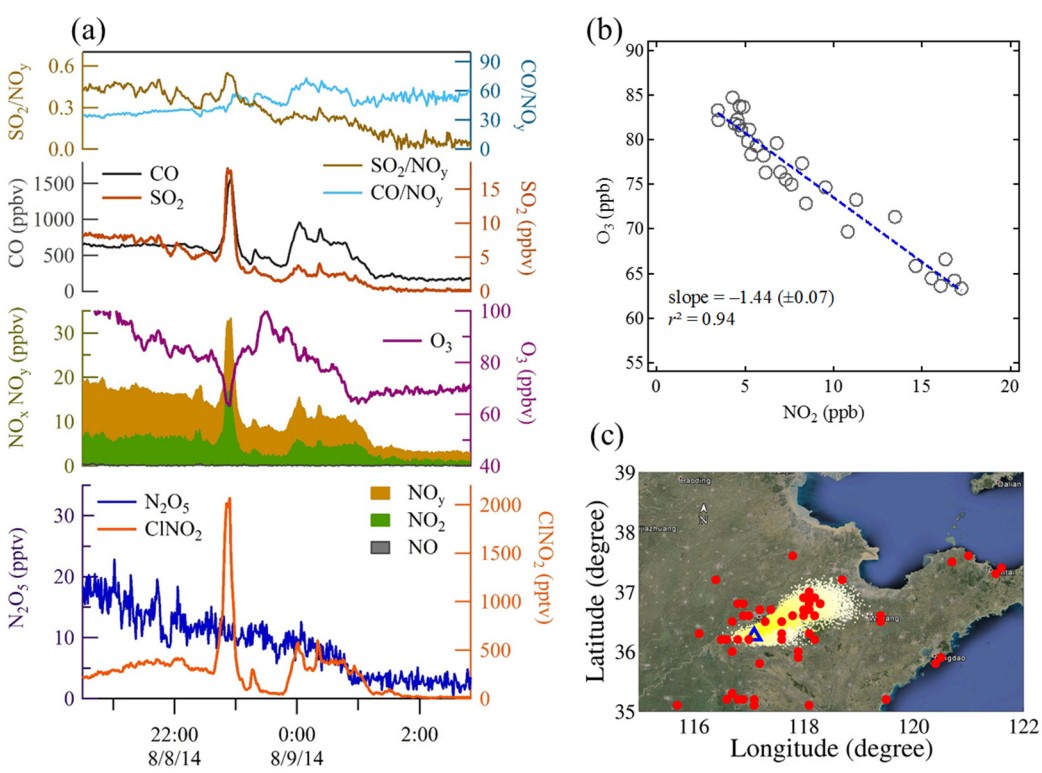

**Figure 5: Same as Figure 4 but for a plume observed during the night of August 8-9, 2014.**

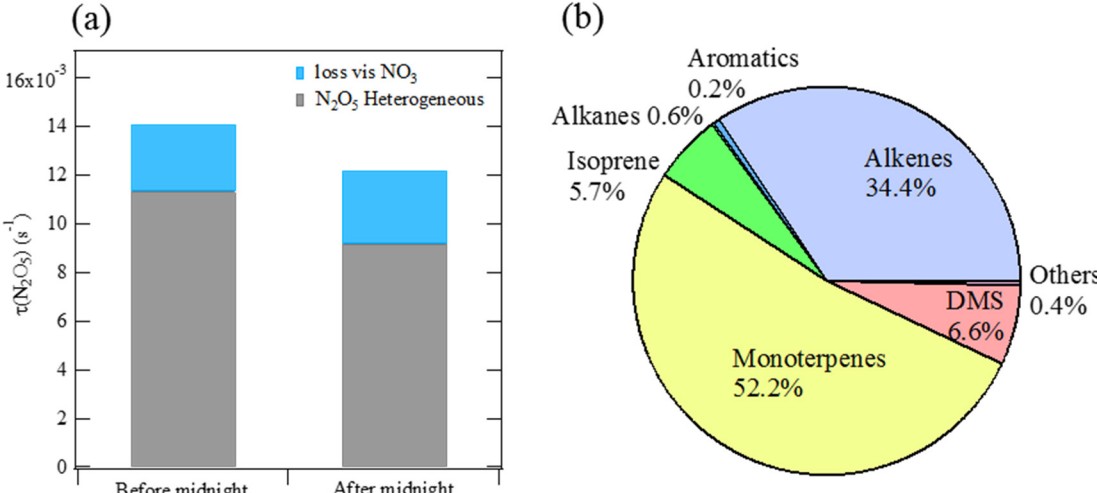

**Figure 6: (a)** Fractions of $N_2O_5$ loss rate coefficients through $NO_3$ loss and the heterogeneous reaction of $N_2O_5$ before (19:00-24:00) and after midnight (1:00-5:00); **(b)** pie chart shoing the average nighttime contributions of different categories of VOCs to $NO_3$ reactivity during the study period.

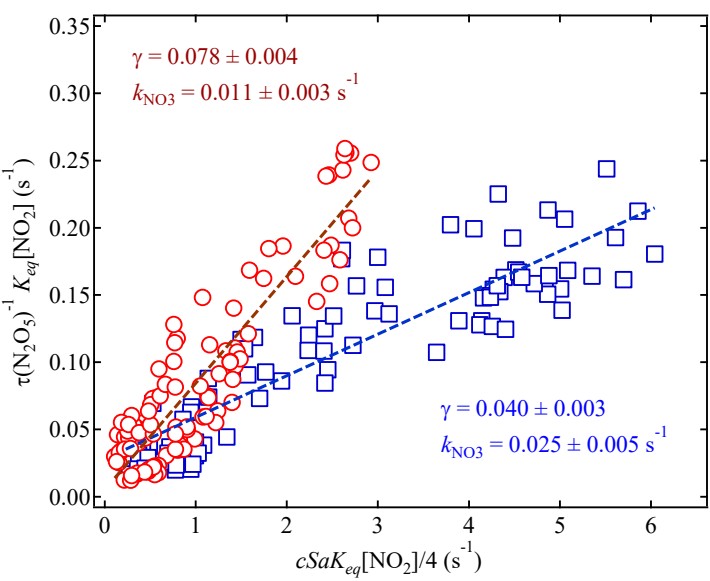

**Figure 7: Example fits of inverse $N_2O_5$ steady-state lifetimes according to Eq. (5) for two cases observed on the nights of August 2 and 21, 2014. The best fit values of $\gamma$ and $k_{NO3}$ are shown.**

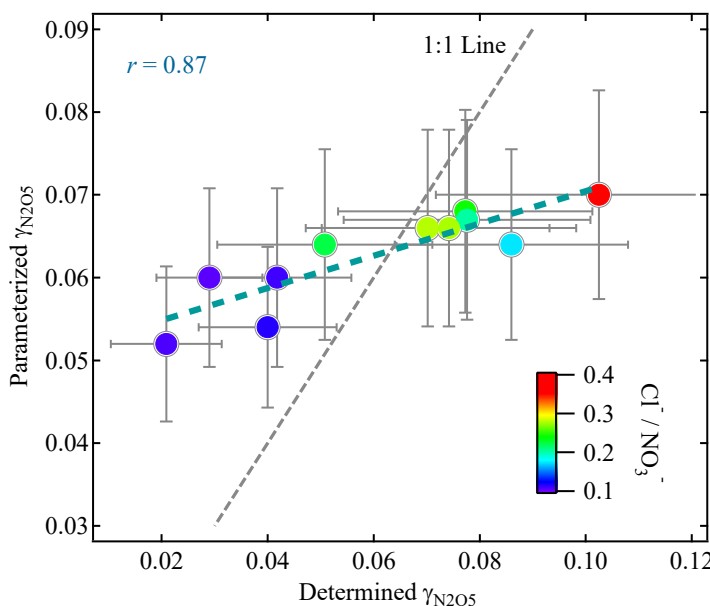

**Figure 8: Comparison of field-determined γ with that derived from the parameterization of Bertram and Thornton, (2009). The colors of the markers indicate the corresponding concentrations ratio of particulate chloride to nitrate. The error bars represent the total aggregate uncertainty associated with measurement and derivation.**

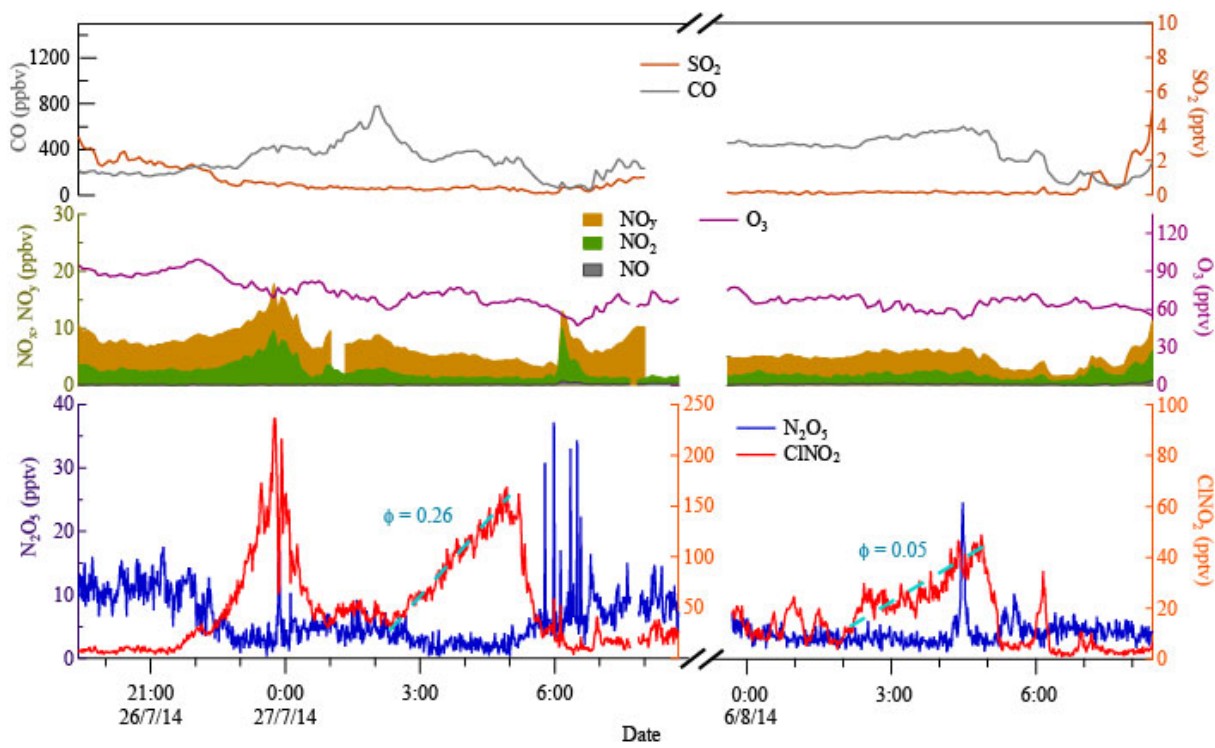

**Figure 9: Examples of ClNO₂ yields determined for two cases on July 27 and August 6, 2014. The ClNO₂ mixing ratios increased steadily, while those of NOₓ, O₃, and SO₂ did not change significantly during the studied periods.**

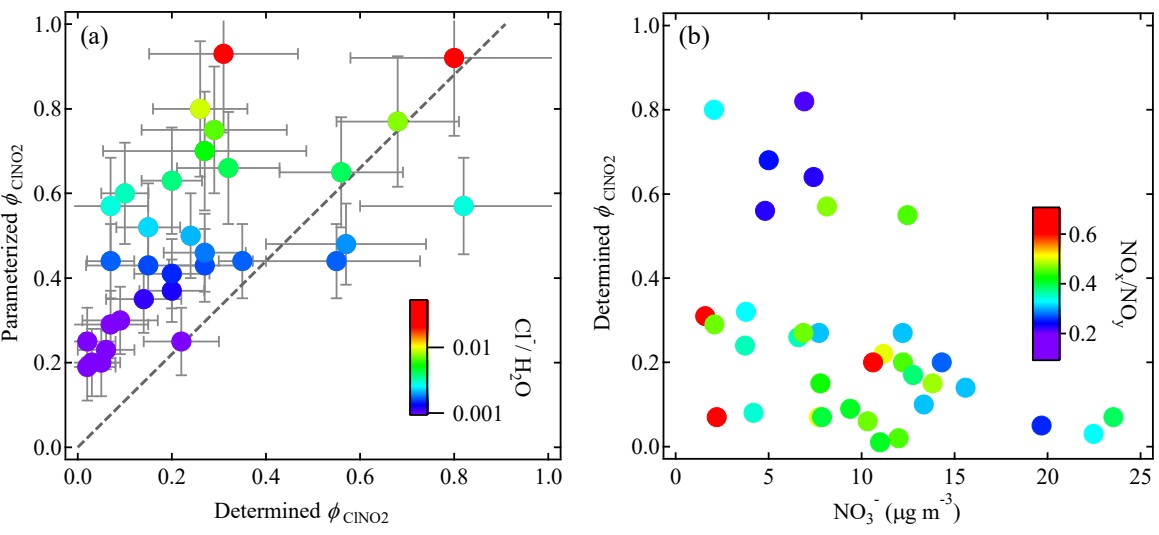

5    **Figure 10: (a) Comparison of field-determined $\phi$ with that derived from parameterization (Eq. 7), and the colors of the markers represent the corresponding Cl⁻/H₂O ratio; (b) relationship between field-determined $\phi$ and measure nitrate concentrations in aerosols, and colors of markers represent the corresponding NOₓ/NOᵧ ratio. The error bars represent the total aggregate uncertainty as similar as Figure 8.**

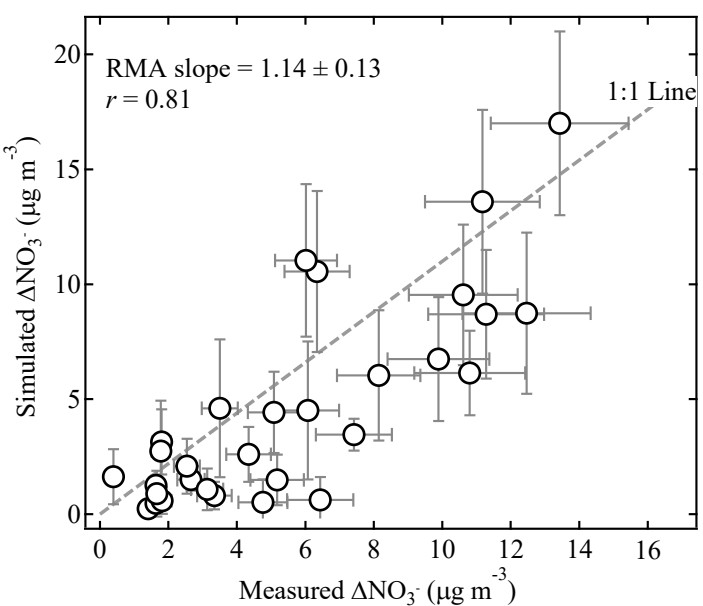

**Figure 11: Comparison of predicted nitrate production based on integrating the derived nitrate formation rate with the measured increase in nitrate concentrations (ΔNO₃⁻) over the analysis time period.**

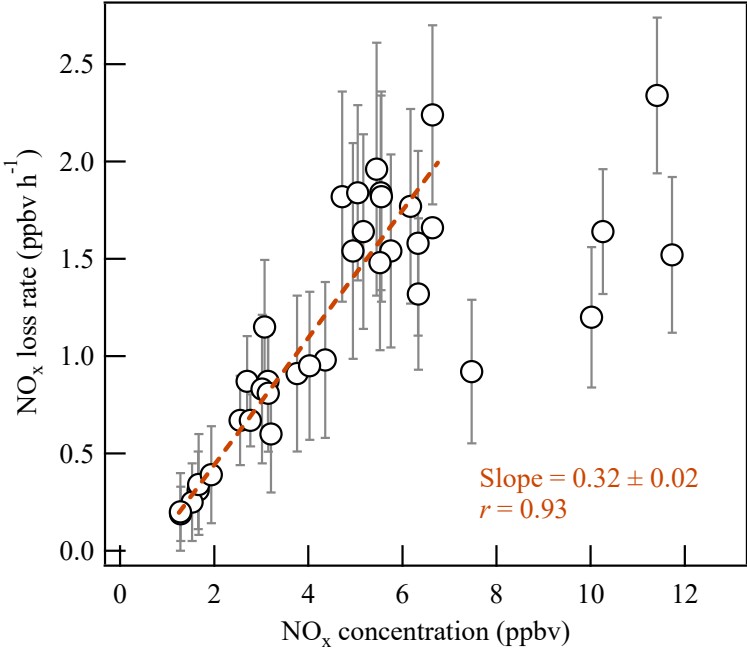

5    **Figure 12: Relationship between determined NOₓ loss rate and observed ambient NOₓ concentration at the measurement site during the study period.**

**Table 1: Chemical characteristics of coal-fired plumes exhibiting high levels of ClNO₂ observed at Mt. Tai during the summer of 2014**

| Date | Duration | $N_2O_5$ (pptv) Mean | $N_2O_5$ (pptv) Maximum | $ClNO_2$ (pptv) Mean | $ClNO_2$ (pptv) Maximum | $O_3$ | $NO_x$ | $NO_x/NO_y$ | $\Delta SO_2/\Delta NO_y$ [a] | $\Delta CO/\Delta NO_y$ [b] | $Cl^-$ (µg cm$^{-3}$) | $t_{plume}$ | $\phi_{ClNO2}$ |
|---|---|---|---|---|---|---|---|---|---|---|---|---|---|
| 30-31 Jul | 23:40-0:45 | 5.9 | 14.2 | 528 | 1265 | 70 | 6.5 | 0.49 | 0.57 | 83 | 2.34 | 3.2 | 0.57 |
| 3-4 Aug | 23:30-0:00 | 20.1 | 23.8 | 506 | 833 | 106 | 2.8 | 0.22 | 2.43 | 108 | NA [c] | 4.9 | 0.64 |
| 7 Aug | 21:30-23:30 | 10.5 | 14.9 | 606 | 976 | 91 | 5.8 | 0.36 | 1.36 | 50 | 2.24 | 5.5 | 0.35 [d] |
| 8 Aug | 22:00-23:10 | 11.0 | 15.1 | 841 | 2065 | 76 | 8.5 | 0.45 | 0.65 | 45 | NA | 2.1 | 0.90 |
| 8-9 Aug | 23:40-01:15 | 6.8 | 12.6 | 315 | 599 | 77 | 4.3 | 0.41 | 0.54 | 85 | NA | 4.4 | 0.23 |
| 10 Aug | 0:00-2:00 | 10.5 | 15.5 | 692 | 1684 | 72 | 6.2 | 0.43 | 1.67 | 50 | 1.10 | 4.6 | 0.55 |
| 17-18 Aug | 22:00-01:30 | 3.5 | 7.7 | 409 | 802 | 60 | 9.5 | 0.55 | 0.48 | 33 | 1.01 | 4.6 | 0.26 [d] |
| 25-26 Aug | 0:00-5:00 | 12.1 | 40.1 | 301 | 534 | 74 | 11.8 | 0.62 | 2.10 | NA | 1.88 | 3.0 | 0.20 |

[a] It represents the slope of $SO_2$ vs $NO_y$ in plumes, and the overall slope for entire campaign was 0.31 with $r^2$ of 0.31.

[b] Same to above note with the campaign overall slope of 15.7 and $r^2$ of 0.23.

[c] Data not available in the case.

[d] For $t_{plumes}$ longer than the nocturnal processing period since sunset, the time since sunset was used in the $ClNO_2$ yield calculation.

**Table 2: Statistical summary of determined $N_2O_5$ uptake coefficients $\gamma$, $ClNO_2$ yields $\phi$, nitrate formation rates and nocturnal $NO_x$ removal rates at Mt. Tai during the study period.**

| | $\gamma_{N2O5}$ | $k_{NO3}$ | $\phi_{ClNO2}$ | $NO_3^-$ formation rate (ppt s$^{-1}$) | $NO_3^-$ formation rate (µg m$^{-3}$ h$^{-1}$) | $NO_x$ removal rate (ppb h$^{-1}$) | $NO_x$ loss rate coefficient (h$^{-1}$) |
|---|---|---|---|---|---|---|---|
| Mean | 0.061 | 0.015 | 0.27 | 0.29 | 2.2 | 1.12 | 0.24 |
| SD | 0.025 | 0.010 | 0.24 | 0.18 | 1.4 | 0.63 | 0.08 |
| Median | 0.070 | 0.011 | 0.20 | 0.26 | 2.0 | 0.98 | 0.24 |
| Min | 0.021 | 0.003 | 0.02 | 0.02 | 0.2 | 0.19 | 0.05 |
| Max | 0.102 | 0.034 | 0.90 | 0.62 | 4.8 | 2.34 | 0.38 |