# Peer review of "Fast heterogeneous N2O5 uptake and ClNO2 production in coal-fired plumes observed in the nocturnal residual layer over the North China Plain"

_Atmospheric Chemistry and Physics, 2017_

## Referee Comment (RC1) · Anonymous Referee #1 · 23 Jun 2017

Wang et al. report measurements of N2O5 and ClNO2 at a mountain site on the North China Plain. The data contain occasional coal power plant plumes in which ClNO2 mixing ratios were enhanced. N2O5 uptake parameters and ClNO2 yields were calculated from a steady state model and found to be fast (i.e., large gamma and high phi) compared to other regions. Calculated uptake parameters were rationalized in terms of aerosol ionic composition (nitrate, chloride, and water calculated using a thermodynamic model). Overall the manuscript reads well and would be of interest to the community. Some of the methodology (such as the calculations of uptake parameters, equations 6 and 8,) require some clarification and may need refinement.

[Figure]

Specific comments

pg 3 line 11 Consider citing Tham, Y., C. Yan, L. Xue, Q. Zha, X. Wang, and T. Wang (2014), Presence of high nitryl chloride in Asian coastal environment and its impact on atmospheric photochemistry, Chin. Sci. Bull., 59(4), 356-359, doi: 10.1007/s11434-013-0063-y.

pg 3, line 12 "such as Colorado, Hessen and Canada" Hessen and Colorado are States of Germany and of the U.S.A. To be consistent, please list the third one as Alberta (a province in Canada).

pg, line 13. The Faxon et al 2015 study was in SE Texas and close to the coast and shouldn't be cited here. Instead consider citing Mielke, L. H., A. Furgeson, C. A. Odame-Ankrah, and H. D. Osthoff (2016), Ubiquity of $ClNO_2$ in the nocturnal boundary layer of Calgary, AB, Canada, Canadian Journal of Chemistry, 94(4), 414-423, doi: 10.1139/cjc-2015-0426.

pg 3 line 14. Following " Anthropogenic sources of chlorine including coal combustion in power plants, industries, and biomass burning may potentially facilitate $ClNO_2$ production" cite Riedel, T. P., et al. (2013), Chlorine activation within urban or power plant plumes: Vertically resolved $ClNO_2$ and $Cl_2$ measurements from a tall tower in a polluted continental setting, J. Geophys. Res., 118(15), 8702-8715, doi: 10.1002/jgrd.50637.

pg 3, line 15 Please state the uncertainty of this number (4.7 ppbv +/- ?)

pg 3, line 26. For readers not familiar with the NCP – please indicate the relative locations of Wangdu and Jinan – perhaps as dots in Figure 1?

pg 3, line 31 "In the present study, we measured the concentrations of $N_2O_5$, $ClNO_2$, and related species at a mountaintop site in the NCP and characterized the nighttime nitrogen chemistry within the residual layer over a polluted region of northern China. " please state month or season and the year here as the season (i.e., temperatures) are

relevant for N2O5 chemistry

pg 4 line 14 " the measurement site is located in an area that is not frequently visited and therefore, should not be significantly affected by local anthropogenic emissions" Since Mt. Tai has been the site of other studies, the data ought to exist to definitively say whether the site IS or IS NOT affected by local emissions.

pg 4, line 17. Along the same lines, since a dozen power plants are mentioned, do the data from the earlier suggest that the site is impact by coal power plant emissions (e.g., high SO2, black carbon, or sulfate levels)?

Is this site sufficiently close or far enough to the coast to be impacted (or not) by sea salt aerosol?

pg 4 line 24 " .... which was replaced daily and washed in the ultrasonic bath to minimize wall loss caused by deposited particles (Wang et al., 2016)." Was there any change in signal levels after the line was washed? What time of day was the tubing changed?

pg 4 line 27 "Manual calibrations of N2O5 and ClNO2 were conducted daily to monitor the instrument sensitivity and background." Please state what the CIMS response factors and associated uncertainties and background levels for each of the ions monitored were.

pg 4, line 28 " The N2O5 standard was synthesized on-line from the reaction between NO2 and O3, and the ClNO2 was produced by passing a known concentration of N2O5 through a NaCl slurry " Please state how the N2O5 and ClNO2 concentrations of this source were quantified or what assumptions were made (e.g., ClNO2 yield from NaCl + N2O5).

pg 5, line 10 " Water soluble ionic compositions of PM2.5... were measured hourly... " It would be great to show a time series of these data. Were chloride concentrations higher in power plant plumes?

pg 5 line 26 and Figure 2. There is a lot of sustained N2O5 during daytime which is unusual especially since temperatures frequently were >20C during the day. Is this a real signal? I am a bit doubtful. In any case, it warrants discussion that N2O5 at 3 pm was in greater abundance than at 3 am (Figure 3).

I am also suspicious about the relatively low levels at night. Was the N2O5 transmission efficiency monitored? If so, please show those data. If not, please state that it was not.

It also looks like there is a hardly any NO during the day which is consistent with there being N2O5, but is strange also as NO2 should photo-dissociate and sustain NO. Perhaps the O3 levels are high enough and photolysis rates low enough for this to happen. In my opinion, this warrants a bit of analysis & discussion – are the levels consistent with a simple photostationary analysis, or is the pss severely perturbed (i.e., too much NO2 relative to NO)?

pg 6 "The elevated ClNO2 levels observed at Mt. Tai are similar to recent measurements at a surface rural site (Wangdu) in northern China (Tham et al., 2016) and a mountain site (Tai Mo Shan) in southern China (Wang et al., 2016), but are slightly higher than previous measurements conducted in coastal (e.g., Osthoff et al., 2008; Riedel et al., 2012; Mielke et al., 2013) and inland sites (e.g., Thornton et al., 2010; Phillips et al., 2012; Riedel et al., 2013) in other regions of the world." This is an interesting comparison though more information is needed to put this particular study site and the above comparisons in context. Is the study site affected by sea salt aerosol from the Pacific Ocean? Based on that information, what levels of ClNO2 would have been expected?

pg 6 line 13 "88.6 ppbv" and "24.4 ppbv" - are these averages? If so, also state the range of values (or standard error).

pg 6, line 16 "16.4 ppbv" same comment as above

pg 6, line 23 – section 3.2 This plume is very interesting. It may be that all the SO2 has

generated enough sulfate to make the aerosol quite acidic. Was Cl2 monitored by any chance?

pg 8, line 20 " Because of the lack of concurrent VOCs measurements in the present study, we used the average VOC speciations measured before sunrise and in the evening at Mt. Tai during our previous study in 2007 ". Would these VOC levels be sustained in the power plant plume, where the P(NO3) is greater than in surrounding air? Consider adding a statement that this estimate is likely an upper limit.

pg 9, lines 19-20. "the observed $\gamma$ values in the present study are significantly higher". This could also be indicatative of a measurement bias such as N2O5 inlet losses (see earlier comment). Considering that the authors claim very large gamma values, attention should be paid to biases introduced by measurement uncertainties. Please add a couple of sentences about this potential issue to the manuscript.

pg 10, line 1. There a multiple versions of the E-AIM model. Please be specific as to what was used (inputs & model).

pg 10, line 21 "nitrate can suppress the N2O5 uptake (Bertram and Thornton, 2009)" The nitrate effect is well documented and goes back to Mentel, T. F., M. Sohn, and A. Wahner (1999), Nitrate effect in the heterogeneous hydrolysis of dinitrogen pentoxide on aqueous aerosols, Phys. Chem. Chem. Phys., 1(24), 5451-5457, doi: 10.1039/a905338g, and not Bertram and Thornton. In general, gamma scales inversely with nitrate as the reaction of NO2+ with nitrate reverses N2O5 uptake and is consistent with Figure S1 shows. The paragraph on lines 17-25 is unnecessarily confusing in that the discussion here neglects the simultaneous effects of other ions (e.g., chloride – discussed in next paragraph). Consider tightening this paragraph.

pg 11, equation 6. The math here is not sound – it should not be dN2O5/dt in the denominator (if N2O5 achieves steady state, as argued earlier, this quantity would be zero). Please remove the first 2 terms.

The assumption here is that ClNO2 is conserved and is not produced by other sources (such as Cl + NO2) – please state these assumptions.

The paragraph that follows equation 6 does not provide enough information. How was the yield of ClNO2 actually calculated? Was the N2O5 uptake loss truly integrated using a time-integrated box-model? Or was it approximated using the ratio of observed mixing ratios (as suggested by the first term in eqn 6)? Note that rates change from the point of emission to the point of observation.

pg 12, line 5 "The parameterized ÏT values exhibit positive dependence on the aerosol chloride concentration and the Cl-/H2O ratio. " I am assuming this is based on the colors shown in Figure 10? How good is the water estimate?

Also, if concentrations of N2O5 are underestimated due to a measurement bias and real concentrations of N2O5 were higher, things fall into place ... (see earlier comments)

pg 12, line 2. Please state that the chloride concentration was measured and the water content is based on a thermodynamical model and perform an error estimate (so that appropriate error bars can be added to Figure 10).

pg 12, eqn 8. This equation is not correct. It ought to be the loss rate of N2O5, not its rate of change (d[N2O5]/dt) which would be zero at steady state. There are major assumptions made here – that aerosol nitrate is conserved, i.e., absence of aerosol deposition and volatilization via NH4NO3 formation etc. These assumptions should be clearly stated.

pg 12, line 31. "The NO3 formation was predicted by integrating each derived formation rate over the corresponding ..." the formation rate of nitrate changes from the point emission to the point of observation, which was not taken into account here.

pg 13, line 16. Wang et al., 2016 is not the best reference. Please cite Behnke, W., C. George, V. Scheer, and C. Zetzsch (1997), Production and decay of ClNO2, from

the reaction of gaseous N2O5 with NaCl solution: Bulk and aerosol experiments, J. Geophys. Res., 102(D3), 3795-3804, doi: 10.1029/96JD03057 instead.

pg 13, lines 16/17 "For simplicity, the reactions of NO3 with VOCs can be assumed to result in the complete removal of reactive nitrogen (Wagner et al., 2013)". This is likely not true in this study, where NO3 primarily reacts with terpenes. Wangberg et al. (1997), Product and mechanistic study of the reaction of NO3 radicals with alpha-pinene, Environm. Sci. Technol., 31(7), 2130-2135, and others since have showed that a significant fraction of NO2 is ultimately released again.

pg 13, line 22. "The NOx removal rate varied from 0.19 to 2.34 ppb h-1, with a mean of 1.12 $\pm$ 0.63 ppb h-1. This loss rate is higher than that determined from tower measurements during wintertime in Colorado, with integrated nocturnal NO2 loss ranging from 2.2 to 4.4 ppbv (Wagner et al., 2013)" Comparing absolute loss rates may not be meaningful since the overall NOx levels may be different – consider normalizing, for example, through division of average nocturnal NOx mixing ratios at both locations to derive a pseudo-first order loss rate coefficient.

pg 14, line 21 " The results demonstrate the significance of heterogeneous N2O5-ClNO2 chemistry in the polluted residual layer over the NCP, ..." Are these rates significant? Perhaps. The question is: significant in what sense or compared to what? Consider adding more context (e.g., daytime nitrate formation rates, rates of other locations, etc.)

pg 19, Figure 1. Please indicate the scales of Fig. 1a (lat/long) and 1b (km/km)

pg 19, Figure 2 Please give a vertical scale for jNO2

pg 21 Figure 4c – is the triangle the measurement location? If so, indicate in the caption.

Figure 4b and 5b – state the fit uncertainties (-1.77+/-?; -1.44+/-?) on the figure

pg 22 Figure 7 - state the fit uncertainties (0.011+/-? and 0.040+/-?) on the figure

pg 22 Figure 8 – show the error bars for the experimental values

pg 24 Figure 10 is there a difference between phi(N2O5) in Figure 10a and phi(ClNO2) in Figure 10b (and Tables 1 and 2)? The main manuscript defines only phi without subscript.

---

## Referee Comment (RC2) · Anonymous Referee #2 · 27 Jun 2017

The paper reports on measurements of N2O5 and ClNO2 on a mountain top site in the North China Plane (NCP), and examines the chemistry of N2O5 to ClNO2 conversion in power plant plumes that were observed during the project. This study is a very useful addition to the growing literature on this important chlorine activation pathway. In general the paper is clear and very well written and should be publishable pending the handling of the following comments and questions.

General Comments I would like to see a better description of the aerosol particle characteristics and chemistry. For example, surface area, organic fraction, in addition to nitrate and chloride could be included in Figure 1. This would be particularly useful

since this is likely the major difference between the environment in this study relative to the studies in Europe or North America. It would also be helpful if instead of mass concentration, some of the correlations in figures (Figure 11, Figure S1) could be also done with molar concentration, which is how the lab studies (Bertram and Thornton, 2009, Roberts et al, 2009) were parameterized.

Specific Comments Abstract, Line 12. I know what you mean when you say effect the next day's photochemistry, but someone not familiar with ClNO2 would first need to know that it photolyzes to yield chlorine atoms, so some additional explanation would be good here. Abstract Line 18. A brief phrase describing how you got the uptake coefficient and yield would be good here. Abstract, Line 22. When you use the word "determined" it sounds like a measurement. It would more accurate to say 'estimated' or 'modeled'. Page 3, Line 2. Not sure what is meant when you say "the field determination of (phi) is limited". Do you mean that there are not very many reported determinations of (phi) from field measurements? Page 3, Line 13. The Thornton et al., 2010 reference should be included in this list. Page 4, Line 19. It would be more proper to say 'iodide ion chemical ionization mass spectrometry with a quadrupole mass spectrometer'. Page 11, Eq. 6. The term dN2O5/dt should really be the loss rate of N2O5, which are corrected shown in the next two terms in the equation. Page 11, Lines 15-20. The big problem with this analysis is that it assumes that the growth rates that are inferred from Figure 9 correspond to the actual kinetic time within the plume. There is no way to know if that is true. The features in Figure 9 could be due to something completely different, e.g. a gradual shift in wind direction so that the plume as gradually influencing the site, starting with the dilute edge. There is simply no way to know what the physical circumstances were, with the evidence at hand. Another approach needs to be found, or the analysis should be abandoned. Page 11-12, Eq. 7 and Lines 1-2. With the equation written as is, k' would then be 1/450, not 450 as stated, to match the parameterization of Roberts et al., 2009. I believe the correct expression was used to generate the points in Figure 10a since Eq. 7 as written would generate phi's that were quite a bit <1. Page 12, Eq. 8. Same problem as Eq. 6,

dN2O5/dt is not the proper term here.

---

## Referee Comment (RC3) · Anonymous Referee #3 · 7 Jul 2017

General comments:

This paper presents measurement of N2O5 and ClNO2 from a polluted mountaintop site in the North China Plain during summer 2014. Measurements of these nighttime reactive nitrogen species in the polluted residual layer of China are novel and a valuable contribution to the literature. The authors attribute several of the plumes encountered at the mountaintop site to emissions from regional coal fired power plants. They further use several standard analysis metrics to interpret the data and provide estimates of N2O5 uptake coefficients and ClNO2 yields, along with the overall influence of nighttime chemistry on aerosol nitrate formation in the region.

Results for N2O5 uptake coefficients are generally larger and ClNO2 yields are generally smaller than previous literature determinations. These observations may not be unrelated. If the analysis method biases the N2O5 uptake coefficient to large values, then the same analysis will tend to predict lower ClNO2 yields. The authors should be careful to consider uncertainties that could lead to such a bias, especially in the aerosol size distribution measurement and in the assessment of NO3-VOC reactivity. Alternatively, the very high relative humidity at this site could lead to exactly the effect that is found here, producing faster N2O5 reactivity but also a larger fraction tending toward HNO3 rather than ClNO2. The paper could make this point explicitly in its comparison to previous work (e.g. Phillips, et al., Wagner et al.).

The paper should be published subject to these comments and the minor comments below.

Specific comments.

Page 4, line 16-17. Is there a database showing the location of major coal fired power plants that could be included with the map in Figure 1? This would help to clarify the number of sources and their distance from the observatory.

Page 4, line 27. What were the results of the manual calibrations for N2O5 and ClNO2? Give some sense for reproducibility, and lack of either N2O5 loss or ClNO2 generation on the inlets.

Page 6, line 5. Elevated CO is not normally associated with coal fired power plant emissions, at least in the U.S and Europe.

Page 6, line 26. What are the slopes of correlation plots of SO2 vs NOy, SO2 vs CO and CO vs NOy? This information is important in the attribution of this plume to a coal fired plant, since coal typically has larger SO2/NOy and lower CO/NOy than that from urban emission. These values could be included in Table 1.

For comparison, what was the overall relationship between CO and NOy or SO2 and

[Figure]

NOy for the entire campaign? If most of the NOy is urban, then the global relationships might define the urban numbers so that the power plants could be more easily distinguished.

Page 7, line 17. Authors probably mean a slope steeper than -1 (rather than +1).

Page 8, line 25. The determination of N2O5 reactivity is not quite clear. Text implies that equation 3 is used, and that k(NO3)/Keq[NO2] is subtracted from this number based on the measured VOCs from a different year. Correct? If so, this should be stated explicitly, possibly with an equation.

If the above is correct, then for the sake of clarity, the NO3 loss rates quoted in line 23 should be divided by Keq[NO2] to make it obvious how the budget was done.

What is not given here is a sense for the uncertainty (e.g., N2O5 contributions of 80% and 71% given to two significant figures with no uncertainty). Since VOC measurements from a separate year are used, and since the NO3 reactivity is dominated by reaction with monoterpenes, which are variable and quite temperature dependent, there could be substantial year to year variability and thus considerable uncertainty in this budget. At the very least, this uncertainty should be qualitatively noted. If the authors have data that would quantify year to year or night to night variability in the NO3 losses, then those numbers should be used to formulate a quantitative error budget.

Is there any potential for unmeasured VOC that contributes more to the NO3 reactivity budget? Have the authors considered reaction of NO3 with peroxy radicals?

Page 9, line 14. Similar comment regarding error analysis in Table 2. The authors should provide error bars for the determined gamma and phi values based on measurement uncertainties. Especially important, but not discussed, is the uncertainty in the aerosol size distribution measurement to determine Sa in equation (5). Such measurements often have considerable uncertainty that can be limiting for the gamma determinations.

Page 10, line 17. What does a plot of gamma vs. NO3-/H2O or Cl-/H2O look like? Especially for the nitrate effect, the dependence against the nitrate to liquid water ratio should give the most information.

Page 11, equation (6). The analysis measured ClNO2 production relative to N2O5 loss. The denominator is difficult to determine with certainty, and especially if the N2O5 loss rates are too large (see concerns about aerosol surface area and NO3 loss to VOC above), the analysis will produce too small a value for phi(ClNO2). These caveats should be noted. There should be production of aerosol nitrate or nitric acid together with the N2O5 loss. Are any trends in aerosol nitrate or NOz (=NOy-NOx) during the periods of ClNO2 increase available to corroborate the analysis? This approach could be more quantitative than one based on N2O5 loss.
* * *

---

## Author Comment (AC1) · 23 Aug 2017

Wang et al. report measurements of N2O5 and ClNO2 at a mountain site on the North China Plain. The data contain occasional coal power plant plumes in which ClNO2 mixing ratios were enhanced. N2O5 uptake parameters and ClNO2 yields were calculated from a steady state model and found to be fast (i.e., large gamma and high phi) compared to other regions. Calculated uptake parameters were rationalized in terms of aerosol ionic composition (nitrate, chloride, and water calculated using a thermodynamic model). Overall the manuscript reads well and would be of interest to the community. Some of the methodology (such as the calculations of uptake parameters, equations 6 and 8,) require some clarification and may need refinement.

We thank the reviewer for the time spent with our manuscript and the constructive suggestions. We have revised the manuscript and clarified the methodology according to the reviewer's comments. Our responses, including changes made to the manuscript, are listed below.

*Reviewer comments are in italics.* Author responses are in plain face. Changes to the text are in blue.

**Specific comments**

1. pg 3 line 11 Consider citing Tham, Y., C. Yan, L. Xue, Q. Zha, X. Wang, and T. Wang (2014), Presence of high nitryl chloride in Asian coastal environment and its impact on atmospheric photochemistry, Chin. Sci. Bull., 59(4), 356-359, doi: 10.1007/s11434-013-0063-y.

pg 3, line 12 "such as Colorado, Hessen and Canada" Hessen and Colorado are States of Germany and of the U.S.A. To be consistent, please list the third one as Alberta (a province in Canada).

pg, line 13. The Faxon et al 2015 study was in SE Texas and close to the coast and shouldn't be cited here. Instead consider citing Mielke, L. H., A. Furgeson, C. A. Odame-Ankrah, and H. D. Osthoff (2016), Ubiquity of ClNO2 in the nocturnal boundary layer of Calgary, AB, Canada, Canadian Journal of Chemistry, 94(4), 414-423, doi: 10.1139/cjc-2015-0426.

pg 3 line 14. Following "Anthropogenic sources of chlorine including coal combustion in power plants, industries, and biomass burning may potentially facilitate ClNO2 production" cite Riedel, T. P., et al. (2013), Chlorine activation within urban or power plant plumes: Vertically resolved ClNO2 and Cl2 measurements from a tall tower in a polluted continental setting, J. Geophys. Res., 118(15), 8702-8715, doi: 10.1002/jgrd.50637.

Response: The suggested references are added or changed according to the reviewer's suggestions.

2. pg 3, line 15 Please state the uncertainty of this number (4.7 ppbv +/-?)

Response: This value was a 1-min averaged maximum concentration, with an uncertainty of  $\pm 0.8$  ppbv. The information is clarified in the revised text.

3. pg 3, line 26. For readers not familiar with the NCP – please indicate the relative locations of Wangdu and Jinan – perhaps as dots in Figure 1?

Response: The locations of the Wangdu and Jinan are indicated in Figure 1, as shown below:

4. pg 3, line 31 "In the present study, we measured the concentrations of N2O5, ClNO2, and related species at a mountaintop site in the NCP and characterized the nighttime nitrogen chemistry within the residual layer over a polluted region of northern China. " please state month or season and the year here as the season (i.e., temperatures) are relevant for N2O5 chemistry.

Response: The information of the season and the year is added.

5. pg 4 line 14 " the measurement site is located in an area that is not frequently visited and therefore, should not be significantly affected by local anthropogenic emissions" Since Mt. Tai has been the site of other studies, the data ought to exist to definitively say whether the site IS or IS NOT affected by local emissions.

pg 4, line 17. Along the same lines, since a dozen power plants are mentioned, do the data from the earlier suggest that the site is impact by coal power plant emissions (e.g., high SO2, black carbon, or sulfate levels)?

Is this site sufficiently close or far enough to the coast to be impacted (or not) by sea salt aerosol?

Response: Yes, as the reviewer points out, previous studies and data had proven the site was not affected by local emissions. Previous studies had also observed occasional combustion plumes with high levels of SO2 (>20ppbv), sulfate (>20  $\mu$ g cm-3) and black carbon (>5  $\mu$ g cm-3) at the summit of Mt. Tai (Zhou et al., 2009; Wang et al., 2011).

The mountain site is about 230 km from the Bohai Sea and the Yellow Sea as illustrated in Figure 1a, and is likely to receive minor influences from sea salt aerosol.

To clarify, we have updated the Figure 1a, and revised the text to include this information, as follows:

"Mt. Tai is 230 km away from the Bohai and Yellow Seas, and the cities of Tai'an and Jinan (the capital of Shandong Province) are located 15 km south and 60 km north of the measurement site, respectively."

"Previous studies at this site indicated that the site is regionally representative without significant local anthropogenic emissions, and affected by the regional aged air masses and occasional combustion plumes from fossil fuel or biomass in the region (e.g., Zhou et al., 2009; Wang et al., 2011, Guo et al., 2012)."

References:

- Guo, J., Wang, Y., Shen, X., Wang, Z., Lee, T., Wang, X., Li, P., Sun, M., Collett Jr, J. L., Wang, W., and Wang, T.: Characterization of cloud water chemistry at Mount Tai, China: Seasonal variation, anthropogenic impact, and cloud processing, *Atmos. Environ.*, 60, 467-476, 2012.
- Wang, Z., Wang, T., Gao, R., Xue, L., Guo, J., Zhou, Y., Nie, W., Wang, X., Xu, P., Gao, J., Zhou, X., Wang, W., and Zhang, Q.: Source and variation of carbonaceous aerosols at Mount Tai, North China: Results from a semicontinuous instrument, Atmospheric Environment, 45, 1655-1667, 2011.
- Zhou, Y., Wang, T., Gao, X. M., Xue, L. K., Wang, X. F., Wang, Z., Gao, J. A., Zhang, Q. Z., and Wang, W. X.: Continuous observations of water-soluble ions in PM2.5 at Mount Tai (1534 ma.s.l.) in central-eastern China, Journal of Atmospheric Chemistry, 64, 107-127, 10.1007/s10874-010-9172-z, 2009.
- 6. pg 4 line 24 " .... which was replaced daily and washed in the ultrasonic bath to minimize wall loss caused by deposited particles (Wang et al., 2016)." Was there any change in signal levels after the line was washed? What time of day was the tubing changed?

pg 4 line 27 "Manual calibrations of N2O5 and ClNO2 were conducted daily to monitor the instrument sensitivity and background." Please state what the CIMS response factors and associated uncertainties and background levels for each of the ions monitored were.

pg 4, line 28 " The N2O5 standard was synthesized on-line from the reaction between NO2 and O3, and the ClNO2 was produced by passing a known concentration of N2O5 through a NaCl slurry " Please state how the N2O5 and ClNO2 concentrations of this source were quantified or what assumptions were made (e.g., ClNO2 yield from NaCl + N2O5).

Response: The detailed calibration, testing and uncertainties of the CIMS measurement have been described in our previous papers (Wang et al., 2016, Tham et al., 2016). The same configuration was used in the present study. For clarity, we have included more detailed information in the revised text, as follows,

"The inlet was installed ~ 1.5 m above the roof of a single-story building, and the sampling line was a 5.5 m PFA-Teflon tubing (1/4 in. o.d.) which was replaced daily in the afternoon before sunset and washed in the ultrasonic bath to minimize wall loss caused by deposited particles (Wang et al., 2016). A small proportion (1.7 SLPM) of total sampling flow (~ 11 SLPM) was diverted to the CIMS system, to reduce the residence time of the air samples in the sampling line. A standard addition of N2O5 into the ambient inlet was performed before and after the tubing replacement to monitor the transmission efficiency, and this practice limited the loss of N2O5 in the inlet to <10% in the 'clean' tubing and about 30% in the next afternoon. Manual calibrations of N2O5 and ClNO2 were conducted daily to determine the instrument sensitivity, and the average of which during the

observation period was  $2.0 \pm 0.6$  for N2O5 and  $2.2 \pm 0.6$  Hz pptv-1 for ClNO2, respectively. The N2O5 standard was synthesized on-line from the reaction between NO2 and O3, and the produced N2O5 were determined from the decrease in NO2 (Wang et al., 2014). This method has been validated with a Cavity Ring Down Spectrometer (CRDS) measurement in previous campaign (Wang et al., 2016). The ClNO2 was produced by passing a known concentration of N2O5 through a NaCl slurry assuming unity conversion efficiency (Roberts et al., 2009) and negligible ClNO2 loss in the system (Wang et al., 2016). The field background was determined by passing the ambient sample through a filter packed with activated carbon, with average levels of 7.8 ± 1.9 and 6.0 ± 1.6 Hz for N2O5 and ClNO2, respectively. The reported concentrations were derived by subtracting the background levels. The detection limit was 4 pptv for both N2O5 and ClNO2 (2 $\sigma$ , 1 min-averaged data), and the uncertainty of the nighttime measurement is estimated to be ± 25% (Tham et al., 2016)."

**References:**

- Roberts, J. M., Osthoff, H. D., Brown, S. S., Ravishankara, A. R., Coffman, D., Quinn, P., and Bates, T.: Laboratory studies of products of N2O5 uptake on Cl- containing substrates, Geophys. Res. Lett., 36, 10.1029/2009gl040448, 2009.
- Tham, Y. J., Wang, Z., Li, Q., Yun, H., Wang, W., Wang, X., Xue, L., Lu, K., Ma, N., Bohn, B., Li, X., Kecorius, S., Größ, J., Shao, M., Wiedensohler, A., Zhang, Y., and Wang, T.: Significant concentrations of nitryl chloride sustained in the morning: Investigations of the causes and impacts on ozone production in a polluted region of northern China, Atmos. Chem. Phys., 16, 14959-14977, 2016.
- Wang, X., Wang, T., Yan, C., Tham, Y. J., Xue, L., Xu, Z., and Zha, Q.: Large daytime signals of N2O5 and NO3 inferred at 62 amu in a TD-CIMS: chemical interference or a real atmospheric phenomenon?, Atmos. Meas. Tech., 7, 1-12, 10.5194/amt-7-1-2014, 2014.
- Wang, T., Tham, Y. J., Xue, L., Li, Q., Zha, Q., Wang, Z., Poon, S. C. N., Dubé, W. P., Blake, D. R., Louie, P. K. K., Luk, C. W. Y., Tsui, W., and Brown, S. S.: Observations of nitryl chloride and modeling its source and effect on ozone in the planetary boundary layer of southern China, J. Geophys. Res. -Atmos., 10.1002/2015jd024556, 10.1002/2015jd024556, 2016.
- 7. pg 5, line 10 "Water soluble ionic compositions of PM2.5... were measured hourly..." It would be great to show a time series of these data. Were chloride concentrations higher in power plant plumes?

Response: The time series of selected ionic species, including sulfate, nitrate and chloride are included in the revised Figure 2. The chloride concentrations in the coal-fired plumes are higher than the campaign average, and this information has been summarized in the revised Table 1. More description about the aerosol composition is also added in the revised text, as follows,

"During the campaign at Mt. Tai, the average concentrations of aerosol sulfate and nitrate were  $14.8 \pm 9.0$  and  $6.0 \pm 4.7 \ \mu g \ m^{-3}$ , accounting for 29.5% and 12.0% of PM2.5 mass, respectively. The aerosol organic-to-sulfate ratio, a parameter that potentially affects the uptake process (Bertram et al., 2009b), was 0.74 on average and much lower than those from studies mentioned above in Europe and US. Moreover, the nighttime averaged Cl- concentration was  $0.89 \pm 0.86 \ \mu g \ cm^{-3}$ , and was an order of magnitude higher than Na+, indicating abundant non-oceanic sources of chloride (e.g., from coal combustion and biomass burning in the NCP) (Tham et al., 2016), which could enhance the production of ClNO2."

| Date      | Duration _  | N 2 O 5 (pptv) |         | ClNO 2 (pptv) |         | 0   | NO   | NO x  | $\Delta SO_2$                  | ΔCO               | Cl⁻ (µg            | t , | ¢ chica           |
|-----------|-------------|--------------------------------------|---------|--------------------------|---------|-----|------|------------------|--------------------------------|-------------------|--------------------|------------|-------------------|
|           |             | Mean                                 | Maximum | Mean                     | Maximum | 03  | INOX | /NO y | /∆NO y a | / $\Delta NO_y^b$ | cm -3 ) | lplume     | $\psi$ CINO2      |
| 30-31 Jul | 23:40-0:45  | 5.9                                  | 14.2    | 528                      | 1265    | 70  | 6.5  | 0.49             | 0.57                           | 83                | 2.34               | 3.2        | 0.57              |
| 3-4 Aug   | 23:30-0:00  | 20.1                                 | 23.8    | 506                      | 833     | 106 | 2.8  | 0.22             | 2.43                           | 108               | NA °               | 4.9        | 0.64              |
| 7 Aug     | 21:30-23:30 | 10.5                                 | 14.9    | 606                      | 976     | 91  | 5.8  | 0.36             | 1.36                           | 50                | 2.24               | 5.5        | 0.35 d |
| 8 Aug     | 22:00-23:10 | 11.0                                 | 15.1    | 841                      | 2065    | 76  | 8.5  | 0.45             | 0.65                           | 45                | NA                 | 2.1        | 0.90              |
| 8-9 Aug   | 23:40-01:15 | 6.8                                  | 12.6    | 315                      | 599     | 77  | 4.3  | 0.41             | 0.54                           | 85                | NA                 | 4.4        | 0.23              |
| 10 Aug    | 0:00-2:00   | 10.5                                 | 15.5    | 692                      | 1684    | 72  | 6.2  | 0.43             | 1.67                           | 50                | 1.10               | 4.6        | 0.55              |
| 17-18 Aug | 22:00-01:30 | 3.5                                  | 7.7     | 409                      | 802     | 60  | 9.5  | 0.55             | 0.48                           | 33                | 1.01               | 4.6        | 0.26 d |
| 25-26 Aug | 0.00-2.00   | 12.1                                 | 40.1    | 301                      | 534     | 74  | 11.8 | 0.62             | 2 10                           | NA                | 1.88               | 3.0        | 0.20              |

Table 1: Chemical characteristics of coal-fired plumes exhibiting high levels of ClNO2 observed at Mt. Tai during the summer of 2014

a It represents the slope of SO2 vs NOy in plumes, and the overall slope for entire campaign was 0.31 with  $r^2$  of 0.31.

b Same to above note with the campaign overall slope of 15.7 and  $r^2$  of 0.23.

° Data not available in the case.

d For *t*plumes longer than the nocturnal processing period since sunset, the time since sunset was used in the ClNO2 yield calculation.

---

## Author Comment (AC2) · 23 Aug 2017

*The paper reports on measurements of N2O5 and ClNO2 on a mountain top site in the North China Plane (NCP), and examines the chemistry of N2O5 to ClNO2 conversion in power plant plumes that were observed during the project. This study is a very useful addition to the growing literature on this important chlorine activation pathway. In general the paper is clear and very well written and should be publishable pending the handling of the following comments and questions.*

We thank the reviewer for the valuable feedback, and we have revised the manuscript according to the comments. Our responses, including changes made to the manuscript, are listed below.

*Reviewer comments are in italics.* Author responses are in plain face. Changes to the text are in blue.

*General Comments*

*1. I would like to see a better description of the aerosol particle characteristics and chemistry. For example, surface area, organic fraction, in addition to nitrate and chloride could be included in Figure 1. This would be particularly useful since this is likely the major difference between the environment in this study relative to the studies in Europe or North America.*

Response: We thank the reviewer for the valuable suggestion. Additional description and discussion of the aerosol characteristic are included in the revised text. The time series of measured aerosol nitrate, sulfate, chloride and surface area was also added in the revised Figure 2, as follows:

"The average nighttime mixing ratios of $O_3$ and $NO_2$ were 77 and 3.0 ppbv, respectively, with an average nitrate radical production rate $p(NO_3)$ of $0.45 \pm 0.40$ ppb $h^{-1}$, which is indicative of potentially active $NO_3$ and $N_2O_5$ chemistry during the study period. However, the low $N_2O_5$ mixing ratios observed during most of the nights suggest a rapid loss of $N_2O_5$, which is consistent with the observed high aerosol surface area (Sa), varied from ~100 to 7800 $\mu m^2$ $cm^{-3}$ with a mean value of 1440 $\mu m^2$ $cm^{-3}$."

"The elevated $ClNO_2$ levels observed at Mt. Tai are similar to recent measurements at a surface rural site (Wangdu) in northern China (Tham et al., 2016) and a mountain site (Tai Mo Shan) in southern China (Wang et al., 2016), but are slightly higher than previous measurements conducted in coastal (e.g., Osthoff et al., 2008; Riedel et al., 2012; Mielke et al., 2013) and inland sites (e.g., Thornton et al., 2010; Phillips et al., 2012; Riedel et al., 2013) in other regions of the world. During the campaign at Mt. Tai, the average concentrations of aerosol sulfate and nitrate were $14.8 \pm 9.0$ and $6.0 \pm 4.7$ $\mu g$ $m^{-3}$, accounting for 29.5% and 12.0% of $PM_{2.5}$ mass, respectively. The aerosol organic-to-sulfate ratio, a parameter that potentially affects the uptake process (Bertram et al., 2009b), was 0.74 on average and much lower than those from studies mentioned above in Europe and US. Moreover, the nighttime averaged $Cl^-$ concentration was $0.89 \pm 0.86$ $\mu g$ $cm^{-3}$, and was an order of magnitude higher than $Na^+$, indicating abundant non-oceanic sources of chloride (e.g., from coal combustion and biomass burning in the NCP) (Tham et al., 2016), which could enhance the production of $ClNO_2$."

[Figure]

**Figure 2 : Time series for N2O5, ClNO2, related trace gases, aerosol properties, and meteorological data measured at Mt. Tai from July 24 to August 27, 2014.**

References:

Tham, Y. J., Wang, Z., Li, Q., Yun, H., Wang, W., Wang, X., Xue, L., Lu, K., Ma, N., Bohn, B., Li, X., Kecorius, S., Größ, J., Shao, M., Wiedensohler, A., Zhang, Y., and Wang, T.: Significant concentrations of nitryl chloride sustained in the morning: Investigations of the causes and impacts on ozone production in a polluted region of northern China, Atmos. Chem. Phys., 16, 14959-14977, 2016.

*2. It would also be helpful if instead of mass concentration, some of the correlations in figures (Figure 11, Figure S1) could be also done with molar concentration, which is how the lab studies (Bertram and Thornton,2009, Roberts et al, 2009) were parameterized.*

Response: We changed the aerosol nitrate and chloride concentrations in the supplement figures to molar concentrations, as shown below. For the Figure 11, it was intended to compare the predicted nitrate concentration with the measured increase of nitrate concentration, so we think the comparison in mass concentration is straightforward and just keep it as before.

[Figure]

**Figure S2: Relationship between derived $\gamma_{N2O5}$ from the measurements with (a) the molar concentration of aerosol nitrate and (b) the molar ratio of aerosol water to nitrate during the study period.**

[Figure]

**Figure S3: Relationship between derived $\gamma_{N2O5}$ from the measurements with (a) the molar concentration of aerosol chloride and (b) the molar ratio of aerosol chloride to nitrate during the study period.**

*Specific Comments*

*3. Abstract, Line 12. I know what you mean when you say effect the next day's photochemistry, but someone not familiar with ClNO2 would first need to know that it photolyzes to yield chlorine atoms, so some additional explanation would be good here.*

Response: We revised the abstract to clarify the effects as follows:

"Dinitrogen pentoxide ($N_2O_5$) and nitryl chloride ($ClNO_2$) are key species in nocturnal tropospheric chemistry, and have significant effects on particulate nitrate formation and the following day's photochemistry through chlorine radical production and $NO_x$ recycling upon photolysis of $ClNO_2$."

*4. Abstract Line 18. A brief phrase describing how you got the uptake coefficient and yield would be good here.*

Response: As the reviewer suggested, we added the methodology information in the abstract to clarify it as follows:

"The heterogeneous $N_2O_5$ uptake coefficient ($\gamma$) and $ClNO_2$ yield ($\phi$) were estimated from steady-state analysis and observed growth rate of $ClNO_2$. The derived $\gamma$ and $\phi$ exhibited high variability, with means of $0.061 \pm 0.025$ and $0.27 \pm 0.24$, respectively."

*5. Abstract, Line 22. When you use the word "determined" it sounds like a measurement. It would more accurate to say 'estimated' or 'modeled'.*

Response: We changed the word of 'determined' to 'estimated'.

*6. Page 3, Line 2. Not sure what is meant when you say "the field determination of (phi) is limited". Do you mean that there are not very many reported determinations of (phi) from field measurements?*

Response: The reviewer's understanding is correct. To clarify, we revised the text as follows:

"There are only a few studies on the determination of $\phi$ from field measurement, and the possible effects of real atmospheric aerosols (including organic composition, mixing state, and chloride partitioning between particle sizes, etc.) have not been well characterized (Mielke et al., 2013; Phillips et al., 2016)."

*7. Page 3, Line 13. The Thornton et al., 2010 reference should be included in this list.*

Response: The suggested reference is added to the list.

*8. Page 4, Line 19. It would be more proper to say 'iodide ion chemical ionization mass spectrometry with a quadrupole mass spectrometer'.*

Response: We corrected it according to the suggestion.

*9. Page 11, Eq. 6. The term dN2O5/dt should really be the loss rate of N2O5, which are corrected shown in the next two terms in the equation.*

Response: We agree with the reviewer's suggestion, and the equation was corrected by removing the first two terms.

*10.     Page 11, Lines 15-20. The big problem with this analysis is that it assumes that the growth rates that are inferred from Figure 9 correspond to the actual kinetic time within the plume. There is no way to know if that is true. The features in Figure 9 could be due to something completely different, e.g. a gradual shift in wind direction so that the plume as gradually influencing the site, starting with the dilute edge. There is simply no way to know what the*

*physical circumstances were, with the evidence at hand. Another approach needs to be found, or the analysis should be abandoned.*

Response: We agree with the reviewer that the growth rate estimation could be biased due to the potential variation of the plume, and simplified estimation would result in some uncertainties. We were aware of these limitations in the estimation, and therefore had carefully inspected the time series to choose the data during a period when related parameters in the air mass were relatively constant. It is likely that the assumptions are reasonable during the short time periods, usually around 2 to 3 hours. In addition, we have applied an alternative approach to derive the $ClNO_2$ yields from the ratio of observed enhancements of $ClNO_2$ and total nitrate (aerosol $NO_3^-$ and $HNO_3$), according to the method suggested by Riedel et al. (2013). The derived $\phi$ values from this approach exhibit reasonable agreement with the original analysis, and most of the differences between two groups of data are within 40% (see figure below). Although either approach requires assumptions and would introduce some uncertainties, the general consistency can serve as a check to corroborate the yield analysis.

To clarify, we have revised the text by elaborating these assumptions in the estimation and the criteria for selecting cases, and also added the comparison results between two different approaches in the revised version, as follows:

"For regional diffuse pollution cases, the $\phi$ defined in R3 can be estimated from the ratio between $ClNO_2$ production rate and $N_2O_5$ loss rate, as the first term in below equation:

$$\phi = \frac{dClNO_2/dt}{k(N_2O_5)_{het}[N_2O_5]} = \frac{[ClNO_2]}{\int k(N_2O_5)_{het}[N_2O_5]\,dt} \qquad (6)$$

$k(N_2O_5)$ values can be determined using the inverse steady-state lifetime analysis described above in Eq. 3, and the production rate of $ClNO_2$ can be derived from the near-linear increase in $ClNO_2$ mixing ratio observed during a period, when the related species (e.g., $NO_x$, $SO_2$) and environmental variables (e.g., temperature, RH) were roughly constant. The approach here assumes that the relevant properties of the nocturnal air mass are conserved, and neglects other possible sources and sinks of $ClNO_2$ in the air mass history. For the intercepted coal-fired plumes exhibiting sharp $ClNO_2$ peaks, the $ClNO_2$ yield can be estimated from the ratio of the observed $ClNO_2$ mixing ratio to the integrated $N_2O_5$ uptake loss over the plume age (i.e., the second term in Eq. 6). The analysis assumes that no $ClNO_2$ was present at the point of plume emission from the combustion sources and no $ClNO_2$ formation before sunset, and that the $\gamma$ and $\phi$ within the plumes did not change during the transport from the source to the measurement site. The potential variability in these quantities likely bias the estimates, but these assumptions are a necessary simplification to represent the averaged values that best describe the observations. It should be noted that the steady-state $N_2O_5$ loss rate is crucial in the yield estimation, which could be underestimated by potentially overestimating the loss rate in some cases with large uncertainties in $N_2O_5$ measurement and $NO_3$ reactivity analysis. Therefore, an alternative approach suggested by Riedel et al. (2013) was also applied to derive the $ClNO_2$ yield from the ratio of enhancements of $ClNO_2$ and total nitrate (aerosol $NO_3^-$ + $HNO_3$) in the cases. Given the low time resolution of nitrate data that could potentially introduce large uncertainties, this approach will only be used as a reference to validate the former analysis based on Eq. 6."

"The determined $\phi$ for the seven coal-fired plumes are also listed in Table 1. During the measurement period, $\phi$ varied from 0.02 to 0.90, with an average of 0.28 ± 0.24 and a median of 0.22. In comparison, the $\phi$ derived from the production ratio approach showed comparable results with an average of 0.25 ± 0.17, and the $\phi$ values from two different approaches match reasonably well with a Reduced Major Axis Regression (RMA) slope of 0.78 ± 0.08 and $r^2$ of 0.73 (cf Figure S4), which corroborates the yield analysis and indicates that the differences are within the overall uncertainty of 40%."

[Figure]

**Figure S4: Comparison of estimated ClNO$_2$ yields from two different approaches: approach A using the ratio of the observed ClNO$_2$ growth rate to steady-state N$_2$O$_5$ loss rate based on Eq. 6; approach B using the production ratio of observed enhancements of ClNO$_2$ and total nitrate, $\phi = 2/(\Delta NO_3^-/\Delta ClNO_2 +1)$ according to Riedel et al., 2013.**

References:

Riedel, T. P., Wagner, N. L., Dubé, W. P., Middlebrook, A. M., Young, C. J., Öztürk, F., Bahreini, R., VandenBoer, T. C., Wolfe, D. E., Williams, E. J., Roberts, J. M., Brown, S. S., and Thornton, J. A.: Chlorine activation within urban or power plant plumes: Vertically resolved ClNO2 and Cl2 measurements from a tall tower in a polluted continental setting, J. Geophys. Res. -Atmos., 118, 8702-8715, 10.1002/jgrd.50637, 2013.

11.    *Page 11-12, Eq.7 and Lines 1-2. With the equation written as is, k' would then be 1/450, not 450 as stated, to match the parameterization of Roberts et al., 2009. I believe the correct expression was used to generate the points in Figure 10a since Eq. 7 as written would generate phi's that were quite a bit <1.*

Response: We corrected the typo in the revised version.

12.    *Page 12, Eq. 8. Same problem as Eq. 6, dN2O5/dt is not the proper term here.*

Response: The equation is corrected by removing the first term.

---

## Author Comment (AC3) · 23 Aug 2017

**General comments:**

This paper presents measurement of N2O5 and ClNO2 from a polluted mountaintop site in the North China Plain during summer 2014. Measurements of these nighttime reactive nitrogen species in the polluted residual layer of China are novel and a valuable contribution to the literature. The authors attribute several of the plumes encountered at the mountaintop site to emissions from regional coal fired power plants. They further use several standard analysis metrics to interpret the data and provide estimates of N2O5 uptake coefficients and ClNO2 yields, along with the overall influence of nighttime chemistry on aerosol nitrate formation in the region. *Results for N2O5 uptake coefficients are generally larger and ClNO2 yields are generally smaller* than previous literature determinations. These observations may not be unrelated. If the analysis method biases the N2O5 uptake coefficient to large values, then the same analysis will tend to predict lower ClNO2 yields. The authors should be careful to consider uncertainties that could lead to such a bias, especially in the aerosol size distribution measurement and in the assessment of NO3-VOC reactivity. Alternatively, the very high relative humidity at this site could lead to exactly the effect that is found here, producing faster N2O5 reactivity but also a larger fraction tending toward HNO3 rather than ClNO2. The paper could make this point explicitly in its comparison to previous work (e.g. Phillips, et al., Wagner et al.). The paper should be published subject to these comments and the minor comments below.

We thank the reviewer for the valuable suggestions. We are aware of the uncertainties in the measurement and analysis that could potentially bias the results. In the revised text, we have included a more detailed description of the measurement uncertainty in the methodology part, including N2O5 transmission efficiency, calibrations and surface area calculation. We have also clarified the assumptions and uncertainties in the NO3 reactivity estimation and determination of uptake coefficient and yield. Additional approach for yield estimation is also applied as a check to corroborate the analysis and results. More discussions on the observed high uptake coefficient and the possible effects of high humidity and aerosol characteristics comparing to previous studies are also added in the revised text.

Our responses to the comments, including changes made to the manuscript, are listed point-bypoint below.

*Reviewer comments are in italics*. Author responses are in plain face. Changes to the text are in blue.

**Specific comments.**

1. Page 4, line 16-17. Is there a database showing the location of major coal fired power plants that could be included with the map in Figure 1? This would help to clarify the number of sources and their distance from the observatory.

Response: The location information of major coal-fired facilities in the industry (cement and steel production) and power plants in the region is included in the maps in Figure 4c and 5c, to aid the discussion in the section of coal-fired plumes (section 3.2). Figure 4c is shown below for reference:

2. Page 4, line 27. What were the results of the manual calibrations for N2O5 and ClNO2? Give some sense for reproducibility, and lack of either N2O5 loss or ClNO2 generation on the inlets.

Response: To clarify, we added a more detailed description of the  $N_2O_5/CINO_2$  measurement, including transmission efficiency, calibrations and the uncertainties in the methodology part. The revised text reads,

"The inlet was installed  $\sim 1.5$  m above the roof of a single-story building, and the sampling line was a 5.5 m PFA-Teflon tubing (1/4 in. o.d.) which was replaced daily in the afternoon before sunset and washed in the ultrasonic bath to minimize wall loss caused by deposited particles (Wang et al., 2016). A small proportion (1.7 SLPM) of total sampling flow (~ 11 SLPM) was diverted to the CIMS system, to reduce the residence time of the air samples in the sampling line. A standard addition of N2O5 into the ambient inlet was performed before and after the tubing replacement to monitor the transmission efficiency, and this practice limited the loss of N2O5 in the inlet to <10% in the 'clean' tubing and about 30% in the next afternoon. Manual calibrations of N2O5 and ClNO2 were conducted daily to determine the instrument sensitivity, and the average of which during the observation period was  $2.0 \pm 0.6$  for N2O5 and  $2.2 \pm 0.6$  Hz pptv-1 for ClNO2, respectively. The N2O5 standard was synthesized on-line from the reaction between NO2 and O3, and the produced N2O5 were determined from the decrease in NO2 (Wang et al., 2014). This method has been validated with a Cavity Ring Down Spectrometer (CRDS) measurement in previous campaign (Wang et al., 2016). The ClNO2 was produced by passing a known concentration of N2O5 through a NaCl slurry assuming unity conversion efficiency (Roberts et al., 2009) and negligible ClNO2 loss in the system (Wang et al., 2016). The field background was determined by passing the ambient sample through a filter packed with activated carbon, with average levels of 7.8  $\pm$  1.9 and 6.0  $\pm$  1.6 Hz for N2O5 and ClNO2, respectively. The reported concentrations were derived by subtracting the background levels. The detection limit was 4 pptv for both N2O5 and ClNO2 (2 $\sigma$ , 1 min-averaged data), and the uncertainty of the nighttime measurement is estimated to be  $\pm 25\%$  (Tham et al., 2016)."

**References:**

Roberts, J. M., Osthoff, H. D., Brown, S. S., Ravishankara, A. R., Coffman, D., Quinn, P., and Bates, T.: Laboratory studies of products of N2O5 uptake on Cl- containing substrates, Geophys. Res. Lett., 36, 10.1029/2009gl040448, 2009.

Tham, Y. J., Wang, Z., Li, Q., Yun, H., Wang, W., Wang, X., Xue, L., Lu, K., Ma, N., Bohn, B., Li, X., Kecorius, S., Größ, J., Shao, M., Wiedensohler, A., Zhang, Y., and Wang, T.: Significant concentrations of nitryl chloride sustained in the morning: Investigations of the causes and impacts on ozone production in a polluted region of northern China, Atmos. Chem. Phys., 16, 14959-14977, 2016.

- Wang, X., Wang, T., Yan, C., Tham, Y. J., Xue, L., Xu, Z., and Zha, Q.: Large daytime signals of N2O5 and NO3 inferred at 62 amu in a TD-CIMS: chemical interference or a real atmospheric phenomenon?, Atmos. Meas. Tech., 7, 1-12, 10.5194/amt-7-1-2014, 2014.
- Wang, T., Tham, Y. J., Xue, L., Li, Q., Zha, Q., Wang, Z., Poon, S. C. N., Dubé, W. P., Blake, D. R., Louie, P. K. K., Luk, C. W. Y., Tsui, W., and Brown, S. S.: Observations of nitryl chloride and modeling its source and effect on ozone in the planetary boundary layer of southern China, J. Geophys. Res. -Atmos., 10.1002/2015jd024556, 10.1002/2015jd024556, 2016.
- 3. Page 6, line 5. Elevated CO is not normally associated with coal fired power plant emissions, at least in the U.S and Europe.

Response: The reviewer makes an excellent point here. In north China, the industry sector, including the cement kilns, iron and steel industry, etc., contributes the largest portion of CO emission, whereas the power plants are the fourth contributor to CO emission because of the better combustion efficiency (Streets et al., 2006; Zhang et al., 2009; Saikawa et al., 2017). For both SO2 and NOx emissions, the power plants and industry are the two largest source sectors. Therefore, from the measurement results, we can certainly attribute the plumes with a steep increase of SO2 to coal combustion origin, and the ratio of CO to SO2 or NOy may give some indications of its source from power plants or industrial plants.

To clarify, we changed the definition of the high  $CINO_2$  plume from 'power plant plumes' to 'coalfired plumes', and revised the manuscript title as "Fast heterogeneous  $N_2O_5$  uptake and  $CINO_2$ production in coal-fired plumes observed in the nocturnal residual layer over the North China Plain". The relevant description in the text was also revised accordingly.

References:

- Saikawa, E., Kim, H., Zhong, M., Avramov, A., Zhao, Y., Janssens-Maenhout, G., Kurokawa, J. I., Klimont, Z., Wagner, F., Naik, V., Horowitz, L. W., and Zhang, Q.: Comparison of emissions inventories of anthropogenic air pollutants and greenhouse gases in China, Atmos. Chem. Phys., 17, 6393-6421, 10.5194/acp-17-6393-2017, 2017.
- Streets, D. G., Zhang, Q., Wang, L., He, K., Hao, J., Wu, Y., Tang, Y., and Carmichael, G. R.: Revisiting China's CO emissions after the Transport and Chemical Evolution over the Pacific (TRACE-P) mission: Synthesis of inventories, atmospheric modeling, and observations, Journal of Geophysical Research: Atmospheres, 111, 10.1029/2006JD007118, 2006.
- Zhang, Q., Streets, D. G., Carmichael, G. R., He, K. B., Huo, H., Kannari, A., Klimont, Z., Park, I. S., Reddy, S., Fu, J. S., Chen, D., Duan, L., Lei, Y., Wang, L. T., and Yao, Z. L.: Asian emissions in 2006 for the NASA INTEX-B mission, Atmos. Chem. Phys., 9, 5131-5153, 10.5194/acp-9-5131-2009, 2009.
- 4. Page 6, line 26. What are the slopes of correlation plots of SO2 vs NOy, SO2 vs CO and CO vs NOy? This information is important in the attribution of this plume to a coal fired plant, since coal typically has larger SO2/NOy and lower CO/NOy than that from urban emission. These values could be included in Table 1.

For comparison, what was the overall relationship between CO and NOy or SO2 and NOy for the entire campaign? If most of the NOy is urban, then the global relationships might define the urban numbers so that the power plants could be more easily distinguished.

Response: We appreciate the reviewer's helpful suggestions. We examined the correlations among  $SO_2$ ,  $NO_y$  and CO in these cases to figure out the source information. The slopes of  $SO_2$  vs  $NO_y$  in these cases ranged from 0.48 to 2.43, and were all higher than the overall slope of 0.31 for the entire campaign, indicating the coal combustion source of these plumes. The slope of CO vs  $NO_y$  varied from 33 to 108, and was also higher than the campaign global slope of 15.7. As we stated

in the previous response, both high  $SO_2/NO_y$  and  $CO/NO_y$  slopes (and ratios) suggest that the plumes were likely originated from coal-fired facilities in industrial plants, whereas high  $SO_2/NO_y$  with relatively lower  $CO/NO_y$  possibly suggest the source from power plants.

In the revised version, we added the derived slopes of SO2 vs NOy and CO vs NOy in Table 1, and also included the ratios of SO2/NOy and CO/NOy in Figure 4a and 5a. The text and discussions are also clarified and revised to match these changes, as follows,

"As described above, several plumes with elevated ClNO2 concentrations (> 500 pptv) were observed during the measurement period. Figure 4a illustrates the high ClNO2 case observed during the night of July 30-31, 2014. The CINO2 concentration peaked sharply at 1265 pptv, which was accompanied by a steep rise in the concentrations of SO2, NOx and CO. The SO2/NOy ratio increased from ~0.1 to 0.6 in the plume center, with a  $\Delta SO_2/\Delta NO_y$  slope of 0.57, indicating the coal combustion source of the plume. The coincident increase in CO/NOv ratio from ~30 to 90 suggests that it was likely originated from coal-fired industry facilities, such as cement and steel production plants, which is the largest emitting sector of CO in north China (Streets et al., 2006; Zhang et al., 2009). The 12-h backward particle dispersion trajectories calculated from the HYSPLIT model revealed that the air masses mostly moved slowly from the west, and passed over the region with cement and steel production industry and power plants before arriving at the measurement site. Figure 5a shows the highest ClNO2 case (2065 ppbv) observed on the night of August 8, 2014. The simultaneous increases in SO2, NOx and CO concentrations, together with the higher SO2/NOy ratio ( $\sim$ 0.5) comparing to that outside of plume ( $\sim$ 0.1) and the campaign average (0.24), again indicate the coal combustion origin of the plume. The relatively lower CO/NOy ratio of ~50 possibly suggests the plume affected by power plant emission, as shown by the derived backward particle dispersion trajectories. Table 1 summarizes the chemical characteristics of the eight cases of high-ClNO2 coal-fired plumes during the study period. In these cases, the average SO2 mixing ratios ranged from 2.3 to 18.7 ppbv, and the maximum ClNO2 and N2O5 mixing ratios ranged from 534 to 2065 ppbv and 7.3 to 40.1 ppbv, respectively, with corresponding ClNO2/N2O5 ratios of 25 to 118. The mixing ratios for O3 and NO2 ranged from 60 to 106 ppbv and 2.8 to 11.8 ppbv, respectively, resulting in high  $p(NO_3)$  values of 0.60 to 1.59 ppbv h-1. The aerosol chloride concentration ranged from 1.01 to 2.34 µg cm-3, which was higher than the nighttime average (0.89  $\mu$ g cm-3) and conducive to ClNO2 production from R3."

| Date      | Duration    | N 2 O 5 (pptv) |         | ClNO 2 (pptv) |         | 02  | NO   | NO x  | $\Delta SO_2$    | ΔCO                | Cl- (µg            | Inluma | \$ CINO2          |
|-----------|-------------|--------------------------------------|---------|--------------------------|---------|-----|------|------------------|------------------|--------------------|--------------------|--------|-------------------|
|           |             | Mean                                 | Maximum | Mean                     | Maximum | 03  | 1101 | /NO y | $/\Delta NO_y^a$ | /∆NOy b | cm -3 ) | epiume | φ cinoz           |
| 30-31 Jul | 23:40-0:45  | 5.9                                  | 14.2    | 528                      | 1265    | 70  | 6.5  | 0.49             | 0.57             | 83                 | 2.34               | 3.2    | 0.57              |
| 3-4 Aug   | 23:30-0:00  | 20.1                                 | 23.8    | 506                      | 833     | 106 | 2.8  | 0.22             | 2.43             | 108                | NA c    | 4.9    | 0.64              |
| 7 Aug     | 21:30-23:30 | 10.5                                 | 14.9    | 606                      | 976     | 91  | 5.8  | 0.36             | 1.36             | 50                 | 2.24               | 5.5    | 0.35 d            |
| 8 Aug     | 22:00-23:10 | 11.0                                 | 15.1    | 841                      | 2065    | 76  | 8.5  | 0.45             | 0.65             | 45                 | NA                 | 2.1    | 0.90              |
| 8-9 Aug   | 23:40-01:15 | 6.8                                  | 12.6    | 315                      | 599     | 77  | 4.3  | 0.41             | 0.54             | 85                 | NA                 | 4.4    | 0.23              |
| 10 Aug    | 0:00-2:00   | 10.5                                 | 15.5    | 692                      | 1684    | 72  | 6.2  | 0.43             | 1.67             | 50                 | 1.10               | 4.6    | 0.55              |
| 17-18 Aug | 22:00-01:30 | 3.5                                  | 7.7     | 409                      | 802     | 60  | 9.5  | 0.55             | 0.48             | 33                 | 1.01               | 4.6    | 0.26 d |
| 25-26 Aug | 0:00-5:00   | 12.1                                 | 40.1    | 301                      | 534     | 74  | 11.8 | 0.62             | 2.10             | NA                 | 1.88               | 3.0    | 0.20              |

Table 1: Chemical characteristics of coal-fired plumes exhibiting high levels of CINO2 observed at Mt. Tai during the summer of 2014

a It represents the slope of SO2 vs NOy in plumes, and the overall slope for entire campaign was 0.31 with  $r^2$  of 0.31.

b Same to above note with the campaign overall slope of 15.7 and  $r^2$  of 0.23.

c Data not available in the case.

d For *tplumes* longer than the nocturnal processing period since sunset, the time since sunset was used in the ClNO2 yield calculation.

---

## Author Response (AR2)

*Wang et al. report measurements of N2O5 and ClNO2 at a mountain site on the North China Plain. The data contain occasional coal power plant plumes in which ClNO2 mixing ratios were enhanced. N2O5 uptake parameters and ClNO2 yields were calculated from a steady state model and found to be fast (i.e., large gamma and high phi) compared to other regions. Calculated uptake parameters were rationalized in terms of aerosol ionic composition (nitrate, chloride, and water calculated using a thermodynamic model). Overall the manuscript reads well and would be of interest to the community. Some of the methodology (such as the calculations of uptake parameters, equations 6 and 8,) require some clarification and may need refinement.*

We thank the reviewer for the time spent with our manuscript and the constructive suggestions. We have revised the manuscript and clarified the methodology according to the reviewer's comments. Our responses, including changes made to the manuscript, are listed below.

*Reviewer comments are in italics*. Author responses are in plain face. Changes to the text are in blue.

**Specific comments**

*1. pg 3 line 11 Consider citing Tham, Y., C. Yan, L. Xue, Q. Zha, X. Wang, and T. Wang (2014), Presence of high nitryl chloride in Asian coastal environment and its impact on atmospheric photochemistry, Chin. Sci. Bull., 59(4), 356-359, doi: 10.1007/s11434- 013-0063-y.*

*pg 3, line 12 "such as Colorado, Hessen and Canada" Hessen and Colorado are States of Germany and of the U.S.A. To be consistent, please list the third one as Alberta (a province in Canada).*

*pg, line 13. The Faxon et al 2015 study was in SE Texas and close to the coast and shouldn't be cited here. Instead consider citing Mielke, L. H., A. Furgeson, C. A. Odame-Ankrah, and H. D. Osthoff (2016), Ubiquity of ClNO2 in the nocturnal boundary layer of Calgary, AB, Canada, Canadian Journal of Chemistry, 94(4), 414-423, doi: 10.1139/cjc-2015-0426.*

*pg 3 line 14. Following " Anthropogenic sources of chlorine including coal combustion in power plants, industries, and biomass burning may potentially facilitate ClNO2 production" cite Riedel, T. P., et al. (2013), Chlorine activation within urban or power plant plumes: Vertically resolved ClNO2 and Cl2 measurements from a tall tower in a polluted continental setting, J. Geophys. Res., 118(15), 8702-8715, doi: 10.1002/jgrd.50637.*

Response: The suggested references are added or changed according to the reviewer's suggestions.

*2. pg 3, line 15 Please state the uncertainty of this number (4.7 ppbv +/- ?)*

Response: This value was a 1-min averaged maximum concentration, with an uncertainty of ±0.8 ppbv. The information is clarified in the revised text.

*3. pg 3, line 26. For readers not familiar with the NCP – please indicate the relative locations of Wangdu and Jinan – perhaps as dots in Figure 1?*

Response: The locations of the Wangdu and Jinan are indicated in Figure 1, as shown below:

[Figure]

*4. pg 3, line 31 "In the present study, we measured the concentrations of N2O5, ClNO2, and related species at a mountaintop site in the NCP and characterized the nighttime nitrogen chemistry within the residual layer over a polluted region of northern China. " please state month or season and the year here as the season (i.e., temperatures) are relevant for N2O5 chemistry.*

Response: The information of the season and the year is added.

*5. pg 4 line 14 " the measurement site is located in an area that is not frequently visited and therefore, should not be significantly affected by local anthropogenic emissions" Since Mt. Tai has been the site of other studies, the data ought to exist to definitively say whether the site IS or IS NOT affected by local emissions.*

*pg 4, line 17. Along the same lines, since a dozen power plants are mentioned, do the data from the earlier suggest that the site is impact by coal power plant emissions (e.g., high SO2, black carbon, or sulfate levels)?*

*Is this site sufficiently close or far enough to the coast to be impacted (or not) by sea salt aerosol?*

Response: Yes, as the reviewer points out, previous studies and data had proven the site was not affected by local emissions. Previous studies had also observed occasional combustion plumes with high levels of $SO_2$ (>20ppbv), sulfate (>20 $\mu g\ cm^{-3}$) and black carbon (>5 $\mu g\ cm^{-3}$) at the summit of Mt. Tai (Zhou et al., 2009; Wang et al., 2011).

The mountain site is about 230 km from the Bohai Sea and the Yellow Sea as illustrated in Figure 1a, and is likely to receive minor influences from sea salt aerosol.

To clarify, we have updated the Figure 1a, and revised the text to include this information, as follows:

"Mt. Tai is 230 km away from the Bohai and Yellow Seas, and the cities of Tai'an and Jinan (the capital of Shandong Province) are located 15 km south and 60 km north of the measurement site, respectively."

"Previous studies at this site indicated that the site is regionally representative without significant local anthropogenic emissions, and affected by the regional aged air masses and occasional combustion plumes from fossil fuel or biomass in the region (e.g., Zhou et al., 2009; Wang et al., 2011, Guo et al., 2012)."

References:

Guo, J., Wang, Y., Shen, X., Wang, Z., Lee, T., Wang, X., Li, P., Sun, M., Collett Jr, J. L., Wang, W., and Wang, T.: Characterization of cloud water chemistry at Mount Tai, China: Seasonal variation, anthropogenic impact, and cloud processing, *Atmos. Environ.*, 60, 467-476, 2012.

Wang, Z., Wang, T., Gao, R., Xue, L., Guo, J., Zhou, Y., Nie, W., Wang, X., Xu, P., Gao, J., Zhou, X., Wang, W., and Zhang, Q.: Source and variation of carbonaceous aerosols at Mount Tai, North China: Results from a semi-continuous instrument, Atmospheric Environment, 45, 1655-1667, 2011.

Zhou, Y., Wang, T., Gao, X. M., Xue, L. K., Wang, X. F., Wang, Z., Gao, J. A., Zhang, Q. Z., and Wang, W. X.: Continuous observations of water-soluble ions in PM2.5 at Mount Tai (1534 ma.s.l.) in central-eastern China, Journal of Atmospheric Chemistry, 64, 107-127, 10.1007/s10874-010-9172-z, 2009.

6. *pg 4 line 24 " .... which was replaced daily and washed in the ultrasonic bath to minimize wall loss caused by deposited particles (Wang et al., 2016)." Was there any change in signal levels after the line was washed? What time of day was the tubing changed?*

   *pg 4 line 27 "Manual calibrations of N2O5 and ClNO2 were conducted daily to monitor the instrument sensitivity and background." Please state what the CIMS response factors and associated uncertainties and background levels for each of the ions monitored were.*

   *pg 4, line 28 " The N2O5 standard was synthesized on-line from the reaction between NO2 and O3, and the ClNO2 was produced by passing a known concentration of N2O5 through a NaCl slurry " Please state how the N2O5 and ClNO2 concentrations of this source were quantified or what assumptions were made (e.g., ClNO2 yield from NaCl + N2O5).*

Response: The detailed calibration, testing and uncertainties of the CIMS measurement have been described in our previous papers (Wang et al., 2016, Tham et al., 2016). The same configuration was used in the present study. For clarity, we have included more detailed information in the revised text, as follows,

"The inlet was installed ~ 1.5 m above the roof of a single-story building, and the sampling line was a 5.5 m PFA-Teflon tubing (1/4 in. o.d.) which was replaced daily in the afternoon before sunset and washed in the ultrasonic bath to minimize wall loss caused by deposited particles (Wang et al., 2016). A small proportion (1.7 SLPM) of total sampling flow (~ 11 SLPM) was diverted to the CIMS system, to reduce the residence time of the air samples in the sampling line. A standard addition of $N_2O_5$ into the ambient inlet was performed before and after the tubing replacement to monitor the transmission efficiency, and this practice limited the loss of $N_2O_5$ in the inlet to <10% in the 'clean' tubing and about 30% in the next afternoon. Manual calibrations of $N_2O_5$ and $ClNO_2$ were conducted daily to determine the instrument sensitivity, and the average of which during the

observation period was 2.0 ± 0.6 for $N_2O_5$ and 2.2 ± 0.6 Hz $pptv^{-1}$ for $ClNO_2$, respectively. The $N_2O_5$ standard was synthesized on-line from the reaction between $NO_2$ and $O_3$, and the produced $N_2O_5$ were determined from the decrease in $NO_2$ (Wang et al., 2014). This method has been validated with a Cavity Ring Down Spectrometer (CRDS) measurement in previous campaign (Wang et al., 2016). The $ClNO_2$ was produced by passing a known concentration of $N_2O_5$ through a NaCl slurry assuming unity conversion efficiency (Roberts et al., 2009) and negligible $ClNO_2$ loss in the system (Wang et al., 2016). The field background was determined by passing the ambient sample through a filter packed with activated carbon, with average levels of 7.8 ± 1.9 and 6.0 ± 1.6 Hz for $N_2O_5$ and $ClNO_2$, respectively. The reported concentrations were derived by subtracting the background levels. The detection limit was 4 pptv for both $N_2O_5$ and $ClNO_2$ (2σ, 1 min-averaged data), and the uncertainty of the nighttime measurement is estimated to be ± 25% (Tham et al., 2016)."

References:

Roberts, J. M., Osthoff, H. D., Brown, S. S., Ravishankara, A. R., Coffman, D., Quinn, P., and Bates, T.: Laboratory studies of products of N2O5 uptake on Cl− containing substrates, Geophys. Res. Lett., 36, 10.1029/2009gl040448, 2009.

Tham, Y. J., Wang, Z., Li, Q., Yun, H., Wang, W., Wang, X., Xue, L., Lu, K., Ma, N., Bohn, B., Li, X., Kecorius, S., Größ, J., Shao, M., Wiedensohler, A., Zhang, Y., and Wang, T.: Significant concentrations of nitryl chloride sustained in the morning: Investigations of the causes and impacts on ozone production in a polluted region of northern China, Atmos. Chem. Phys., 16, 14959-14977, 2016.

Wang, X., Wang, T., Yan, C., Tham, Y. J., Xue, L., Xu, Z., and Zha, Q.: Large daytime signals of N2O5 and NO3 inferred at 62 amu in a TD-CIMS: chemical interference or a real atmospheric phenomenon?, Atmos. Meas. Tech., 7, 1-12, 10.5194/amt-7-1-2014, 2014.

Wang, T., Tham, Y. J., Xue, L., Li, Q., Zha, Q., Wang, Z., Poon, S. C. N., Dubé, W. P., Blake, D. R., Louie, P. K. K., Luk, C. W. Y., Tsui, W., and Brown, S. S.: Observations of nitryl chloride and modeling its source and effect on ozone in the planetary boundary layer of southern China, J. Geophys. Res. -Atmos., 10.1002/2015jd024556, 10.1002/2015jd024556, 2016.

*7. pg 5, line 10 " Water soluble ionic compositions of PM2.5... were measured hourly..." It would be great to show a time series of these data. Were chloride concentrations higher in power plant plumes?*

Response: The time series of selected ionic species, including sulfate, nitrate and chloride are included in the revised Figure 2. The chloride concentrations in the coal-fired plumes are higher than the campaign average, and this information has been summarized in the revised Table 1. More description about the aerosol composition is also added in the revised text, as follows,

"During the campaign at Mt. Tai, the average concentrations of aerosol sulfate and nitrate were 14.8 ± 9.0 and 6.0 ± 4.7 μg $m^{-3}$, accounting for 29.5% and 12.0% of $PM_{2.5}$ mass, respectively. The aerosol organic-to-sulfate ratio, a parameter that potentially affects the uptake process (Bertram et al., 2009b), was 0.74 on average and much lower than those from studies mentioned above in Europe and US. Moreover, the nighttime averaged $Cl^-$ concentration was 0.89 ± 0.86 μg $cm^{-3}$, and was an order of magnitude higher than $Na^+$, indicating abundant non-oceanic sources of chloride (e.g., from coal combustion and biomass burning in the NCP) (Tham et al., 2016), which could enhance the production of $ClNO_2$."

**Table 1: Chemical characteristics of coal-fired power plant and industrial plumes exhibiting high levels of ClNO₂ observed at Mt. Tai during the summer of 2014**

| Date | Duration | $N_2O_5$ (pptv) | | $ClNO_2$ (pptv) | | $O_3$ | $NO_x$ | $NO_x$ /$NO_y$ | $\Delta SO_2$ /$\Delta NO_y$ [a] | $\Delta CO$ /$\Delta NO_y$ [b] | $Cl^-$ (µg cm⁻³) | $t_{plume}$ | $\phi$ $ClNO2$ |
|---|---|---|---|---|---|---|---|---|---|---|---|---|---|
| | | Mean | Maximum | Mean | Maximum | | | | | | | | |
| 30-31 Jul | 23:40-0:45 | 5.9 | 14.2 | 528 | 1265 | 70 | 6.5 | 0.49 | 0.57 | 83 | 2.34 | 3.2 | 0.57 |
| 3-4 Aug | 23:30-0:00 | 20.1 | 23.8 | 506 | 833 | 106 | 2.8 | 0.22 | 2.43 | 108 | NA [c] | 4.9 | 0.64 |
| 7 Aug | 21:30-23:30 | 10.5 | 14.9 | 606 | 976 | 91 | 5.8 | 0.36 | 1.36 | 50 | 2.24 | 5.5 | 0.35 [d] |
| 8 Aug | 22:00-23:10 | 11.0 | 15.1 | 841 | 2065 | 76 | 8.5 | 0.45 | 0.65 | 45 | NA | 2.1 | 0.90 |
| 8-9 Aug | 23:40-01:15 | 6.8 | 12.6 | 315 | 599 | 77 | 4.3 | 0.41 | 0.54 | 85 | NA | 4.4 | 0.23 |
| 10 Aug | 0:00-2:00 | 10.5 | 15.5 | 692 | 1684 | 72 | 6.2 | 0.43 | 1.67 | 50 | 1.10 | 4.6 | 0.55 |
| 17-18 Aug | 22:00-01:30 | 3.5 | 7.7 | 409 | 802 | 60 | 9.5 | 0.55 | 0.48 | 33 | 1.01 | 4.6 | 0.26 [d] |
| 25-26 Aug | 0:00-5:00 | 12.1 | 40.1 | 301 | 534 | 74 | 11.8 | 0.62 | 2.10 | NA | 1.88 | 3.0 | 0.20 |

[a] It represents the slope of $SO_2$ vs $NO_y$ in plumes, and the overall slope for entire campaign was 0.31 with $r^2$ of 0.31.

[b] Same to above note with the campaign overall slope of 15.7 and $r^2$ of 0.23.

[c] Data not available in the case.

[d] For $t_{plumes}$ longer than the nocturnal processing period since sunset, the time since sunset was used in the $ClNO_2$ yield calculation.

[Figure]

**Figure 2 : Time series for N₂O₅, ClNO₂, related trace gases, aerosol properties, and meteorological data measured at Mt. Tai from July 24 to August 27, 2014.**

References:

Tham, Y. J., Wang, Z., Li, Q., Yun, H., Wang, W., Wang, X., Xue, L., Lu, K., Ma, N., Bohn, B., Li, X., Kecorius, S., Größ, J., Shao, M., Wiedensohler, A., Zhang, Y., and Wang, T.: Significant concentrations of nitryl chloride sustained in the morning: Investigations of the causes and impacts on ozone production in a polluted region of northern China, Atmos. Chem. Phys., 16, 14959-14977, 2016.

*8. pg 5 line 26 and Figure 2. There is a lot of sustained N2O5 during daytime which is unusual especially since temperatures frequently were >20C during the day. Is this a real signal? I am a bit doubtful. In any case, it warrants discussion that N2O5 at 3 pm was in greater abundance than at 3 am (Figure 3).*

*I am also suspicious about the relatively low levels at night. Was the N2O5 transmission efficiency monitored? If so, please show those data. If not, please state that it was not.*

*It also looks like there is a hardly any NO during the day which is consistent with there being N2O5, but is strange also as NO2 should photo-dissociate and sustain NO. Perhaps the O3 levels are high enough and photolysis rates low enough for this to happen. In my opinion, this warrants a bit of analysis & discussion – are the levels consistent with a simple photostationary analysis, or is the pss severely perturbed (i.e., too much NO2 relative to NO)?*

Response: We thank the reviewer for bringing the daytime $N_2O_5$ signals to our attention. According to the suggestions, we have further examined the daytime data. Because the daily maintenance and calibrations of CIMS were usually performed in the early afternoon, there were only a few cases with available afternoon data for analysis, which also accounts for the larger variability of $N_2O_5$ in the early afternoon. Using the photostationary analysis proposed by Brown et al (2005; 2016), we also calculated the daytime steady-state concentration of $N_2O_5$ for cases with daytime peaks. The predicted $N_2O_5$ concentrations showed an increasing trend in the afternoon, similar to the observation pattern for diurnal average, but the absolute values varied for different cases, as the examples shown in the below figure. The calculated $N_2O_5$ concentrations around 15:00 were much lower for a clean sky case with high photolysis rate (panel a in below figure), but were of the same magnitude as the observation for a reduced photolysis and foggy condition with higher $NO_3$ production rate (panel b in below figure). The limited daytime data do not allow us to perform further analysis to investigate the detailed reasons for this daytime phenomenon. Whether there were some interference signals or any other reasons still require additional studies.

To clarify, we have added the discussions in the revised text, and also included the comparison between steady-state predicted and observed $N_2O_5$ for two daytime cases in the supplement figure. In addition, as we described in the previous responses #6, the $N_2O_5$ transmission efficiency was monitored before and after the tubing replacement, and this information, as well as the uncertainties in the measurement, have been included in the revised text.

[Figure]

**Figure S1: Observed and steady-state calculated daytime N₂O₅, nitrate radical production rate and meteorological parameters for the case of (a) 27 July and (b) 24 August, 2014.**

The revised text reads,

"It was also noted that a small $N_2O_5$ peak (~10 pptv) with larger variability was present in the early afternoon. A simplified photostationary analysis following Brown et al. (2005; 2016) was performed to predict the daytime steady-state $N_2O_5$ concentrations for the few cases with daytime peaks. The predicted concentrations all showed increasing trends in the afternoon, similar to the observation pattern. However, for individual cases, the absolute values around 15:00 were much lower than observation under clean sky condition, but of the same magnitude as the observation for reduced photolysis and foggy conditions with higher $NO_3$ production rate (c.f. Figure S1 in the supplement). Daytime $N_2O_5$ signals with few pptv have also been observed by a CRDS at a mountain site in southern China (Brown et al., 2016), where the concentrations were in accord with steady state estimation in an average sense. Because daily maintenance and calibrations of the CIMS were usually performed during early afternoon periods, the limited daytime data in the present study was not sufficient to make clear whether there were any daytime interferences or sensitivity fluctuations. Thus additional studies are needed to validate the daytime phenomenon and examine the potential reasons, and the following analysis in the present work will mostly focus on nocturnal process."

*The assumption here is that ClNO2 is conserved and is not produced by other sources (such as Cl + NO2) – please state these assumptions.*

*The paragraph that follows equation 6 does not provide enough information. How was the yield of ClNO2 actually calculated? Was the N2O5 uptake loss truly integrated using a time-integrated box-model? Or was it approximated using the ratio of observed mixing ratios (as suggested by the first term in eqn 6)? Note that rates change from the point of emission to the point of observation.*

Response: We thank the reviewer for the valuable suggestion. The equation has been revised and the first two terms have been removed. The description on ClNO₂ yield calculation is clarified, and assumptions made in the estimation are included in the revised text, as follows:

"For regional diffuse pollution cases, the $\phi$ defined in R3 can be estimated from the ratio between ClNO₂ production rate and N₂O₅ loss rate, as the first term in below equation:

$$\phi = \frac{d\text{ClNO}_2/dt}{k(\text{N}_2\text{O}_5)_{\text{het}}[\text{N}_2\text{O}_5]} = \frac{[\text{ClNO}_2]}{\int k(\text{N}_2\text{O}_5)_{\text{het}}[\text{N}_2\text{O}_5]\,dt} \qquad (6)$$

$k(\text{N}_2\text{O}_5)$ values can be determined using the inverse steady-state lifetime analysis described above in Eq. 3, and the production rate of ClNO₂ can be derived from the near-linear increase in ClNO₂ mixing ratio observed during a period, when the related species (e.g., NOₓ, SO₂) and environmental variables (e.g., temperature, RH) were roughly constant. The approach here assumes that the relevant properties of the nocturnal air mass are conserved, and neglects other possible sources and sinks of ClNO₂ in the air mass history. For the intercepted coal-fired plumes exhibiting sharp ClNO₂ peaks, the ClNO₂ yield can be estimated from the ratio of the observed ClNO₂ mixing ratio to the integrated N₂O₅ uptake loss over the plume age (i.e., the second term in Eq. 6). The analysis assumes that no ClNO₂ was present at the point of plume emission from the combustion sources and no ClNO₂ formation before sunset, and that the γ and ϕ within the plumes did not change during the transport from the source to the measurement site. The potential variability in these quantities likely bias the estimates, but these assumptions are a necessary simplification to represent the averaged values that best describe the observations. It should be noted that the steady-state N₂O₅ loss rate is crucial in the yield estimation, which could be underestimated by potentially overestimating the loss rate in some cases with large uncertainties in N₂O₅ measurement and NO₃ reactivity analysis. Therefore, an alternative approach suggested by Riedel et al. (2013) was also applied to derive the ClNO₂ yield from the ratio of enhancements of ClNO₂ and total nitrate (aerosol NO₃⁻ + HNO₃) in the cases. Given the low time resolution of nitrate data that could potentially introduce large uncertainties, this approach will only be used as a reference to validate the former analysis based on Eq. 6. "

    *Also, if concentrations of N2O5 are underestimated due to a measurement bias and real concentrations of N2O5 were higher, things fall into place ... (see earlier comments)*

    *pg 12, line 2. Please state that the chloride concentration was measured and the water content is based on a thermodynamical model and perform an error estimate (so that appropriate error bars can be added to Figure 10).*

Response: The reviewer's observation is correct; the dependence of parameterized $\phi$ on Cl$^-$/H$_2$O is derived from the color code in the Figure 10a. We have made this clear in the revised text, as follows,

"The parameterized $\phi$ values exhibit positive dependence on the aerosol chloride concentration and the Cl$^-$/H$_2$O ratio, as shown by the color code in Fig 10a."

The method for calculating the water content is also clarified in the revised text. For the error in water content calculation, sensitivity tests showed that a 3% change in RH implied an uncertainty in the particle liquid water content of ~5%. Thus we included this error information and the overall uncertainty estimation in the revised text, and added the error bars in the revised Fig 8 and Fig 10.

In addition, as described in previous comments, the measurement uncertainty of N$_2$O$_5$ is included in the revised text, and the propagated uncertainty of estimated uptake coefficients ($\gamma$) and yield ($\phi$) accounting for errors associated with the measurements and statistical uncertainty in the calculations, are also added in the revised text and figures.

The clarified and revised text reads,

"An error estimation showed that a 3% change in RH implies an uncertainty in the particle liquid water content of ~5%. In the calculation, mean values of $V/S$ (64.8 - 77.2 nm) measured in the present study instead of empirical pre-factor $A$ were used, and the reaction rate coefficients were employed as the empirical values suggested by Bertram and Thornton (2009)."

"We compared the field-derived values to the parameterization for cases with available aerosol compositions, using an empirical $k'$ of 1/450, as recommended by Roberts et al. (2009). The particle liquid water content [H$_2$O] was calculated from the thermodynamic model (E-AIM model IV) based on measured aerosols composition, as described above."

[Figure]

**Figure 8: Comparison of field-determined $\gamma$ with that derived from the parameterization of Bertram and Thornton, (2009). The colors of the markers indicate the corresponding concentrations ratio of particulate chloride to nitrate. The error bars represent the total aggregate uncertainty associated with measurement and derivation.**

[Figure]

**Figure 10: (a)** Comparison of field-determined $\phi$ with that derived from parameterization (Eq. 7), and the colors of the markers represent the corresponding $Cl^-/H_2O$ ratio; **(b)** relationship between field-determined $\phi$ and measure nitrate concentrations in aerosols, and colors of markers represent the corresponding $NO_x/NO_y$ ratio. The error bars represent the total aggregate uncertainty as similar as Figure 8.

*18.     pg 12, eqn 8. This equation is not correct. It ought to be the loss rate of N2O5, not its rate of change (d[N2O5]/dt) which would be zero at steady state. There are major assumptions made here – that aerosol nitrate is conserved, i.e., absence of aerosol deposition and volatilization via NH4NO3 formation etc. These assumptions should be clearly stated.*

*pg 12, line 31. "The NO3 formation was predicted by integrating each derived formation rate over the corresponding ..." the formation rate of nitrate changes from the point emission to the point of observation, which was not taken into account here.*

Response: We thank the reviewer's helpful suggestions. The equation has been corrected by removing the first term. We also clarified the definition of nitrate formation rate here as the nitrate exclusively produced from $N_2O_5$ reactions. The assumptions made for estimating the formation rate and production of nitrate are now included and clarified in the revised text, as follows,

"Based on the reactions described above, the formation rate of soluble nitrate from $N_2O_5$ reactions, $p(NO_3^-)$, can be determined from the $ClNO_2$ yield and $N_2O_5$ heterogeneous loss rate as follows:

$$p(NO_3^-) = (2 - \phi)k_{N_2O_5}[N_2O_5] \qquad (8)$$

The $p(NO_3^-)$ values obtained for the select cases during the study period ranged from 0.02 to 0.62 ppt s$^{-1}$, with a mean value of $0.29 \pm 0.18$ ppt s$^{-1}$, corresponding to 0.2- 4.8 µg m$^{-3}$ hr$^{-1}$ and $2.2 \pm 1.4$ µg m$^{-3}$ hr$^{-1}$ (Table 2). The derived rates are comparable to the observed increases in nitrate concentrations (2-5 µg m$^{-3}$ h$^{-1}$) during haze episodes in summer nights at a rural site in the NCP (Wen et al., 2015). By assuming that produced nitrate is conserved and neglecting the deposition and volatilization loss (e.g., via ammonium nitrate), the in-situ $NO_3^-$ formation could be predicted

by integrating each derived formation rate over the corresponding analysis period. Similar to $N_2O_5$ uptake coefficient and $ClNO_2$ yield determination above, the nitrate formation estimation here assumes a conserved air mass with a constant formation rate over the study period. For coal-fired plumes, we equated the measured nitrate concentrations with the increases by assuming that no aerosol nitrate was directly emitted from the nocturnal point sources."

*19.    pg 13, line 16. Wang et al., 2016 is not the best reference. Please cite Behnke, W., C. George, V. Scheer, and C. Zetzsch (1997), Production and decay of ClNO2, from the reaction of gaseous N2O5 with NaCl solution: Bulk and aerosol experiments, J. Geophys. Res., 102(D3), 3795-3804, doi: 10.1029/96JD03057 instead.*

Response: The reference is changed according to the reviewer's suggestion.

*20.    pg 13, lines 16/17 "For simplicity, the reactions of NO3 with VOCs can be assumed to result in the complete removal of reactive nitrogen (Wagner et al., 2013)". This is likely not true in this study, where NO3 primarily reacts with terpenes. Wangberg et al. (1997), Product and mechanistic study of the reaction of NO3 radicals with alphapinene, Environm. Sci. Technol., 31(7), 2130-2135, and others since have showed that a significant fraction of NO2 is ultimately released again.*

Response: We thank the reviewer for pointing out this issue. We have revised the text to include this information and also added the caveat that possible overestimation on $NO_x$ loss because of $NO_2$ recycling from $NO_3$-VOC reactions. The revised text reads,

"$ClNO_2$ mainly functions as a reservoir of $NO_x$, rather than as a sink, because the formation of $ClNO_2$ throughout the night with subsequent morning photolysis recycles $NO_2$ (Behnke et al., 1997). The reactions of $NO_3$ with VOCs would predominantly produce organic nitrate products (Brown and Stutz, 2012 and references therein), but some fraction of $NO_2$ can be regenerated in the $NO_3$ reactions (i.e., with terpenes) (e.g., Wangberg et al., 1997) or released from the decomposition of organic nitrate during the transport (e.g., Francisco and Krylowski, 2005). For simplicity, we neglect the recycling of $NO_2$ from $NO_3$-VOC reactions by assuming the complete removal of reactive nitrogen (Wagner et al., 2013). This would overestimate the $NO_x$ loss since the monoterpenes contribute to around half of $NO_3$ reactivity at the present study, but this assumption does not significantly affect the conclusion because $NO_3$ loss with VOCs was the minor path comparing to $N_2O_5$ heterogeneous loss. Thus, the nocturnal $NO_x$ loss rate can be quantified by the following equation:"

Added references:

Behnke, W., George, C., Scheer, V., and Zetzsch, C.: Production and decay of ClNO2 from the reaction of gaseous N2O5 with NaCl solution: Bulk and aerosol experiments, J. Geophys. Res. -Atmos., 102, 3795-3804, 10.1029/96JD03057, 1997.

Brown, S. S., and Stutz, J.: Nighttime radical observations and chemistry, Chem. Soc. Rev., 41, 6405-6447, 10.1039/C2CS35181A, 2012.

Wängberg, I., Barnes, I., and Becker, K. H.: Product and Mechanistic Study of the Reaction of NO3 Radicals with α-Pinene, Environ. Sci. Technol., 31, 2130-2135, 10.1021/es960958n, 1997.

Francisco, M. A., and Krylowski, J.: Chemistry of Organic Nitrates: Thermal Chemistry of Linear and Branched Organic Nitrates, Industrial & Engineering Chemistry Research, 44, 5439-5446, 10.1021/ie049380d, 2005.

21.    pg 13, line 22. *"The NOx removal rate varied from 0.19 to 2.34 ppb h-1, with a mean of 1.12 ± 0.63 ppb h-1. This loss rate is higher than that determined from tower measurements during wintertime in Colorado, with integrated nocturnal NO2 loss ranging from 2.2 to 4.4 ppbv (Wagner et al., 2013)" Comparing absolute loss rates may not be meaningful since the overall NOx levels may be different – consider normalizing, for example, through division of average nocturnal NOx mixing ratios at both locations to derive a pseudo-first order loss rate coefficient.*

Response: We have included the normalized loss rate coefficient of $NO_x$ in the revised text according to the suggestion. Wagner et al. (2013) did not provide the loss rate or loss rate coefficient in their work, so we have updated the comparison with other references. The revised text reads:

"Using the coefficients described above, we calculated the nocturnal loss rate of $NO_x$ for each case, as summarized in Table 2. The $NO_x$ removal rate varied from 0.19 to 2.34 ppb $h^{-1}$, with a mean of 1.12 ± 0.63 ppb $h^{-1}$, which corresponds to a pseudo-first order loss rate coefficient of 0.24 ± 0.08 $h^{-1}$ in average for the studied cases. This loss rate is higher than that determined from a mountain site measurement in Taunus, Germany (~0.2 ppb $h^{-1}$ with typical $NO_2$ level of 1-2 ppb) (Crowley et al., 2010), and the results from aircraft measurements in US over Ohio and Pennsylvania and downwind region of New York (90% and 50% $NO_x$ loss in a 10-hour night, respectively) (Brown et al., 2006). For reference, this nocturnal average loss rate is approximately equivalent to $NO_2$ loss via reaction with OH at afternoon condition assuming OH concentration around $2 \times 10^6$ molecules $cm^{-3}$, indicating the importance of nocturnal heterogeneous reactions on $NO_x$ processing and budget."

Response: We thank the reviewer for the valuable suggestion. Additional description and discussion of the aerosol characteristic are included in the revised text. The time series of measured aerosol nitrate, sulfate, chloride and surface area was also added in the revised Figure 2, as follows:

"The average nighttime mixing ratios of $O_3$ and $NO_2$ were 77 and 3.0 ppbv, respectively, with an average nitrate radical production rate $p(NO_3)$ of $0.45 \pm 0.40$ ppb $h^{-1}$, which is indicative of potentially active $NO_3$ and $N_2O_5$ chemistry during the study period. However, the low $N_2O_5$ mixing ratios observed during most of the nights suggest a rapid loss of $N_2O_5$, which is consistent with the observed high aerosol surface area (Sa), varied from ~100 to 7800 $\mu m^2$ $cm^{-3}$ with a mean value of 1440 $\mu m^2$ $cm^{-3}$."

"The elevated $ClNO_2$ levels observed at Mt. Tai are similar to recent measurements at a surface rural site (Wangdu) in northern China (Tham et al., 2016) and a mountain site (Tai Mo Shan) in southern China (Wang et al., 2016), but are slightly higher than previous measurements conducted in coastal (e.g., Osthoff et al., 2008; Riedel et al., 2012; Mielke et al., 2013) and inland sites (e.g., Thornton et al., 2010; Phillips et al., 2012; Riedel et al., 2013) in other regions of the world. During the campaign at Mt. Tai, the average concentrations of aerosol sulfate and nitrate were $14.8 \pm 9.0$ and $6.0 \pm 4.7$ $\mu g$ $m^{-3}$, accounting for 29.5% and 12.0% of PM2.5 mass, respectively. The aerosol organic-to-sulfate ratio, a parameter that potentially affects the uptake process (Bertram et al., 2009b), was 0.74 on average and much lower than those from studies mentioned above in Europe and US. Moreover, the nighttime averaged $Cl^-$ concentration was $0.89 \pm 0.86$ $\mu g$ $cm^{-3}$, and was an order of magnitude higher than $Na^+$, indicating abundant non-oceanic sources of chloride (e.g., from coal combustion and biomass burning in the NCP) (Tham et al., 2016), which could enhance the production of $ClNO_2$."

[Figure]

**Figure 2 : Time series for N₂O₅, ClNO₂, related trace gases, aerosol properties, and meteorological data measured at Mt. Tai from July 24 to August 27, 2014.**

*physical circumstances were, with the evidence at hand. Another approach needs to be found, or the analysis should be abandoned.*

Response: We agree with the reviewer that the growth rate estimation could be biased due to the potential variation of the plume, and simplified estimation would result in some uncertainties. We were aware of these limitations in the estimation, and therefore had carefully inspected the time series to choose the data during a period when related parameters in the air mass were relatively constant. It is likely that the assumptions are reasonable during the short time periods, usually around 2 to 3 hours. In addition, we have applied an alternative approach to derive the $ClNO_2$ yields from the ratio of observed enhancements of $ClNO_2$ and total nitrate (aerosol $NO_3^-$ and $HNO_3$), according to the method suggested by Riedel et al. (2013). The derived $\phi$ values from this approach exhibit reasonable agreement with the original analysis, and most of the differences between two groups of data are within 40% (see figure below). Although either approach requires assumptions and would introduce some uncertainties, the general consistency can serve as a check to corroborate the yield analysis.

To clarify, we have revised the text by elaborating these assumptions in the estimation and the criteria for selecting cases, and also added the comparison results between two different approaches in the revised version, as follows:

"For regional diffuse pollution cases, the $\phi$ defined in R3 can be estimated from the ratio between $ClNO_2$ production rate and $N_2O_5$ loss rate, as the first term in below equation:

$$\phi = \frac{dClNO_2/dt}{k(N_2O_5)_{het}[N_2O_5]} = \frac{[ClNO_2]}{\int k(N_2O_5)_{het}[N_2O_5]\, dt} \qquad (6)$$

$k(N_2O_5)$ values can be determined using the inverse steady-state lifetime analysis described above in Eq. 3, and the production rate of $ClNO_2$ can be derived from the near-linear increase in $ClNO_2$ mixing ratio observed during a period, when the related species (e.g., $NO_x$, $SO_2$) and environmental variables (e.g., temperature, RH) were roughly constant. The approach here assumes that the relevant properties of the nocturnal air mass are conserved, and neglects other possible sources and sinks of $ClNO_2$ in the air mass history. For the intercepted coal-fired plumes exhibiting sharp $ClNO_2$ peaks, the $ClNO_2$ yield can be estimated from the ratio of the observed $ClNO_2$ mixing ratio to the integrated $N_2O_5$ uptake loss over the plume age (i.e., the second term in Eq. 6). The analysis assumes that no $ClNO_2$ was present at the point of plume emission from the combustion sources and no $ClNO_2$ formation before sunset, and that the $\gamma$ and $\phi$ within the plumes did not change during the transport from the source to the measurement site. The potential variability in these quantities likely bias the estimates, but these assumptions are a necessary simplification to represent the averaged values that best describe the observations. It should be noted that the steady-state $N_2O_5$ loss rate is crucial in the yield estimation, which could be underestimated by potentially overestimating the loss rate in some cases with large uncertainties in $N_2O_5$ measurement and $NO_3$ reactivity analysis. Therefore, an alternative approach suggested by Riedel et al. (2013) was also applied to derive the $ClNO_2$ yield from the ratio of enhancements of $ClNO_2$ and total nitrate (aerosol $NO_3^-$ + $HNO_3$) in the cases. Given the low time resolution of nitrate data that could potentially introduce large uncertainties, this approach will only be used as a reference to validate the former analysis based on Eq. 6."

"The determined $\phi$ for the seven coal-fired plumes are also listed in Table 1. During the measurement period, $\phi$ varied from 0.02 to 0.90, with an average of $0.28 \pm 0.24$ and a median of 0.22. In comparison, the $\phi$ derived from the production ratio approach showed comparable results with an average of $0.25 \pm 0.17$, and the $\phi$ values from two different approaches match reasonably well with a Reduced Major Axis Regression (RMA) slope of $0.78 \pm 0.08$ and $r^2$ of 0.73 (cf Figure S4), which corroborates the yield analysis and indicates that the differences are within the overall uncertainty of 40%."

[Figure]

**Figure S4: Comparison of estimated ClNO₂ yields from two different approaches: approach A using the ratio of the observed ClNO₂ growth rate to steady-state N₂O₅ loss rate based on Eq. 6; approach B using the production ratio of observed enhancements of ClNO₂ and total nitrate, $\phi = 2/(\Delta NO_3^-/\Delta ClNO_2 + 1)$ according to Riedel et al., 2013.**

Response: The equation is corrected by removing the first term.

**Anonymous Referee #3**

*General comments:*
*This paper presents measurement of N2O5 and ClNO2 from a polluted mountaintop site in the North China Plain during summer 2014. Measurements of these nighttime reactive nitrogen species in the polluted residual layer of China are novel and a valuable contribution to the literature. The authors attribute several of the plumes encountered at the mountaintop site to emissions from regional coal fired power plants. They further use several standard analysis metrics to interpret the data and provide estimates of N2O5 uptake coefficients and ClNO2 yields, along with the overall influence of nighttime chemistry on aerosol nitrate formation in the region. Results for N2O5 uptake coefficients are generally larger and ClNO2 yields are generally smaller than previous literature determinations. These observations may not be unrelated. If the analysis method biases the N2O5 uptake coefficient to large values, then the same analysis will tend to predict lower ClNO2 yields. The authors should be careful to consider uncertainties that could lead to such a bias, especially in the aerosol size distribution measurement and in the assessment of NO3-VOC reactivity. Alternatively, the very high relative humidity at this site could lead to exactly the effect that is found here, producing faster N2O5 reactivity but also a larger fraction tending toward HNO3 rather than ClNO2. The paper could make this point explicitly in its comparison to previous work (e.g. Phillips, et al., Wagner et al.). The paper should be published subject to these comments and the minor comments below.*

We thank the reviewer for the valuable suggestions. We are aware of the uncertainties in the measurement and analysis that could potentially bias the results. In the revised text, we have included a more detailed description of the measurement uncertainty in the methodology part, including $N_2O_5$ transmission efficiency, calibrations and surface area calculation. We have also clarified the assumptions and uncertainties in the $NO_3$ reactivity estimation and determination of uptake coefficient and yield. Additional approach for yield estimation is also applied as a check to corroborate the analysis and results. More discussions on the observed high uptake coefficient and the possible effects of high humidity and aerosol characteristics comparing to previous studies are also added in the revised text.

Our responses to the comments, including changes made to the manuscript, are listed point-by-point below.

*Reviewer comments are in italics.* Author responses are in plain face. Changes to the text are in blue.

*Specific comments.*

*1. Page 4, line 16-17. Is there a database showing the location of major coal fired power plants that could be included with the map in Figure 1? This would help to clarify the number of sources and their distance from the observatory.*

Response: The location information of major coal-fired facilities in the industry (cement and steel production) and power plants in the region is included in the maps in Figure 4c and 5c, to aid the discussion in the section of coal-fired plumes (section 3.2). Figure 4c is shown below for reference:

[Figure]

*2. Page 4, line 27. What were the results of the manual calibrations for N2O5 and ClNO2? Give some sense for reproducibility, and lack of either N2O5 loss or ClNO2 generation on the inlets.*

Response: To clarify, we added a more detailed description of the $N_2O_5$/$ClNO_2$ measurement, including transmission efficiency, calibrations and the uncertainties in the methodology part. The revised text reads,

"The inlet was installed ~ 1.5 m above the roof of a single-story building, and the sampling line was a 5.5 m PFA-Teflon tubing (1/4 in. o.d.) which was replaced daily in the afternoon before sunset and washed in the ultrasonic bath to minimize wall loss caused by deposited particles (Wang et al., 2016). A small proportion (1.7 SLPM) of total sampling flow (~ 11 SLPM) was diverted to the CIMS system, to reduce the residence time of the air samples in the sampling line. A standard addition of $N_2O_5$ into the ambient inlet was performed before and after the tubing replacement to monitor the transmission efficiency, and this practice limited the loss of $N_2O_5$ in the inlet to <10% in the 'clean' tubing and about 30% in the next afternoon. Manual calibrations of $N_2O_5$ and $ClNO_2$ were conducted daily to determine the instrument sensitivity, and the average of which during the observation period was $2.0 \pm 0.6$ for $N_2O_5$ and $2.2 \pm 0.6$ Hz pptv$^{-1}$ for $ClNO_2$, respectively. The $N_2O_5$ standard was synthesized on-line from the reaction between $NO_2$ and $O_3$, and the produced $N_2O_5$ were determined from the decrease in $NO_2$ (Wang et al., 2014). This method has been validated with a Cavity Ring Down Spectrometer (CRDS) measurement in previous campaign (Wang et al., 2016). The $ClNO_2$ was produced by passing a known concentration of $N_2O_5$ through a NaCl slurry assuming unity conversion efficiency (Roberts et al., 2009) and negligible $ClNO_2$ loss in the system (Wang et al., 2016). The field background was determined by passing the ambient sample through a filter packed with activated carbon, with average levels of $7.8 \pm 1.9$ and $6.0 \pm 1.6$ Hz for $N_2O_5$ and $ClNO_2$, respectively. The reported concentrations were derived by subtracting the background levels. The detection limit was 4 pptv for both $N_2O_5$ and $ClNO_2$ ($2\sigma$, 1 min-averaged data), and the uncertainty of the nighttime measurement is estimated to be $\pm$ 25% (Tham et al., 2016)."

*For comparison, what was the overall relationship between CO and NOy or SO2 and NOy for the entire campaign? If most of the NOy is urban, then the global relationships might define the urban numbers so that the power plants could be more easily distinguished.*

Response: We appreciate the reviewer's helpful suggestions. We examined the correlations among $SO_2$, $NO_y$ and CO in these cases to figure out the source information. The slopes of $SO_2$ vs $NO_y$ in these cases ranged from 0.48 to 2.43, and were all higher than the overall slope of 0.31 for the entire campaign, indicating the coal combustion source of these plumes. The slope of CO vs $NO_y$ varied from 33 to 108, and was also higher than the campaign global slope of 15.7. As we stated in the previous response, both high $SO_2/NO_y$ and $CO/NO_y$ slopes (and ratios) suggest that the plumes were likely originated from coal-fired facilities in industrial plants, whereas high $SO_2/NO_y$ with relatively lower $CO/NO_y$ possibly suggest the source from power plants.

In the revised version, we added the derived slopes of $SO_2$ vs $NO_y$ and CO vs $NO_y$ in Table 1, and also included the ratios of $SO_2/NO_y$ and $CO/NO_y$ in Figure 4a and 5a. The text and discussions are also clarified and revised to match these changes, as follows,

[revised manuscript text omitted]

**Revised Figure 4a and Figure 5a: Time series for $ClNO_2$, $N_2O_5$, and related trace gases observed within the high-$ClNO_2$ coal-fired plume during the nights of July 30-31, 2014 (left panel) and August 8-9, 2014 (right panel)**

5. *Page 7, line 17. Authors probably mean a slope steeper than -1 (rather than +1).*

Response: The typo is corrected in the revised text.

*6. Page 8, line 25. The determination of N2O5 reactivity is not quite clear. Text implies that equation 3 is used, and that k(NO3)/Keq[NO2] is subtracted from this number based on the measured VOCs from a different year. Correct? If so, this should be stated explicitly, possibly with an equation.*

*If the above is correct, then for the sake of clarity, the NO3 loss rates quoted in line 23 should be divided by Keq[NO2] to make it obvious how the budget was done.*

*What is not given here is a sense for the uncertainty (e.g., N2O5 contributions of 80% and 71% given to two significant figures with no uncertainty). Since VOC measurements from a separate year are used, and since the NO3 reactivity is dominated by reaction with monoterpenes, which are variable and quite temperature dependent, there could be substantial year to year variability and thus considerable uncertainty in this budget. At the very least, this uncertainty should be qualitatively noted. If the authors have data that would quantify year to year or night to night variability in the NO3 losses, then those numbers should be used to formulate a quantitative error budget.*

Response: We thank the reviewer for the comments on the uncertainty and pointing out the unclear description. In the revised version, we have clarified the description of the determination of $N_2O_5$ reactivity, and also added more statement on the uncertainty of $NO_3$ relativity estimation. Moreover, we also changed the text to avoid using the exact number of 80% and 71%, but reported the results in range to reflect the potential uncertainty.

The clarified and revised text reads:

"The determined nighttime $k(NO_3)$ was $1.33 \times 10^{-2}\,s^{-1}$ for the first half of the night and $1.07 \times 10^{-2}\,s^{-1}$ for the period after midnight, which is equivalent to an $NO_3$ lifetime of approximately 1.5 min. The estimated $k(NO_3)$ could be considered as an upper limit for coal-fired plumes because of potential lower biogenic VOC levels within the plumes. The estimation here does not account for the VOC changes between years and the night to night variability, which may result in uncertainties. The $k(NO_3)$ derived by another approach, i.e., from the nighttime steady state fits, provides a consistency check and evaluation of the errors, as described below.

The heterogeneous loss rate, $k(N_2O_5)_{het}$, can be obtained by subtracting the $k(NO_3)/K_{eq}[NO_2]$ from the determined $\tau(N_2O_5)^{-1}$ in Eq.3. Figure 6a shows the averaged total $N_2O_5$ reactivity and fractions of $N_2O_5$ loss via $NO_3$ ($k(NO_3)/K_{eq}[NO_2]$) and heterogeneous $N_2O_5$ loss during the study period. As shown, the heterogeneous loss was dominant, accounting for 70-80% of total $N_2O_5$ reactivity with higher fraction before midnight. Figure 6b shows the contribution of different VOC categories to the average first-order $NO_3$ loss rate coefficients, $k(NO_3)$. Biogenic monoterpenes accounted for more than half of the $NO_3$ reactivity, followed by anthropogenic alkenes (such as butene), isoprene and dimethyl sulfide (DMS). Aromatics and alkanes made small contributions (<1%) to the total $NO_3$ reactivity. Although some unmeasured organic species (e.g., peroxy radicals) could also contribute to a small fraction of $NO_3$ loss (Brown et al., 2011; Edwards, et al., 2017), the dominant $NO_3$ reactivity by biogenic VOCs is similar to that observed at a mountain site in southern China (Brown et al., 2016) and aircraft measurement in residual layer in southeast US (Edwards, et al., 2017), whereas the anthropogenic contribution is much higher in the present study. The estimated

NO$_3$ activity is slightly lower than that obtained from surface site measurements in the NCP (Tham et al., 2016; Wang et al., 2017b), which is in line with the higher abundances of VOCs in the polluted boundary layer."

"The average $k$(NO$_3$) derived from the steady state fits is $0.015 \pm 0.010$ s$^{-1}$, which is comparable to that predicted from the VOC concentrations described above, indicating that the estimated results in the present study are reliable and likely representative of averaged conditions in the region. The agreement between these two methods also corroborates the determination of the uptake coefficient from steady state analysis. The estimated uncertainty in each individual determination varied from 35 to 100%, including statistical errors and uncertainty associated with measurements of gaseous and aerosol species (Tham et al., 2016)."

The Figure 8 and 10 are also updated to include the error bars, as shown below.

[Figure]

**Figure 8: Comparison of field-determined γ with that derived from the parameterization of Bertram and Thornton, (2009). The colors of the markers indicate the corresponding concentrations ratio of particulate chloride to nitrate. The error bars represent the total aggregate uncertainty associated with measurement and derivation.**

[Figure]

**Figure 10: (a)** Comparison of field-determined $\phi$ with that derived from parameterization (Eq. 7), and the colors of the markers represent the corresponding $Cl^-/H_2O$ ratio; **(b)** relationship between field-determined $\phi$ and measure nitrate concentrations in aerosols, and colors of markers represent the corresponding $NO_x/NO_y$ ratio. The error bars represent the total aggregate uncertainty as similar as Figure 8.

9. *Page 10, line 17. What does a plot of gamma vs. NO3-/H2O or Cl-/H2O look like? Especially for the nitrate effect, the dependence against the nitrate to liquid water ratio should give the most information.*

Response: We thank the reviewer for the helpful suggestions, and we further examined the relationship between the γ and the aerosol compositions. As the reviewer points out, the γ shows a positive dependence on the ratio of aerosol water to nitrate ($H_2O/NO_3^-$), which is consistent with the nitrate suppression effect and the observed dependence of uptake on nitrate concentration. However, there is no clear dependence of γ on the ratio of $Cl^-/H_2O$, which seems reasonable because of the 'cancel out' effect from both positive relationship with $H_2O/NO_3^-$ and $Cl^-/NO_3^-$, broadly following the parameterization of Bertram and Thornton (2009).

For clarity, we included the relationship of γ with $H_2O/NO_3^-$ in the revised text, and also added the plots of γ vs. $H_2O/NO_3^-$ and $Cl^-/NO_3^-$ in the supplementary, as follows,

"A moderate negative dependence (*r* = 0.54) of determined γ on aerosol nitrate concentration can be inferred, with lower values of γ associated with higher nitrate content (cf. Figure S2a). This pattern is consistent with the nitrate suppress effect on $N_2O_5$ uptake identified from previous laboratory studies (Mentel et al., 1999), and also similar to the anti-correlation of γ and nitrate from tower measurements in the USA and aircraft measurements over the UK (Wagner et al., 2013; Morgan et al., 2015). The relationship between the γ with the aerosol water to nitrate ratio also exhibits consistent trend with the previous observations and parameterizations (e.g., Bertram and Thornton, 2009; Morgan et al., 2015), with increasing uptake as the ratio increases (Figure S2b)."

[Figure]

**Figure S2: Relationship between derived $\gamma_{N2O5}$ from the measurements with (a) the molar concentration of aerosol nitrate and (b) the molar ratio of aerosol water to nitrate during the study period.**

[Figure]

**Figure S3: Relationship between derived $\gamma_{N2O5}$ from the measurements with (a) the molar concentration of aerosol chloride and (b) the molar ratio of aerosol chloride to nitrate during the study period.**

10.    *Page 11, equation (6). The analysis measured ClNO2 production relative to N2O5 loss. The denominator is difficult to determine with certainty, and especially if the N2O5 loss rates are too large (see concerns about aerosol surface area and NO3 loss to VOC above), the analysis will produce too small a value for phi(ClNO2). These caveats should be noted. There should be production of aerosol nitrate or nitric acid together with the N2O5 loss. Are any trends in aerosol nitrate or NOz (=NOy-NOx) during the periods of ClNO2 increase available to corroborate the analysis? This approach could be more quantitative than one based on N2O5 loss.*

Response: We agree with the reviewer that the potential overestimation of $N_2O_5$ loss rate could result in smaller yield value for $ClNO_2$. We were aware of the possible uncertainties in the estimation, and have applied an alternative approach to derive the $ClNO_2$ yields from the ratio of observed enhancements of $ClNO_2$ and total nitrate (aerosol $NO_3^-$ and $HNO_3$), according to the method suggested by Riedel et al. (2013). This approach does not need to determine the steady state $N_2O_5$ loss rate and requires to quantify the branching ratio of observed production of $ClNO_2$ and total nitrate. The derived $\phi$ values from this approach show comparable results with the original analysis, and the results from two approaches exhibit reasonable agreement with a RMA slope of $0.78 \pm 0.08$, with $r^2$ of 0.73. Most of the differences between two groups of data are within 40%. Although either approach requires assumptions and would introduce some uncertainties, the general consistency can serve as a check to corroborate the yield analysis. Given the low-resolution data of aerosol nitrate and gaseous $HNO_3$ in the present work that could bias the derived total nitrate enhancement, the production ratio approach will only be used as a reference to validate the reliability of the results, and further analysis will still be based on the results from the original method.

[Figure]

**Figure S4: Comparison of estimated $ClNO_2$ yields from two different approaches: approach A using the ratio of the observed $ClNO_2$ growth rate to steady-state $N_2O_5$ loss rate based on Eq. 6; approach B using the production ratio of observed enhancements of $ClNO_2$ and total nitrate, $\phi = 2/(\Delta NO_3^-/\Delta ClNO_2 +1)$ according to Riedel et al., 2013.**

To make it clearer to the reader, we elaborated the assumptions for the methods and also clarified the caveat in the revised text. The above comparison figure is added in the supplementary, and the related section and discussions are revised to match the changes, as follow:

"To characterize the formation of $ClNO_2$ from rapid heterogeneous $N_2O_5$ uptake and sufficient particulate chloride, the yields of $ClNO_2$ ($\phi$) were examined for different plumes. For regional diffuse pollution cases, the $\phi$ defined in R3 can be estimated from the ratio between $ClNO_2$ production rate and $N_2O_5$ loss rate, as the first term in below equation.

$$\phi = \frac{dClNO_2/dt}{k(N_2O_5)_{het}[N_2O_5]} = \frac{[ClNO_2]}{\int k(N_2O_5)_{het}[N_2O_5]\,dt} \qquad (6)$$

[revised manuscript text omitted]